



# 1 Forest Diversity and Environmental Factors Shape Contrasting
# 2 Soil-Litter BVOC and Methane Fluxes in Three Central
# 3 Amazonian Ecosystems

Débora Pinheiro-Oliveira[1*], Hella van Asperen[2,3], Murielli Garcia Caetano[4], Michelle Robin[2],
Achim Edtbauer[5], Nora Zannoni[6], Joseph Byron[5], Jonathan Williams[5], Layon Oreste Demarchi[7],
Maria Teresa Fernandez Piedade[7], Jochen Schöngart[7], Florian Wittmann[8], Sergio Duvoisin-
Junior[9], Carla Batista[9], Rodrigo Augusto Ferreira de Souza[10], Eliane Gomes Alves[2,1*]
[1] Graduate Program in Climate and Environment, National Institute for Amazonian Research, Manaus, Brazil
[2] Department of Biogeochemical Processes, Max Planck Institute for Biogeochemistry, Jena, Germany
[3] Institute for Environmental Physics, University of Bremen, Bremen, Germany
[4] Graduate Program in Tropical Forest Sciences, National Institute for Amazonian Research, Manaus, Brazil
[5] Atmospheric Chemistry Department, Max Planck Institute for Chemistry, Mainz, Germany
[6] Institute of Atmospheric Sciences and Climate, National Research Council (CNR-ISAC), Bologna, Italy
[7] Coordination of Environmental Dynamics, National Institute of Amazonian Research, Manaus, Brazil
[8] Department of Wetland Ecology, Karlsruhe Institute of Technology, Rastatt, Germany
[9] Department of Chemistry, Amazonas State University, Manaus, Brazil
[10] Department of Meteorology, Amazonas State University, Manaus, Brazil
*Correspondence to: Débora Pinheiro Oliveira (dpinheiro@bgc-jena.mpg.de); Eliane Gomes Alves (egomes@bgc-
jena.mpg.de)

## 20 Abstract

Biogenic volatile organic compounds (BVOCs) play a crucial role in biosphere-atmosphere
interactions and the global carbon cycle. While vegetation is recognized as the primary source of
BVOC fluxes in forest ecosystems, recent studies suggest that the carbon-rich soil-litter
compartment contributes significantly. However, these fluxes, their underlying drivers, and their
variability across forest types remain poorly understood, with measurements still scarce—
particularly in the Amazon rainforest, the world's largest source of BVOCs. In this study, we
investigated soil-litter BVOC and methane fluxes and their potential drivers—including nutrient
content, microbial biomass, soil temperature and moisture—across three forest types in central
Amazonia: white sand forest (WS), upland forest (UP), and ancient river terrace forest (AR). Our
results showed distinct flux patterns among forest types. WS exhibited both high emissions and





consumption of gases, notably high acetaldehyde and methane emissions, and strong isoprene and
monoterpene uptake. UP showed lower overall fluxes, with moderate emission and consumption
of DMS, isoprene, and acetaldehyde. AR presented no significant fluxes. Linear models identified
soil moisture and temperature as the primary drivers of fluxes in WS, while microbial biomass was
the main driver in UP. Our measurements suggest that, despite covering a relatively small area in
the Amazon basin, WS can be a significant ecosystem for BVOC and methane fluxes, regulated
by soil moisture and temperature. Our findings underscore the need to account for forest-type-
specific fluxes when modeling BVOC and methane emissions in the Amazon, particularly under
changing climate conditions.
**Key words**
Amazon rainforest; Biogenic volatile organic compounds (BVOC); Methane ($CH_4$); rain-induced
emissions; Soil-litter fluxes; Forest heterogeneity; Soil-litter microorganism

## 1. Introduction

Soil and litter constitute an ecological compartment that plays a crucial role in gas fluxes
of biogenic volatile organic compounds (BVOCs) (Peñuelas et al., 2014; Tang et al., 2019) and
greenhouse gases (GHGs) (Fan et al., 2020, 2024). Biological and physical processes are essential
in soil and litter BVOC and GHG fluxes. In terms of biological processes, roots release BVOCs
for communication, defense against herbivory and symbiotic relationships (Gfeller et al., 2013;
Lin et al., 2007; Rasheed et al., 2021; Steeghs et al., 2004; Tang et al., 2019; Trowbridge et al.,
2020), and soil microorganisms produce BVOCs for communication and ecological interactions
(e.g., defense and competition), with these compounds also being released as residual metabolic
products (Isidorov & Jdanova, 2002; Leff & Fierer, 2008; Liu et al., 2024; Monard et al., 2021).
GHGs, such as methane ($CH_4$) and carbon dioxide ($CO_2$), are produced by microorganisms in the
soil. Methane fluxes  are primarily driven by methanogenic (archaea) and methanotrophic
microorganisms in anaerobic and aerobic environments, contributing to the global methane budget
(Conrad, 2009; Hofmann et al., 2016). The decomposition of litter also influences BVOC and
GHG fluxes; particularly physical factors, such as temperature and soil moisture, greatly impact
litter decomposition by influencing the activity of microorganisms, a process that also releases
BVOCs and GHGs (Greenberg et al., 2012; Tang et al., 2019; Mäki et al., 2017; Asensio et al.,





2008). Temperature directly affects gas production and consumption (Conrad, 2009), the evaporation of stored compounds (Aaltonen et al., 2011), and the desorption from leaf litter tissue and soil organic matter (Bachy et al., 2018; Schade & Goldstein, 2001; Tang et al., 2019, Warneke et al., 1999). Soil moisture affects microbial activity (Abis et al., 2020; Liu et al., 2024) and BVOC adsorption (Jiao et al., 2023), thereby directly affecting the magnitude of soil gas fluxes (Conrad, 2009; Liu et al., 2024; Pugliese et al., 2023; Shah et al., 2024; Svendsen et al., 2016). In addition to changes in soil moisture, precipitation events can induce BVOC emissions, e.g., by pushing stored soil BVOC gases out of the soil pore space (Miyama et al., 2020).

Soil type can also influence gas fluxes, with sandy soils facilitating BVOC volatilization and retention due to larger pore spaces that promote water movement and faster evaporation under higher temperatures (Onwuka, 2018). For example, soil texture influences the relationship between methane and soil moisture, with methane emission fluxes being generally higher in sandy soils than in clay soils (Cai et al., 1999), probably due to their larger pore size, which facilitates gas diffusion (Rosace et al., 2020). Additionally, changes in vegetation cover also impacts gas fluxes (Gomes Alves et al., 2022). Plant species composition influences BVOC emissions in terms of chemical composition and emission rates (Bao et al., 2023; Mu et al., 2022; Zhang et al., 2024), and since different soil types often support distinct vegetation (Rodrigues et al., 2018), soil-litter gas fluxes are expected to vary across forest types (Wachiye et al., 2020).

The Amazon basin is the largest source of BVOCs to the global atmosphere (Guenther et al., 2012). BVOCs are crucial for understanding climate dynamics due to their role in atmospheric chemistry. They contribute to the formation of secondary organic aerosols (SOAs) and influence cloud properties, which in turn affect global climate patterns (Fuentes et al., 2000; Jimenez et al., 2009). Although vegetation is considered the main source of these compounds, with large effects on the above-mentioned atmospheric processes, some studies have shown that soils are as important as plants for BVOC emissions (Penuelas et al., 2014).

The Amazon basin has different soil types (Quesada et al., 2011), which determine forest structure (Quesada et al., 2012) and plant species composition (Ter Steege et al., 2013), resulting in a mosaic of different forest types throughout the basin (Oliveira-Filho et al., 2020). These different forest and soil types have been little - or not at all investigated for soil-litter BVOC and



GHG fluxes and, therefore, are not included in model estimates. In this sense, studies integrating
biological and physical measurements are essential to understand the processes controlling soil-
litter BVOC and GHG fluxes across Amazonian forest types. Quantifying BVOC emissions from
soil is essential for accurately modeling these processes and predicting their effect on climate, as
soil can be a significant source of BVOCs.
With a unique set of measurements, we investigated soil-litter BVOC (acetaldehyde,
methanol, m/z 42, dimethyl sulfide, isoprene and monoterpenes) and GHG ($CH_4$ and $CO_2$) fluxes,
soil and litter nutrient content and microbial biomass, and soil temperature and moisture from three
forest types in central Amazonia: (*i*) ancient river terrace forest - a forest that was flooded in the
past and is no longer flooded due to changes in the river course (paleoigapó); (*ii*) white sand forest
(locally called *campinarana*) - a less common forest type that occupies about 5% of the Amazon
basin (Adeney et al., 2016); and (*iii*) upland forest (locally called terra-firme) - the most common
forest in Amazonia, with the highest plant species richness (Emidio et al., 2016; Luize et al., 2018).
We aimed to answer the following questions: (*i*) what is the emission/consumption of gases
(BVOCs, $CO_2$, and $CH_4$) in magnitude and chemical diversity, and?; and (*ii*) what are the main
drivers of soil-litter gas fluxes across these three forest types in central Amazonia?
**2. Material and Methods**
**2.1 Site Description**
This study was conducted in the MAUA–PELD experimental plots (PELD is the
abbreviation in Portuguese for long-term ecological research) (Fig.1) at the Amazon Tall Tower
Observatory (ATTO) experimental site. This site is located 150 km northeast of Manaus and is
part of the Uatumã Sustainable Reserve (USDR), which covers an area of 424,430 hectares
(Andreae et al., 2015). The climate is tropical humid, with average annual rainfall of 2,376 mm
and a temperature of 28°C (Botía et al., 2022). There are two distinct seasons, the wet season from
December to May and the dry season from July to October, with transition seasons in between.
The ATTO site contains three dominant non-flooded ecosystems: a dense upland forest on the
plateau, with an elevation close to 100 m (*terra-firme*); a white sand forest (*campinarana*); and
another type of *terra-firme* vegetation that developed on the lower-laying ancient river terraces
(ancient river terrace forests) (Fig. 1) (Andreae et al., 2015).

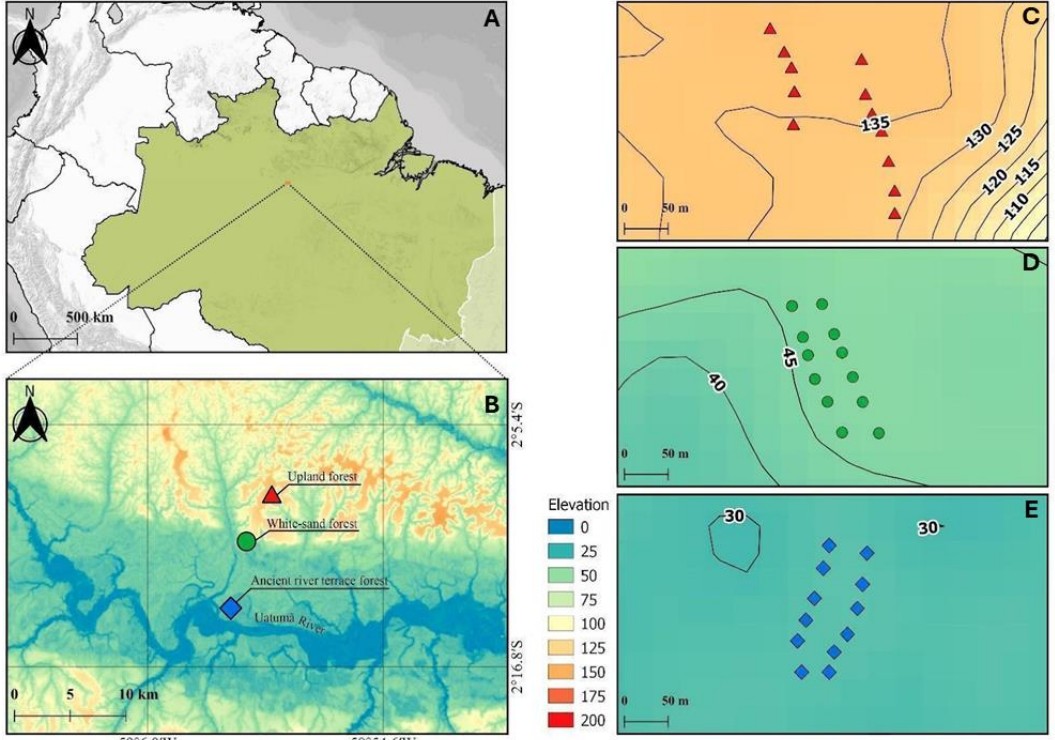


**Figure 1.** (A) Location of the ATTO site. (B) A map illustrating the locations of the different forest
types evaluated in this study: upland forest, white sand forest, and ancient river terrace forest; and
showing the Uatumã River, a tributary of the Amazon River. (C), (D) and (E): The distribution of
sampling points along the transects in each forest type (upland forest - top, white sand forest -
middle, and ancient river terrace forest - bottom); black lines and numbers indicate the elevation
(above sea level – a.s.l.).

Topography is critical to soil formation in the central Amazon region (Quesada et al., 2009). At
the ATTO site, a clear topographic gradient is associated with different soil characteristics (Fig.
1). In the ancient river terrace forest, soil contains more silt and clay (39% sand, 37% silt, 23%
clay) in comparison to the adjacent sandy white sand forest soils (57% sand, 40% silt, 1.50% clay).
Upland forest soils are more clayey and contain very little sand (13% total sand, 34% silt, 52%
clay) (data from this study; supplementary material, Table S1). Upland forest soils, which are
predominantly ferrasols, are known to hold more water than other tropical soils, benefiting forest



activity during the dry season (Quesada et al., 2009). Ancient river terrace forest soils are typically
allisols, younger and richer in nutrients compared to upland ferralsols (Andreae et al., 2015). White
sand forest soils are arenosols, characterized by high permeability and low water retention, with
low specific heat capacity and often nutrient-poor organic layers (Quesada et al., 2011). The study
area in the white sand forest has high water table variability, with a hard subsoil layer that restricts
drainage and can flood the root system during the wet season (Demarchi et al., 2022).

**2.2 Sampling Design**

For each forest type, a PELD-MAUA plot (~1 hectare) was selected, wherein two 150 m transects
were marked. Six collection points, approximately 30 m apart, were determined for each transect,
resulting in a total of 36 soil chamber measurements (Fig. 1). Additionally, three blank chambers
per transect were measured, which consisted of chambers with the same volume but with a
completely bottom-sealed collar. These blank chambers were measured simultaneously and under
the same conditions as the sample chambers covering soil and litter (Fig. 2b).
After each gas flux measurement, soil temperature (T, ºC) (TP-101, Delhi, India) and soil
volumetric water content (VWC, %) (AT SMT150, Cambridge, UK) were measured around the
collar five times, and the average was calculated. Samples from the litter and surface soil layers
were collected inside the chamber and stored for analysis of chemical and physical characteristics
and microbial biomass. Due to expected low variation and limited possibility for laboratory
analyses, nutrient samples from soil and litter (excluding carbon and nitrogen) and soil
granulometry were collected as mixed samples from two collars. To minimize diurnal variation,
each transect was measured between 8:00 and 10:00 (local time), after which collected bag samples
were processed and analyzed for BVOC and GHG concentrations. During the measurements, no
precipitation was observed, but one large rain event occurred just before the measurement of
transect 2 of the white sand forest.

**2.3 Flux Chamber Measurements**

The flux chambers used in this study were produced by the Max Planck Institute for
Biogeochemistry and were made of 100% stainless steel (Fig. 2), with a total volume of 21 L and
a surface area of 855 cm$^2$ (0.0855 m$^2$). Two Teflon inlets were connected to the top of the chamber,





and inside the chamber was a fan that provided air mixing of the gases in the chamber headspace.
A PTFE-coated Viton O-ring was positioned at the edge of the collar over which the chamber was
placed. The collar and the chamber were sealed together with multiple clamps to prevent outside
air from entering the chamber.

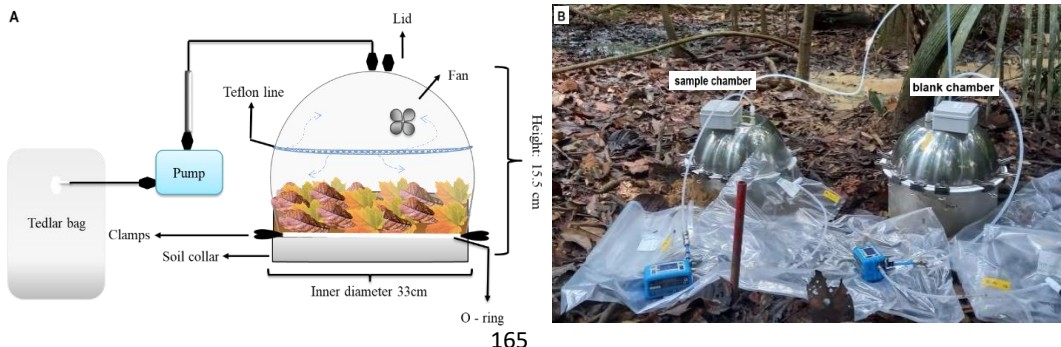


**Figure 2**. (A) Schematic of the flux chamber. (B) Photo of the measurement setup of the sample
and the blank chambers.
Before gas sampling, each soil collar was carefully installed in a non-invasive manner by gently
pressing the collar edge into the soil to avoid damage to plant shoots and roots, and then sealed
with the surrounding soil (Aaltonen et al., 2011). The collars were installed approximately 24 hours
before the measurements.
**2.4 Field gas measurements**
The gas collection took place in December 2021, at the dry-to-wet season transition. Tedlar bags
(CEL Scientific, Cerritos, CA, USA) were used to sample soil-litter gas fluxes (BVOCs, $CO_2$,
$CH_4$). An air sampling pump (GilAir® Plus, Levitt Safety, Ottawa, ON) operated at a flow rate of
500 sccm, ensuring continuous flow at the chamber outlet. After 20 minutes of chamber closure
with continuous flow, a sampling bag was connected to the outlet of the Teflon pump, and a 5 L
sample was collected over 10 minutes. At the end of the 30-minute process, a total of 15 L of air
had flown through the chamber, of which the last 5 L was used for analysis by the PTR-QMS, the
Los Gatos analyzer, and for collecting a cartridge sample (see below). For logistical reasons,
measurements were conducted with three chambers at a time, pairing two sample chambers with
one blank chamber, followed by two additional sets, resulting in measurements of six samples and
three blank chambers per day.



Before placing the lid on the collar, the chamber was manually ventilated to minimize collar-
induced $CO_2$ accumulation. The chamber was then closed, the internal fan was turned on, and the
lid was sealed with clamps. Because air was continuously extracted from the chamber headspace,
both (blank and sample) chambers had an attached 2 m long open Teflon tube, fixed approximately
2 m above the ground and at the same location, which ensured that both chambers were diluted or
affected by ambient air to the same degree. After collection, the bags were stored in a dark box for
transport to the laboratory. Gas concentration analyses were conducted on the same day; first, the
bags were measured using proton-transfer-reaction quadrupole mass spectrometry (PTR-QMS,
IONICON Analytik, Innsbruck, Austria), followed by a Los Gatos Analyzer (see section 2.5).
Subsequently, each bag was sampled using a cartridge (stainless steel tubes filled with Tenax TA
and Carbograph 5 TD adsorbents) to be later qualitatively analyzed through thermal-desorption
gas chromatography time-of-flight mass spectrometry (TD-GC-TOF-MS; Bench ToF Tandem
Ionisation, Markes International, Bridgend, UK). To maintain conciseness and ensure clarity of
our main findings, detailed descriptions of the analyses and results are presented in the
supplementary material, sections 3 and 3.1. The Tedlar bags were analyzed or sampled within 10
hours of analysis, as recommended by Beauchamp et al. (2024).

## 201 2.5 PTR-QMS measurements and Los Gatos analyzer measurements

Tedlar bags were connected to the PTR-QMS (Ionicon Analitik, Austria) for analysis of BVOC
(Lindinger et al., 1998). The PTR-QMS H3O+ mode was used for chemical ionization, which is
extremely sensitive to all BVOCs that have a higher proton affinity than water, covering most
volatile organic compounds (Edtbauer et al., 2021). Seven compounds were analyzed (Table 1).
The PTR-QMS was operated under standard conditions at 2.3 mbar, and E/N 120, with 60°C, with
a drift tube voltage of 600 V. During each PTR-QMS measurement cycle, the following specific
protonated mass-to-charge ratios (m/z) were measured, 21 ($H_3O^{18}+$), 32 ($O_2+$), and 37 ($H_2O$-
$H_3O+$), with a dwell time of 500 ms each; and Methanol (33), compound not identified (m/z 42),
Acetaldehyde (45), Dimethyl sulfide (63), Isoprene (69) and Monoterpenes (137), with a dwell
time of 1 second. We measured approximately 17 cycles for each bag. Mass identifications were
based on the available literature (Warneke et al., 2015), and were consistent with a PTR-MS mass



library database - GLOVOC (Yañez-Serrano et al., 2021) and gas calibration with certified
standards.
**Table 1**.   Compounds analyzed by the PTR-QMS

| BVOC | Chemical formula (H+) | m/z | Group |
|------|----------------------|-----|-------|
| Methanol | $CH_4O+$ | 33 | Alcohol |
| not identified |  | 42 | N-compound |
| Acetaldehyde | $C_2H_4O+$ | 45 | Aldehyde |
| Isoprene | $C_5H_8+$ | 69 | Alkenes |
| Dimethyl sulfide | $C_2H_6S+$ | 63 | Organosulfides |
| Monoterpenes | $C_{15}H_{16}+$ | 137 | Alkenes |


Calibrations were performed before the experiment using a multi-component calibration mix
containing various known concentrations (supplementary material; Table S2) (Apel-Riemer
Environmental, Inc.). Four-point calibration curves were generated by diluting the
multicomponent with synthetic air, humidifying the air stream with a water bubbler filled with
distilled water, and controlling the flow with mass flow controllers (0, 1, 3, and 5 ppb)
(supplementary material; Fig. S1). Curves were calculated considering the normalized counts per
second as a function of the mixing ratio. Previously, some compounds important for soil-litter
processes (Peñuelas et al., 2014), - such as acetone, ethanol, and formaldehyde - were considered
for this study, but they did not show a good fit, they were excluded from this work.
The mass m/z 42 can be attributed to acetonitrile; however, acetonitrile is usually considered a
biomass-burning tracer/or, more generally, a compound of anthropogenic origin (Huangfu et al.,
2021). Acetonitrile can be produced in the oceans (Sanhueza et al., 2004) and can also be
consumed by these ecosystems. It can be produced by microorganisms (Raio et al., 2020), but
there is a lack of evidence to support its emission from the soil. When using the PTR-QMS to
measure m/z 42, it is essential to consider the possibility of interference from fragments and side
reactions (Dunne et al., 2012). Consequently, it remains uncertain whether the signal at m/z 42
was due to acetonitrile since the instrument cannot distinguish between isobaric compounds.
However, we decided to present this mass in our results (section 3), as our measurements showed
substantial amounts of it. In addition, the mass 63 is attributable to DMS. However, earlier studies





in the humid Amazon have found that acetaldehyde (mass 45) can form an agglomerate with water,
resulting in the same mass (63). Thus, results for mass 63 attributed to DMS can be strongly
influenced by acetaldehyde. We have studied the relation of the masses 63 and 45, and found a
correlation coefficient of 0.51, indicating that a part of the observed 63 could indeed be
acetaldehyde. Since we expect that a considerable part of the mass 63 is still originating from
DMS, we focus our discussion of the mass 63 on the possible sources and sinks of DMS.
After PTR-QMS analysis, the bags were connected to a Los Gatos Ultraportable analyzer to
measure the mixing ratios of $CH_4$ and $CO_2$. The sample bag air was measured for 3 minutes with
an airflow of ~ 0.1 LPM, and an average was taken from the last 2 minutes of the measurement.
**2.6 BVOC & GHG flux calculation**
To calculate BVOC and GHG fluxes, the Volumetric Mixing Ratios of the blank chamber bags
(VMRb) were subtracted from the sample chamber bags (VMR):
dVMR = VMR – VMRb                                                    (1)
in which VMR is expressed in pptv or ppbv. By subtracting the mixing ratios of a blank chamber,
dVMR represents the concentration difference attributable solely to soil and litter fluxes, corrected
for potential chamber effects or the influence of ambient air entering the system. A dilution effect
due to the constant sample flow is expected to exist but, at most, may lead to a slight
underestimation of our fluxes. To convert dVMR to fluxes, we used:
F=dVMR * N * (V / A)*(1/T)                                           (2)
where N is the value of fixed molar volume at 25 °C (24.8 L $mol^{-1}$; 40.3 mol $m^{-3}$), V is the chamber
volume (0.021 $m^3$), A is the chamber area (0.0855 $m^2$), and T is the average sampling time (25
min), giving fluxes in nmol $m^{-2}$ $min^{-1}$, then converted to ng $m^{-2}$ $h^{-1}$.
**2.7 Soil and Litter Analyses**
The Thematic Laboratory of Soils and Plants (LTSP, INPA) analyzed soil and litter nutrient
content according to adapted protocols (EMBRAPA, 1999). The nutrients - iron ($Fe^{+2}$), calcium
($Ca^{+2}$), magnesium ($Mg^{+2}$), zinc ($Zn^{+2}$), potassium ($K^+$), manganese ($Mn^+$), phosphorus (P), and
aluminum (Al) - were determined by digestion with a nitro-perchloric acid solution (Malavolta et
al., 1989). Total phosphorus (P) was quantified using colorimetry (Murphy & Riley, 1962; Olsen



& Sommers, 1982) and measured using a UV spectrophotometer (Model 1240, Shimadzu, Kyoto,
Japan). Potassium (K), calcium (Ca), and magnesium (Mg) concentrations were determined by
atomic absorption spectrophotometry (AAS, 1100 B, 250 Perkin Elmer, Ueberlingen, Germany),
as described by Anderson and Ingram (1993). Soil carbon and nitrogen content was determined by
the Routine Measurements & Analyses Lab (RoMA, MPI-BGC) with the elemental analyzer
"varioEL" (Elementar Analysensysteme GmbH, Elementar-Straße 1, D-63505 Langenselbold,
Germany). Soil porosity was analyzed using the pycnometer method described in Flint & Flint
(2002). The amount of water was corrected for soil density.
For analysis of soil and litter microbial Carbon, Nitrogen, and Phosphorus (C, N, and P) contents,
2g of fresh litter and 5g of fresh soil were used from each sample chamber. These were separated
into fumigated and non-fumigated samples. The fumigated samples were left with chloroform for
24 hours and then divided into two sub-samples. For first, 50 mL of KCl (Potassium Chloride) was
added, and total C and N were extracted, and for the second, 50 mL of $NaHCO_3$ (Sodium
Bicarbonate) was added for total P extraction. Following the same extraction protocol, the non-
fumigated samples were prepared for direct extraction without going through the 24-hour
fumigation period. Microbial C, N, and P content was estimated in fumigated and non-fumigated
extracts from the difference in organic C, N, and total P measured by a TOC/TN analyzer
(Jenkinson et al., 2004). The extraction of the microbial biomass was performed at INPA, and the
analyses were done by the Routine Measurements & Analyses Lab (RoMA, MPI-BGC).
**2.8 Statistical analyses**
A total of 36 samples were evaluated (n = 12 per forest type). Gas fluxes were first correlated with
potential predictors (soil and litter characteristics, Table 2), revealing variations between forest
types. Separate regression models were built for each forest type to maximize predictive ability,
with variable selection based on the following criteria: 1) given the statistical power limitation of
models (n = 12), the maximum number of independent variables possible to include was two; thus,
2) we tested all models with one or two independent variable combinations; 3) finally, we chose
the models which showed no multicollinearity and had the highest adjusted R-squared and lowest
Akaike's information criterion (AIC). The "ols_step_all_possible" function from the "olsrr"
package (Hebbali, 2024) was used, and multicollinearity was assessed via VIF (<2.5; Hair et al.,
2009). Principal Component Analysis (PCA) and Pearson's correlation (Hmisc package; Harrell,





2018) were performed to explore variable interactions. Variations within forest types (e.g.,
between transects) were analyzed using t-tests for normal data and Kruskal-Wallis tests for non-
normal data, with a significance level of 0.05. All analyses were conducted in R (v4.3.0; R Core
Team, 2023).

**Table 2.** Variables, their respective codes, and units.

| Variable | Code | Unit |
| --- | --- | --- |
| Soil carbon | c_soil | % |
| Soil nitrogen | n_soil | % |
| Soil phosphorus | p_soil | P mg/kg |
| Soil potassium | k_soil | $K^+$ mg/kg |
| Soil calcium | ca_soil | $Ca^{+2}$ mg/kg |
| Soil magnesium | mg_soil | $Mg^{+2}$ mg/kg |
| Soil aluminum | al_soil | $Al^{+3}$ mg/kg |
| Soil iron | fe_soil | $Fe^{+2}$ mg/kg |
| Soil zinc | zn_soil | $Zn^{+2}$ mg/kg |
| Soil manganese | mn_soil | $Mn^{+2}$ mg/kg |
| Soil ph | ph_soil | pH |
| Soil temperature | soil_temp | Celsius |
| Soil moisture | soil_moisture | % |
| Litter carbon | c_litter | % |
| Litter nitrogen | n_litter | % |
| Litter calcium | ca_litter | $Ca^{+2}$ mg/kg |
| Litter magnesium | mg_litter | $Mg^{+2}$ mg/kg |
| Litter potassium | k_litter | $K^+$ mg/kg |
| Litter iron | fe_litter | $Fe^{+2}$ mg/kg |
| Litter zinc | zn_litter | $Zn^{+2}$ mg/kg |
| Litter manganese | mn_litter | $Mn^{+2}$ mg/kg |
| Microbial biomass soil carbon | c_mic_soil | g/kg |
| Microbial biomass soil nitrogen | n_mic_soil | g/kg |
| Microbial biomass soil phosphorus | p_mic_soil | g/kg |
| Microbial biomass litter carbon | c_mic_litter | g/kg |
| Microbial biomass litter nitrogen | n_mic_litter | g/kg |
| Microbial biomass soil phosphorus | p_mic_litter | g/kg |








## 3. Results

### 3.1 Comparison between forest types

The three forest types showed very different gas fluxes for BVOCs and GHGs (Fig. 3),
with the highest fluxes observed in the white sand forest. Fluxes were very low in the upland forest,
and almost no gas fluxes were observed in the ancient river terrace forest. Acetaldehyde emissions
showed the most significant differences between forest types, with emission averages of 29.911
mg m$^{-2}$ h$^{-1}$ and 0.0885 mg m$^{-2}$ h$^{-1}$ for white sand forest and upland forest, respectively, and low
consumption for the ancient river terrace forest (-0.0140 mg m$^{-2}$ h$^{-1}$). Isoprenoid (isoprene and
monoterpenes) emissions were also high in the white sand forest, and clear differences were found
between forest types concerning the speciation of monoterpenes (supplementary material; Fig. S2).

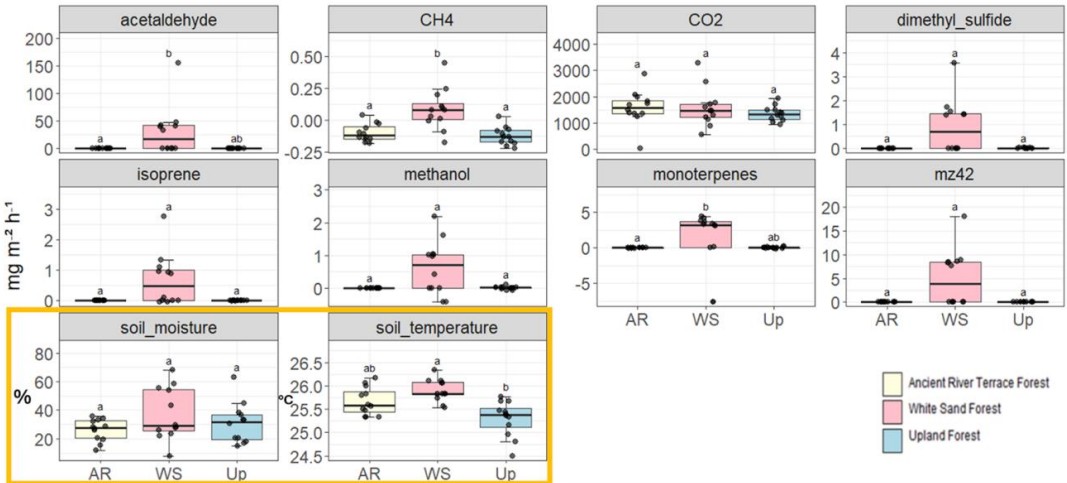


**Figure 3.** BVOC and GHG fluxes from soil and litter across the three forest types: ancient river
terrace forest (AR), white sand forest (WS), and upland forest (Up). Letters indicate statistically
significant differences in fluxes between forest types at $p < 0.05$, N=36 (Kruskal-Wallis test for
non-normal data - BVOC and GHG). The yellow rectangle represents soil moisture (soil moisture
expressed as % and soil temperature expressed as °C), and soil temperature (ANOVA test for
normal distribution). Boxes show median and first and third quartiles, with whiskers and points
distinguished at 1.5 times the interquartile range.





In the white sand forest, in addition to the high isoprenoid emissions, we also observed the
consumption of monoterpenes (-7.628 mg m$^{-2}$ h$^{-1}$, outlier in Fig. 3) and high emission of dimethyl
sulfide (DMS) (0.924 mg m$^{-2}$ h$^{-1}$, on average). Upland and ancient river terrace forests exhibited a
small amount of DMS consumption. CH$_4$ fluxes substantially varied in the white sand forest, with
large uptake and emission fluxes, while ancient river terrace and upland forests both showed
mainly CH$_4$ uptake. There were no statistically significant differences in soil moisture between the
forest types (Fig. 3); however, the white sand forest showed the highest and the lowest soil
moisture values. Differences in soil moisture and temperature were found between transects (Fig.
8). The large difference in soil texture (see supplementary material, table S1) between the sites
will affect how soil moisture translates to the amount of soil moisture available for plants and
microbes. Still, since individual transects were measured on different (consecutive) days, it is
difficult to distinguish temporal from spatial effects.

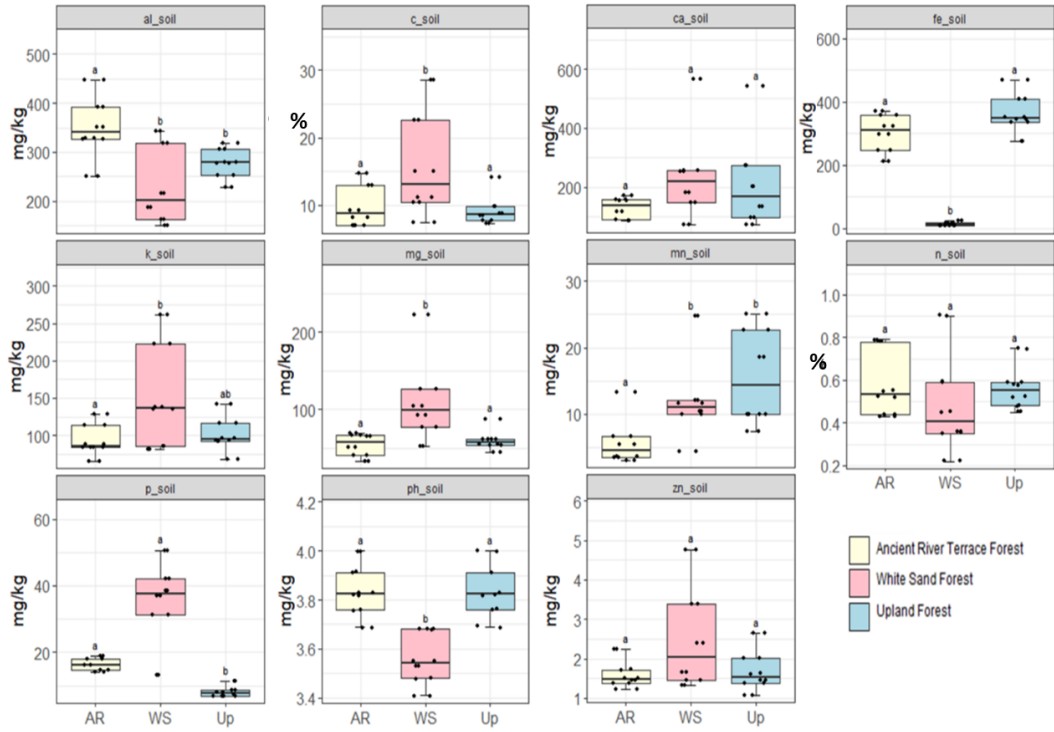


**Figure 4.** Concentrations of soil micro- and macronutrients in the three forest types: ancient river
terrace forest (AR), white sand forest (WS), and upland forest (Up). Letters indicate statistically



significant differences in nutrients between forest types at p < 0.05, N=36. (ANOVA test for
normal data (aluminum) and Kruskal-Wallis test for non-normal data (carbon, calcium, iron,
potassium, magnesium, manganese, nitrogen, phosphorus, pH, and zinc). Boxes show median and
first and third quartiles, with whiskers and points distinguished at 1.5 times the interquartile range.

340         Soil macro- and micronutrients varied considerably between the forest types, with

statistically significant differences in carbon, magnesium, phosphorus, and iron for the white sand
forest. Phosphorus content was the highest in the white sand forest compared to other forest types
(Fig. 4). All litter nutrients exhibited significant differences between forest types: upland forest
showed the highest average concentrations of calcium, iron, manganese, and zinc, while the
ancient river terrace forest had the highest nitrogen, potassium, and phosphorus concentrations,
and the white sand forest had slightly higher carbon concentrations (Fig. 5).

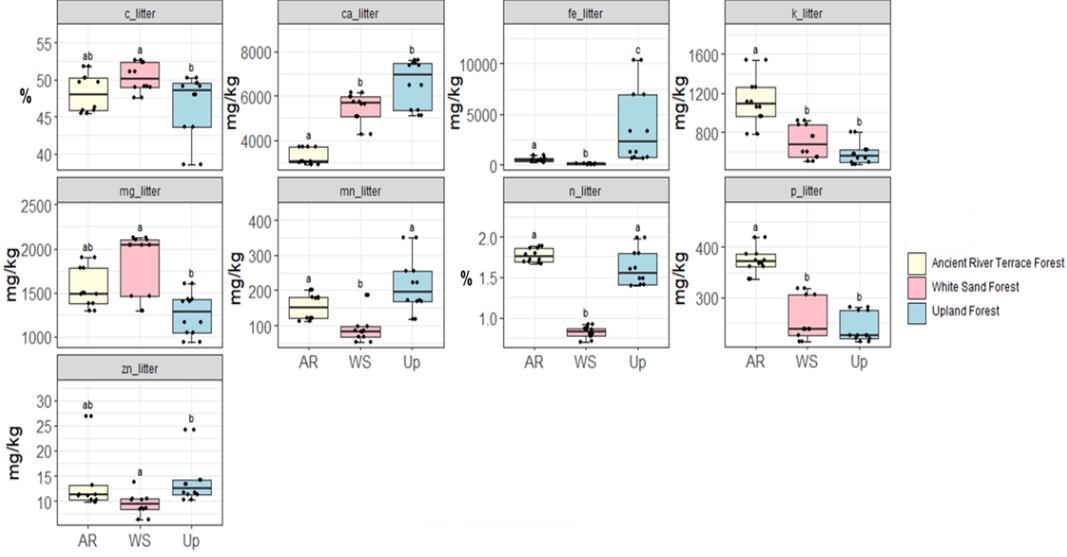


**Figure 5**. Concentrations of litter micro- and macronutrients in the three forest types: ancient river
terrace forest (AR), white sand forest (WS), and upland forest (Up). Letters indicate statistically
significant differences in nutrients between forest types at p < 0.05, N=36. (ANOVA test for
normal data - potassium and nitrogen, and Kruskal-Wallis test for non-normal data - carbon,



calcium, iron, magnesium, manganese, phosphorus, and zinc). Boxes show median and first and
third quartiles, with whiskers and points distinguished at 1.5 times the interquartile range.

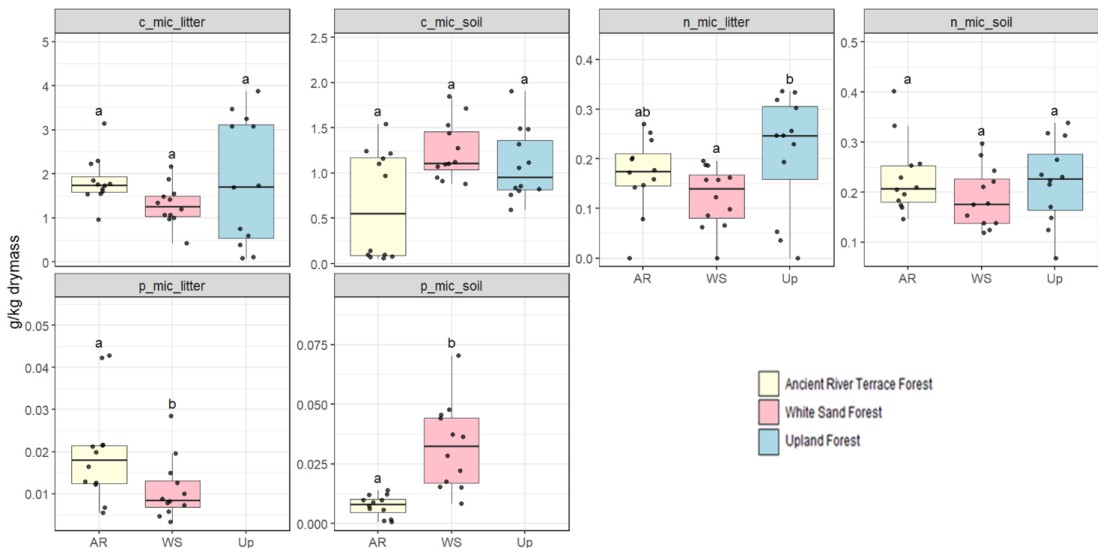


**Figure 6.** Concentrations of C, N, and P microbial biomass ($\mu g\ g^{-1}$ dry mass) of soil and litter in
the three forest types: the ancient river terrace forest (AR), the white sand forest (WS), and the
upland forest (Up). Letters indicate statistically significant differences in microbial biomass
between forest types at $p < 0.05$, N=36. ANOVA test for normal data (litter microbial carbon, soil
microbial phosphorus, and nitrogen) and Kruskal-Wallis test for non-normal data (soil microbial
carbon, litter microbial nitrogen, and litter microbial phosphorus). Boxes show median and first
and third quartiles, with whiskers and points distinguished at 1.5 times the interquartile range.
Microbial biomass (soil and litter) - measured as a potential proxy for microbial activity -
showed significant differences between forest types. Soil microbial phosphorus was significantly
higher in the white sand forest than in the ancient river terrace forest (no data for the upland forest).
In contrast, litter microbial biomass (carbon and nitrogen) was the highest in the upland forest and
the lowest in the white sand forest (carbon, nitrogen, phosphorus) (Fig. 6).
**3.2 Identification of drivers of BVOC and GHG fluxes**
**3.2.1 Principal Component Analysis**




369       A Principal Component Analysis (PCA) of soil and litter characteristics and microbial

biomass, and gas fluxes (BVOC and GHG) indicated that PC1 and PC2 axes accounted for 48.5%
of the data variation (Fig. 7). The first axis explained 31.6% and the second 12.6% (Table 3). The
PCA grouped forest types into two distinct groups: ancient river terrace and upland forests showed
considerable overlap, with lower fluxes linked to litter characteristics, soil and litter microbial
biomass, $CO_2$, and soil pH; in contrast, the white sand forest formed a separate group with higher
fluxes associated with soil temperature, moisture, and elevated levels of phosphorus, magnesium,
and potassium.

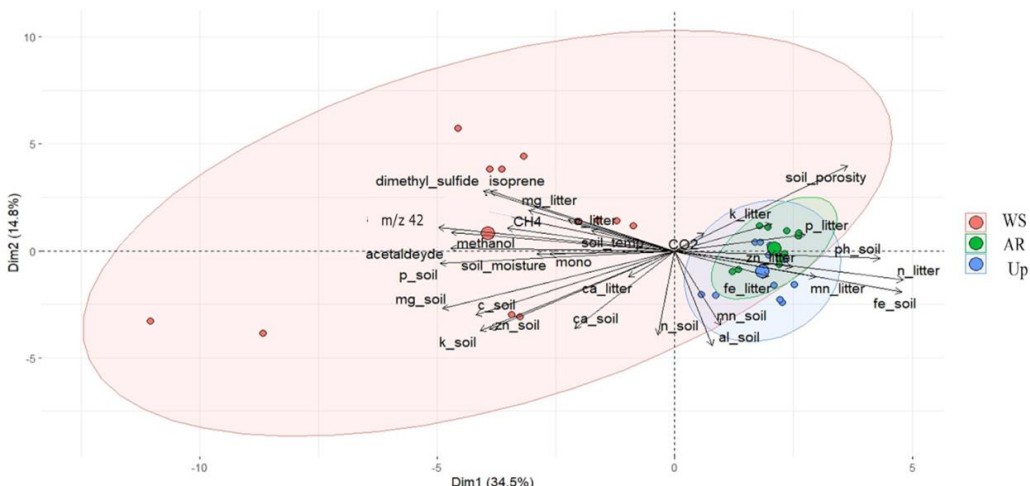


**Figure 7.** Principal Component Analysis (PCA), wherein the vectors reflect their correlation with
the variables, and the colored circles represent the average PCA score related to each ambient. The
analyzed variables are BVOCs (methanol, m/z 42, acetaldehyde, dimethyl sulfide, isoprene and
monoterpenes), greenhouse gases ($CH_4$ and $CO_2$), soil characteristics (carbon, nitrogen,
phosphorus, potassium, calcium, magnesium, aluminum, iron, zinc, manganese, pH, temperature,
and soil moisture), litter characteristics (carbon, nitrogen, calcium, magnesium, potassium, iron,
zinc, manganese), and soil and litter microorganism dynamics (soil microbial nitrogen, soil
microbial phosphorus, soil microbial carbon, litter microbial nitrogen, litter microbial phosphorus
and litter microbial carbon).





**Table 3.** Percentage correlation values extracted from Principal Component Analysis (PCA; Fig. 7).

|  | PC1 |  | PC2 |
|---|---|---|---|
| Soil iron | 83.480 | Isoprene | 52.345 |
| Litter nitrogen | 83.440 | Dimethyl sulfide | 51.718 |
| Litter manganese | 52.649 | Litter Iron | 21.662 |
| Litter phosphorus | 46.040 | Soil moisture | 20.658 |
| Soil magnesium | -79.524 | Soil carbon | -63.818 |
| Methanol | -80.097 | Soil zinc | -64.436 |
| Soil phosphorus | -83.184 | Soil aluminum | -71.054 |
| m/z 42 | -84.917 | Soil nitrogen | -71.553 |

### 3.2.2 Linear regression models for different forest types

We used linear regression models (referred to as linear models) to better understand the relationships between predictor variables and fluxes, as identified by the PCA analyses. Flux predictors showed substantial variation between the forest types (Fig. S3a, b, S4a, b, and S5a, b, in supplementary material). Model comparisons for each forest type revealed similarities between ancient river terrace (Table 4) and upland forests (Table 5). In contrast, the white sand forest (Table 6) was distinct, as also shown by the PCA analysis. In the ancient river terrace forest, linear models for gas fluxes and predictor variables showed coefficients of determination ($R^2$) above 0.8 for methanol, acetaldehyde, isoprene, and monoterpenes (Table 4). The most important nutrients for predicting gas fluxes from the ancient river terrace forest were potassium, manganese, magnesium, iron, carbon, and phosphorus. The linear models for monoterpenes had the soil microbial biomass carbon and litter potassium as predictors. The GHG models had soil temperature, soil moisture, and litter nutrients as predictors.

**Table 4.** Multiple linear regression models with soil and litter characteristics and microbial biomass as predictors of gas fluxes in the ancient river terrace forest. B = unstandardized coefficients. CI = confidence interval. $f^2$ = Cohen's $f^2$ effect size. $R^2$ = R-squared value. $R^2_{adj}$ = Adjusted R-squared value. N = 12.





| Variable | B | 95% CI | P | $f^2$ | $R^2$ | $R^2_{adj}$ |
|---|---|---|---|---|---|---|
| **Methanol** | | | | | 0.839 | 0.803 |
| Soil potassium | 0.034 | 0.021; 0.047 | < .001 | 2.222 | | |
| Litter manganese | 0.000 | 0.000; 0.000 | < .001 | 2.972 | | |
| **Acetaldehyde** | | | | | 0.829 | 0.791 |
| Soil iron | 0.000 | -0.001; 0.000 | < .001 | 0.127 | | |
| Soil manganese | 0.004 | 0.003; 0.005 | < .001 | 4.711 | | |
| **Dimethyl sulfide** | | | | | 0.635 | 0.554 |
| Soil magnesium | 0.006 | 0.001; 0.010 | 0.016 | 0.059 | | |
| Litter magnesium | -0.004 | -0.006; -0.002 | 0.004 | 1.679 | | |
| **Isoprene** | | | | | 0.969 | 0.963 |
| Soil Iron | 0.000 | 0.000; 0.000 | < .001 | 4.452 | | |
| Soil Manganese | 0.001 | 0.000; 0.001 | < .001 | 27.279 | | |
| **Monoterpenes** | | | | | 0.920 | 0.902 |
| Soil microbial carbon | 0.000 | 0.000; 0.000 | < .001 | 10.81 | | |
| Litter potassium | 0.03 | 0.000; 0.05 | 0.026 | 0.78 | | |
| **CH$_4$** | | | | | 0.276 | 0.203 |
| Soil moisture | 0.00 | 0.00; 0.00 | 0.0824 | 0.50 | 0.452 | 0.330 |
| Litter carbon | 0.00 | 0.00;0.00 | 0.024 | 0.32 | | |
| **CO$_2$** | | | | | 0.685 | 0.615 |
| Soil temperature | 9.054 | 3.101; 15.007 | 0.007 | 1.232 | | |
| Litter phosphorus | -87.940 | -156.215; -19.666 | 0.017 | 0.943 | | |


For the upland forest, gas flux models showed $R^2$ higher than 0.8 for isoprene and CH$_4$
(Table 5). Key nutrients for predicting gas fluxes included potassium, iron, manganese, and
carbon. Microbial biomass was significant in predicting gases like methanol and dimethyl sulfide.
Acetaldehyde and isoprene shared soil iron and manganese as predictors, while dimethyl sulfide
and CO$_2$ were linked to litter carbon and microbial nitrogen.
**Table 5.** Multiple linear regression models with soil and litter characteristics as predictors of gas
fluxes in the upland forest. B = unstandardized coefficients. CI = confidence interval. $f^2$ = Cohen's
$f^2$ effect size. $R^2$ = R-squared value. $R^2_{adj}$ = Adjusted R-squared value. N = 12.



| Variable | B | 95% CI | p | $f^2$ | $R^2$ | $R^2_{adj}$ |
|---|---|---|---|---|---|---|
| **Methanol** | | | | | 0.735 | 0.676 |
| Soil potassium | 0.77 | 0.41; 1.1 | 0.001 | 0.82 | | |
| Soil microbial nitrogen | 0.00 | 0.00; 0.00 | 0.002 | 1.96 | | |
| **m/z 42** | | | | | 0.679 | 0.608 |
| Soil potassium | 0.000 | 0.000; 0.001 | 0.002 | 0.687 | | |
| Soil microbial nitrogen | 0.000 | 0.000; 0.000 | 0.006 | 1.417 | | |
| **Acetaldehyde** | | | | | 0.793 | 0.748 |
| Soil iron | 0.00 | 0.00; 0.00 | < .001 | 0.05 | | |
| Soil manganese | 0.02 | 0.1; 0.02 | 0.02 | 3.80 | | |
| **Dimethyl sulfide** | | | | | 0.775 | 0.725 |
| Litter microbial carbon | 0.00 | 0.00; 0.00 | <0.001 | 1.44 | | |
| Litter microbial nitrogen | 0.00 | 0.00; 0.00 | 0.002 | 2.01 | | |
| **Isoprene** | | | | | 0.899 | 0.877 |
| Soil iron | 0.00 | 0.00; 0.00 | < .001 | 5.48e-03 | | |
| Soil manganese | 0.00 | 0.00; 0.00 | < .001 | 5.94 | | |
| **Monoterpenes** | | | | | 0.695 | 0.627 |
| Soil potassium | 1.18 | 1.3;2.3 | < .001 | 2.94 | | |
| Litter microbial nitrogen | 0.00 | 0.00;0.00 | < .001 | 4.63 | | |
| **$CH_4$** | | | | | 0.888 | 0.863 |
| Soil carbon | 0.231 | 0.00; 0.00 | 0.043 | 0.06 | | |
| Soil moisture | 0.00 | 0.00; 0.00 | < .001 | 7.91 | | |
| **$CO_2$** | | | | | 0.626 | 0.543 |
| Litter microbial nitrogen | 0.00 | 0.01; 0.06 | 0.025 | 0.25 | | |
| Litter microbial carbon | 0.00 | 0.00;0.00 | 0.006 | 1.43 | | |


In the white sand forest, models for methanol, m/z 42 and monoterpenes showed high $R^2$
values, explaining over 80% of emission variation (table 6). Key nutrient predictors included
phosphorus, nitrogen, and zinc. All emitted gases (except $CO_2$) were influenced by soil
temperature or moisture. Soil temperature was inversely related to fluxes of methanol, DMS, and
isoprene, while emissions of m/z 42, acetaldehyde, monoterpenes and $CH_4$ increased with soil
moisture.





**Table 6.** Multiple linear regression models with soil and litter characteristics as predictors of gas fluxes in the white sand forest. B = unstandardized coefficients. CI = confidence interval. $f^2$ = Cohen's $f^2$ effect size. $R^2$ = R-squared value. $R^2_{adj}$ = Adjusted R-squared value. N = 12.

| Variable | B | 95% CI | p | $f^2$ | $R^2$ | $R^2_{adj}$ |
|---|---|---|---|---|---|---|
| **Methanol** | | | | | 0.825 | 0.790 |
| Soil temperature | -0.064 | -3.4, -0.46 | < 0.015 | 4.21 | | |
| Litter phosphorus | -8.4 | -16,-0.37 | < 0.042 | 0.62 | | |
| **Acetonitrile** | | | | | 0.866 | 0.837 |
| Soil moisture | 0.187 | 0.099; 0.276 | < .001 | 2.938 | | |
| Litter nitrogen | -54.196 | -75.901; -32.491 | < .001 | 3.545 | | |
| **Acetaldehyde** | | | | | 0.653 | 0.576 |
| Soil moisture | 1.368 | 0.284; 2.452 | 0.019 | 1.022 | | |
| Litter nitrogen | -327.465 | -593.333; -61.596 | 0.021 | 0.863 | | |
| **Dimethyl sulfide** | | | | | 0.784 | 0.736 |
| Soil temperature | -0.32 | -4.9; -1.4 | 0.003 | 1.87 | | |
| Soil phosphorus | -0.06 | -0.09,-0.03 | 0.003 | 1.36 | | |
| **Isoprene** | | | | | 0.764 | 0.712 |
| Soil temperature | -1.988 | -3.6; -0.96 | 0.003 | 1.70 | | |
| Soil phosphorus | -0.05 | -0.007; -0.02 | 0.004 | 1.54 | | |
| **Monoterpene** | | | | | 0.857 | 0.825 |
| Soil moisture | 0.13 | 0.06; 0.20 | 0.003 | 0.36 | | |
| Litter nitrogen | 2.1 | 1.5; 2.8 | < 0.001 | 5.66 | | |
| **Sesquiterpene** | | | | | 0.888 | 0.863 |
| Soil moisture | 1.1 | 0.52;1.7 | 0.002 | 2.50 | | |
| Litter nitrogen | -435 | -5.75; -294 | < 0.001 | 5.43 | | |
| **$CH_4$** | | | | | 0.508 | 0.399 |
| Soil moisture | 0.0 | 0.0; 0.0 | 0.027 | 0.35 | | |
| Litter zinc | 0.0 | 0.0; 0.0 | 0.035 | 0.69 | | |
| **$CO_2$** | | | | | 0.742 | 0. 685 |
| Soil microbial carbon | 0.02 | 0.0; 0.03 | 0.029 | 1.07 | | |
| Litter zinc | 3.5 | 1.6; 5,5 | 0 .003 | 1.81 | | |






### 3.3 Spatial variability within forest types

Figure 8 shows BVOC and GHG fluxes of each transect, and Figure 9 illustrates the spatial variability within and between transects of isoprene and monoterpenes (see supplementary material, Fig. S8, S9, and S10 for other gases). In the ancient river terrace forest, BVOC fluxes were generally lower in transect 1, while GHG fluxes were similar between transects (Fig. 8). The soil temperature was higher in transect 1, while transect 2 was slightly wetter (although not statistically significant). The white sand forest exhibited the greatest variation between transects, with the highest BVOC emissions in transect 2, and significant variations in acetaldehyde, m/z 42, dimethyl sulfide, isoprene, and methanol. In addition, monoterpene fluxes showed high variation in emissions and consumption in transect 1, while transect 2 had low variation and high emissions. Furthermore, methanol was emitted in transect 1 and consumed in transect 2. In the upland forest, significant differences between transects were noted for acetaldehyde, m/z 42, dimethyl sulfide, and isoprene. For the ancient river terrace forest, the gas fluxes between transects were similar with only few variations in GHG fluxes, however significant differences in isoprene fluxes were observed.

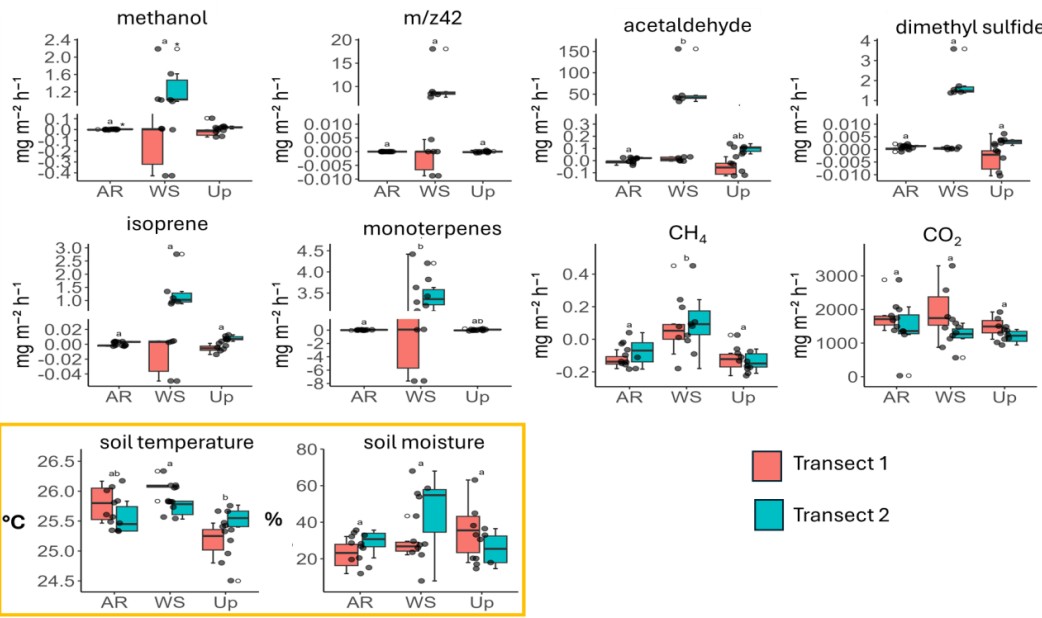



**Figure 8.** Soil and litter BVOC and GHG fluxes in each forest type - ancient river terrace forest (AR), white sand forest (WS), upland forest (Up), and transects within. Letters indicate statistically significant differences in fluxes between the forest types at $p < 0.05$, N=36 (Kruskal-Wallis test for non-normal data - BVOC and GHG). The yellow rectangle represents soil moisture and temperature plots (ANOVA test for normal distribution). Boxes show median and first and third quartiles, with whiskers and points distinguished at 1.5 times the interquartile range. Axes are broken to enhance the visibility of data variation.

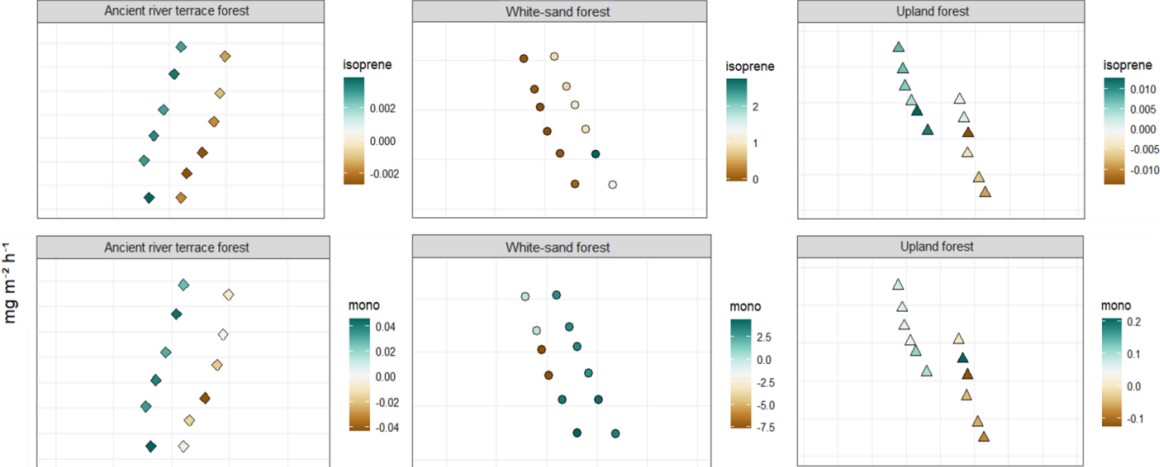

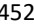

**Figure 9.** Map of the sampling points visualizing the spatial heterogeneity of BVOC fluxes in each of the three forest types. Each transect, sampled on different but subsequent days, contains six sampling points, totaling 12 measurement points per forest type. Left - Transect 1; Right - transect 2. Mono = Monoterpenes. The gas flux data are expressed in mg m⁻² h⁻¹.

## 4. Discussion

Previous studies investigated tropical soil BVOC fluxes with incubation (Bourtsoukidis et al., 2018) and fertilization experiments (Llusià et al., 2022). These studies gave insights into how BVOC soil fluxes respond to drought and nutrients, and suggested their magnitudes are much higher than anticipated. The present study found that soil-litter BVOC and GHG fluxes changed across Amazonian forest types and were influenced by differences in nutrient content, soil



moisture and temperature, and microbial biomass. The main differences in soil-litter properties, soil-litter gas fluxes, their interaction with soil-litter properties across forest types, and the significance of our findings are discussed in the following sections.

Given the extensive number of measured variables, we chose to focus our discussion on the most relevant and novel findings related to BVOC and $CH_4$ fluxes and their drivers, rather than covering all variables and fluxes. However, since these variables may still be of interest to the reader, detailed analyses are provided in the supplementary material.

**4.1 Differences in soil and litter nutrient contents across forest types**

Soil and litter properties showed strong differences between forest types. The ancient river terrace forest stood out for its high litter K and P contents, though the underlying mechanisms—such as nutrient resorption efficiency or soil nutrient availability—remain to be investigated. In the upland forest, the dominance of soil iron is likely related to the intense leaching common in oxisols, resulting in iron enrichment due to the removal of other nutrients (Mosquera et al., 2024); in addition, the formation of iron oxides reduces the mineralization of organic matter, promoting iron accumulation in the leaf litter (Li et al., 2023).

Overall, the white sand forest exhibited distinct soil properties compared to the other studied forest types. Despite its well-documented low fertility (Mendonça et al., 2015; Demarchi et al., 2022), this forest type showed unexpectedly high soil nutrient and carbon concentrations. Phosphorus levels were up to four times higher than in upland and ancient river terrace forests, potentially due to dissolved organic nutrients mitigating nutrient limitations (Lange et al., 2024). While earlier studies reported higher carbon content in upland forests (Marques et al., 2017), the white sand forest's extensive root mats may enhance carbon storage, as observed in structurally analogous ecosystems (Draper et al., 2014). Iron concentrations in the soil of white sand forest were lower than expected (Cornu et al., 1997), possibly due to spatial variability and seasonal dynamics. During the dry season, low water retention in sandy soils induces drought stress, while wet-season leaching redistributes iron, aluminum, and magnesium (García-Villacorta et al., 2016). This process can form cemented horizons, impeding drainage and elevating water tables (Franco & Dezzeo, 1994; Demarchi et al., 2022). Additionally, differences in tree species composition



between forest types may influence nutrient levels (García-Villacorta et al., 2016; Gomes Alves et
al., 2022).

## 4.2 Differences in gas fluxes across the different forest types

Our results revealed that the white sand forest showed the highest emissions and consumption of
gases, accompanied by the greatest chemical diversity in fluxes. This elevated chemical diversity
may be attributed to the distinct characteristics of the white sand forest, such as its unique
microbiome, seasonality, and species composition (Rinnan et al., 2013; Viros et al., 2021;
Vermeuel et al., 2023). Species endemic to this ecosystem may influence BVOC emission patterns
and speciation. Fine et al. (2004, 2006) showed that tree species adapted to very nutrient-poor
sandy soils highly invest in secondary metabolite compounds in defense against herbivory, since
leaves are very energetically costly for the plant. This large quantity of secondary compounds can
directly influence litter decomposition rate (Chomel et al., 2016) and probably release gases and
various compounds into the soil and water (Caetano, 2022).
Isoprenoids were emitted in considerable amounts in the white sand forest. As isoprenoids are not
expected to be emitted from soil (Bach & Rohmer, 2013; Asensio et al., 2008), the observed high
emissions could be attributed to the activity of microorganisms living in the soil and litter
(Carruthers & Lee, 2021; Hernandez-Arranz et al., 2019). In addition, it is important to note that,
although emissions in this study are expected to come from soil and litter, the contribution of root
emissions cannot be ruled out, as the main source of isoprenoids is expected to be the plant
metabolism (Pulido et al., 2012; Thulasiram et al., 2007).
A previous study on experimental rainforest soils - similar to upland forest soils - showed BVOC
soil uptake (under wet conditions) primarily for isoprenoids, carbonyls, and alcohols, as well as
soil emissions of dimethyl sulfide and carbonyl compounds such as acetaldehyde and acetone
(Pugliese et al., 2023). Our upland forest isoprene fluxes exhibited lower soil uptake (-0.1 mg m$^{-2}$
h$^{-1}$) compared to the increased uptake fluxes under drier conditions (~ -2.38 mg m$^{-2}$ h$^{-1}$) observed
by Pugliese et al. (2023). In general, our upland and ancient river terrace forests showed lower
average emissions and uptake than those reported by Pugliese et al. (2023). This could also be due
to a greater observed abundance of atmospheric isoprene as described in  Pugliese et al. (2023),
leading to a larger uptake. A study focusing on methanol fluxes in cropland soils observed values





ranging from 0.53 to 2.93 mg m⁻² h⁻¹ (Liu et al., 2024), which are higher than those observed in
the current study in upland and ancient river terrace forests but interestingly similar to the white
sand forest fluxes (1.5 mg m⁻² h⁻¹). These higher emissions in crop soils can likely be attributed to
factors such as crop species, tillage, fertilization, and irrigation, which can all influence BVOC
emission rates; whereas the high methanol emission observed in our study could be related to the
root growth of white sand forest's extensive root mats (although future studies are necessary to
confirm this hypothesis). Dimethyl sulfide emission fluxes were highest in the white sand forest
($\sim$ 0.92 mg m⁻² h⁻¹), higher than the DMS emission of 5.76 µg m⁻² h⁻¹ reported by Jardine et al.
(2015) for Amazon soils; however, it is important to note that the high magnitude of DMS fluxes
presented here might be influenced by a potential agglomerate of acetaldehyde (mass 45) with
water, resulting in the same mass as DMS (63), suggesting that future studies could make use of
techniques that differentiate these compounds. A compound with a mass-to-charge ratio (m/z) of
42 was observed in the white sand forest, but its identity could not be confirmed due to technical
limitations (Dunne et al., 2012). This m/z 42 is frequently attributed to acetonitrile, a known
biomass burning marker primarily associated with anthropogenic sources (Huangfu et al., 2021).
However, since it can also be emitted by microorganisms (Raio et al., 2020), it is possible that the
microbial communities of the white sand forest contributed to potential acetonitrile (m/z 42)
emissions.
Methane uptake was observed in the upland (-0.12 mg m⁻² h⁻¹) and ancient river terrace (-0.10 mg
m⁻² h⁻¹) forests, whereas emissions were observed in the white sand forest (0.12 mg m⁻² h⁻¹). These
results are probably explained by the shallow water table characteristic of this forest type, which
makes the soil saturated and creates an anaerobic environment that favors the growth of methane-
producing microorganisms (methanogens), contributing to the observed high emissions. In another
central Amazonia site, upland forest methane fluxes of similar magnitude were observed (-0.02 to
-0.09 mg m⁻² h⁻¹) (van Asperen et al., 2020). However,  white sand forest fluxes in their study
showed uptake instead of emission (-0.38 to -0.25 mg m⁻² h⁻¹). This difference is probably
explained by the natural variations across white sand forest ecosystems, especially concerning
water table depth (Franco & Dezzeo, 1994; Demarchi et al., 2022).
**4.3 General Drivers of Soil and Litter Fluxes**

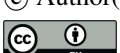



A principal component analysis (PCA) was performed to identify variables that could collectively differentiate forest types and their gas fluxes. As the PCA showed a limited capacity for differentiation due to overlapping ellipses, a further investigation was carried out using linear models (LMs). The LMs showed that soil temperature and soil moisture were important physical drivers for all three forest types, especially for the white sand forest. Since the white sand forest presents a relatively open canopy, with shorter trees and a shallow water table (Adeney et al., 2016; Rossetti et al., 2019), it often experiences extreme conditions and short-term variation, as demonstrated by the highly variable temperature and soil moisture values measured over two transects, in two subsequent days. Here, we discuss the role of soil temperature and moisture, and other potential drivers of gas fluxes.

### 4.3.1 Soil moisture and soil temperature as drivers of soil and litter gas fluxes

Soil temperature and moisture were significant drivers for most gases, especially in the white sand forest, which agrees with what has been observed in other ecosystems (Trowbridge et al., 2020; Pugliese et al., 2023; Liu et al., 2024). For example, Pugliese et al. (2023) observed that rainforest soils acted as net BVOC sinks under moist conditions and as net BVOC sources under dry conditions. By comparing two transects in the upland forest, we observed a similar pattern, with the wetter transect showing BVOC consumption while the drier transect showed emissions. However, in the white sand forest, high BVOC emissions were observed in the wetter transect, while low emissions and uptake were observed in the drier transect. This shows that Amazonian soil emissions may respond differently to soil moisture depending on the soil and forest type. Interestingly, soil moisture was shown to be a predictor for methane for all three forest types, suggesting a general mechanism influencing methane fluxes regardless of the forest type, and in agreement with several studies showing the relationship between methane flux and soil moisture (Bridgham et al., 2013; Conrad, 2009; Shah et al., 2024; Van Den Pol-van Dasselaar et al., 1998).

Generally, BVOC fluxes exhibited a positive correlation with soil moisture and an inverse relationship with soil temperature. However, since high soil moisture often coincides with low temperatures, it remains challenging to ascertain whether low temperatures or high moisture levels drive increased fluxes under field conditions. This complexity is particularly relevant given that BVOC uptake and emission are closely tied to biological processes, which typically correlate





positively with temperature. High temperatures can enhance BVOC emissions and, at the same
time, stimulate biological uptake (Baggesen et al., 2022). Notably, biological uptake may respond
more vigorously to elevated temperatures than biological BVOC emission, potentially resulting in
lower net emissions or even uptake (Penuelas et al., 2014; Jiao et al., 2023). However, as this study
was based on field measurements, wherein soil moisture and temperature are intertwined, it is not
possible to disentangle their combined effects.

**4.3.2 Forest type-specific drivers of soil and litter gas fluxes**

In general, ancient river terrace and upland forests showed many similarities in the predictors of
certain gases. In contrast, other drivers were found in the white sand forest. Here we discuss the
observed key drivers (soil potassium, carbon, phosphorus and microbial biomass) for each forest
type.
For ancient river terrace and upland forests, soil potassium was a significant factor influencing soil
fluxes, being identified as a predictor of methanol and monoterpenes. In addition, it was also
identified as a predictor of m/z 42 fluxes in the upland forest. Although we have not found studies
relating BVOC and GHG fluxes to soil potassium content, potassium is an essential macronutrient
for plant growth and metabolism. Its availability can affect plant physiological processes (Wang
et al., 2013) and microbial activity (Mazahar & Umar, 2022), which in turn can influence BVOC
production and release. In the upland forest, methane consumption fluxes correlated well with soil
carbon (in conjunction with soil moisture, as mentioned previously). Soil organic carbon is known
to play an important role in supporting methanotrophic bacteria, which are responsible for methane
oxidation (Lee et al., 2023); therefore, we suggest that the total soil carbon observed in our study
might affect methane uptake through a similar process.
Phosphorus, like carbon, is a key nutrient in the soil and significantly affected BVOC fluxes,
especially for methanol in the white sand forest. The relationship between phosphorus and BVOC
emissions is well documented for plants since the availability of phosphorus can influence the
production and emission of BVOCs (Ndah et al., 2022). However, some fertilization studies have
also shown that increasing soil nutrient status (nitrogen, phosphorus, and potassium) can modify
pH levels, affecting microorganisms and their health state (Stotzky et al., 1976) and directly or



indirectly promoting or inhibiting BVOC fluxes (Liu et al., 2024; Raza et al., 2017). Our findings in the white sand forest are consistent with this observation.

Interestingly, our results suggested that lower phosphorus levels were associated with higher isoprene emissions. The mechanisms behind this relationship remain unclear. However, Llusià et al., (2022) found that phosphorus fertilization is less efficient than nitrogen fertilization in increasing monoterpene and sesquiterpene emissions (they did not find isoprene emissions) in a tropical forest. They observed that emissions increased when the soil was fertilized only with nitrogen—consistent with a phosphorus-limited system—because excess nitrogen stimulates the enzymes responsible for producing monoterpenes and sesquiterpenes. Conversely, the addition of phosphorus likely redirected this nutrient toward plant growth, resulting in lower emissions of monoterpenes and sesquiterpenes in the phosphorus-fertilized plots compared to those fertilized with nitrogen. As in this study, there was no fertilization or a controlled environment, so we can not draw similar conclusions. However, our findings provide valuable insights into the possible interactions between phosphorus, nitrogen, and soil BVOC fluxes in tropical ecosystems. These observations align with previous studies on the influence of soil nutrients (Liu et al., 2024; Llusià et al., 2022) and we suggest future soil fertilization studies to explore these relationships across soil and forest types in Amazonia.

For the upland forest, it was found that microbial biomass was a significant driver for almost all gas fluxes, except for isoprene and methane. This aligns with previous studies that have identified microbial biomass as an important driver for soil gas fluxes (Leff & Fierer, 2008; Lamers et al., 2013; Mancuso et al., 2015; Carrion et al., 2017; Tang et al., 2019). For example, Lehnert et al. (2023) demonstrated that the degradation of organic matter is an important source of DMS emissions, highlighting the role of microorganisms associated with decomposition. Jardine et al., (2015) point out that DMS emissions in Amazonian soils are related to microbial processes, which was also reported by Kesselmeier and Hubert (2002). DMS can be produced in anaerobic environments, such as saturated soils or lakes (Carrion et al., 2017; Lehnert et al., 2023). This may explain the high emissions observed in transect 2 (wetter and more saturated) of the white sand forest, where conditions favorable to anaerobic processes are common and frequently linked to the production of sulfur compounds such as DMS. In contrast, in the drier transect 1 of the upland forest, DMS consumption was observed, suggesting the occurrence of microbial uptake processes.





Previous studies, such as the one carried out by Eyice et al. (2015), have shown that bacteria can consume carbon from DMS as an energy source. Therefore, the observed uptake may be the result of microorganisms utilizing the carbon present in DMS as an energy source, leading to uptake rather than production. This dual role of microorganisms - as both producers and consumers of DMS - highlights the complexity of sulfur cycling in terrestrial ecosystems.

From the few studies investigating the relationship between microorganisms and BVOC dynamics, it has been shown that some Proteobacteria, Actinobacteria, and Firmicutes can produce isoprene (Kuzma et al., 1995; McGenity et al., 2018). *Bacillus subtilis* can produce isoprene in response to stress; however, the mechanism is still not clear (McGenity et al., 2018). Some studies have shown that reduced soil microbial diversity can increase BVOC fluxes and alter the chemical composition of emitted compounds (Abis et al., 2020; Saunier et al., 2020; Sillo et al., 2024). Although microbial community data were unavailable in this study, we suggest that potential differences in microbial diversity have influenced emission and consumption patterns. Therefore, we strongly recommend that future studies investigate gas flux measurements with microbial community analyses to better understand these dynamics.

### 4.4. Spatial and temporal variability effects on BVOC fluxes

While efforts were made in this study to minimize the effects of spatial variability, such as by measuring equidistant points and selecting homogeneous areas, it is important to consider that spatial variability still inevitably influenced our results, as observed in other studies (Durán & Delgado-Baquerizo, 2020). With respect to temporal variability, the transects were measured at the same time (08:00–10:00 am, local time) but on consecutive days, so differences in gas fluxes, soil moisture, and temperature may partly reflect external factors like prior precipitation, cloudiness, and air temperature changes. Just before the measurement of transect 2 at the white sand forest, a heavy rainfall occurred. This coincides with the observation of significantly high BVOC emissions in this transect, while transect 1 showed much lower emissions and more uptake. Bourtsoukidis et al., (2018) also found that sesquiterpenes emissions from upland forest soils in the dry season (after a rain event) were comparable to those from vegetation, suggesting that soil moisture is a crucial factor influencing sesquiterpenes emissions from Amazonian soils. As we observed substantially high isoprene, monoterpenes and acetaldehyde emissions in transect 2 of





the white sand forest, we argue that these observed BVOC emissions represent a burst induced by the preceding rainfall event, similar to the observed increase in BVOC emissions during and immediately after rainfall in a *Ponderosa pine* plantation (Greenberg et al., 2012). Likewise, Jardine et al., (2016) observed a peak in DMS soil emissions after rainfall. Therefore, higher emissions are expected to result from the interlinked effects of soil temperature and moisture and, as described above, the possible physical effects of rainfall (Miyama et al., 2020).

**4.5 The relevance of white sand forest ecosystems**

This study showed large variability across forest types and unexpectedly high BVOC emissions from the white sand forest. Relatively few studies have been performed on white sand forests, which can partly be explained by the challenging conditions of this forest type, such as flooding and extreme temperatures, which require specific infrastructure for data collection (Adeney et al., 2016). In addition, the complex nature of this ecosystem - characterized by scattered patches of differentiated vegetation distributed within extensive upland forests (Demarchi et al., 2022) - can make access to these sites even more difficult. It is acknowledged that BVOC and GHG studies in white sand forests are limited: so far, only one study has provided data on BVOC fluxes with soil incubation lab measurements (Bourtsoukidis et al., 2018), and another measuring GHGs in situ (van Asperen et al., 2020). Despite representing only 5% of the Amazon basin area (Adeney et al., 2016) and 8% of the Reserve of this study (Demarchi et al., 2022), white sand forests are extremely important environments. Their sandy, nutrient-poor soil type has created a challenging ecosystem for plant growth (Fine & Baraloto, 2016), and this unique condition has selected specialized flora and fauna adapted to thrive in these ecosystems (Adeney et al., 2016; Demarchi et al., 2022). This high level of endemism contributes significantly to the overall biodiversity of the Amazon Basin (García-Villacorta et al. 2016). Moreover, white sand forests have been shown to play a crucial role in the chemistry of dissolved organic matter (DOM) in Amazonian blackwater rivers, linking terrestrial ecosystem processes to aquatic biogeochemistry (Simon et al., 2021). Our results showed that white sand forest gas fluxes clearly depend on physical drivers (more than other forest types), which indicates a possible sensitivity to upcoming climate extremes. Although Costa et al. (2023) did not focus specifically on the white sand forest, they showed that regions of the Amazon with shallow water tables—such as the white sand forest—can act as hydrological refuges during droughts. In these areas, higher productivity under dry conditions may help offset the substantial



carbon losses typically observed in deep water table (upland) forests during drought. Therefore, it
is crucial to recognize that white sand forests have historically been neglected, even with their
critical role in regulating the carbon cycle and maintaining Amazonian biodiversity (Rossetti et
al., 2019). As for BVOC and GHG measurements, even less information is available for this
ecosystem. However, our results suggest that white sand forests may play a significant role in both
the emission and uptake of these compounds, reinforcing their importance in regional carbon and
trace gas fluxes. Notably, a recent study reported high atmospheric isoprene concentrations in the
northwestern Amazon throughout most of the year (Wells et al., 2022) — a region characterized
by extensive and continuous white sand forest cover (Borges et al., 2014). Together, these findings
highlight the need to better integrate white sand forests into future flux studies and atmospheric
models.

## 5. Summary and future directions

Multiple interconnected factors influence BVOC and GHG soil fluxes in the central Amazon. This
study highlights the significant roles of soil and litter properties, as well as microbial biomass, in
driving these fluxes, with distinct patterns observed across forest types. Given the complexity of
the mechanisms influencing BVOC and GHG emissions, future studies should prioritize microbial
activity and diversity, along with diurnal and seasonal cycles, to better identify the key drivers of
emissions and consumption in these diverse forest ecosystems.
It is important to note that this research serves as a pilot study aimed at scoping out general trends,
and many sampling issues can be addressed in future work. For instance, utilizing a PTR-ToF-MS
could alleviate the challenges associated with measuring acetaldehyde, DMS and m/z 42. Longer
sampling periods, ideally continuous, would allow for capturing daily variations in emissions.
Surprisingly, despite being the least fertile and diverse forest type, the white sand forest exhibited
the highest uptake and emission fluxes. This is likely due to intrinsic environmental factors, such
as soil temperature and moisture, influencing microbial activity and gas fluxes, as well as the
unique vegetation composition of the white sand forest. Furthermore, external factors, like the
preceding rainfall event, could have contributed to the observed high emissions. Therefore, future
extending the measurement duration would provide a clearer understanding of how rainfall events
influence average soil BVOC emissions. The exceptionally high emissions observed in the white



sand forest may reflect short-term bursts following rainfall, which could be moderated when averaging over longer periods that capture the full range of environmental conditions in these ecosystems. Still, white sand forests may serve as BVOC emission hotspots after rain events, potentially becoming even more significant under climate change. Despite their limited area, they could have substantial ecological and atmospheric impacts. We encourage further research into this ecosystem to better understand its ecological processes and role in atmospheric dynamics, as forest BVOC fluxes influence key physical and chemical processes in the atmosphere, ultimately affecting the climate system.

**Code/Data availability**

All data supporting the findings of this study will be made available in a public repository upon publication.

**Authors' contributions**

Débora Pinheiro Oliveira, Hella van Asperen, and Eliane Gomes Alves contributed to the development and design of the study, as well as the collection, processing, and statistical analysis of the datasets. Murielli Garcia Caetano and Michelle Robin contributed to field data collection and data analysis. Achim Edtbauer helped design the methodology used in the PTR-QMS and contributed to its calibration improvement. Nora Zannoni, Joseph Byron, Jonathan Williams, Sergio Duvoisin-Junior, and Carla Batista contributed to the chemical analysis of BVOC samples with the GC-TOF-MS and GC-MS. Layon Demarchi and Maria T. F. Piedade contributed to the data analysis of the white sand forest. Maria T. F. Piedade, Jochen Schöngart, and Florian Wittmann contributed to the dataset for the initial selection of the points in the PELD-MAUA project plots where the soil chambers were installed. Rodrigo Augusto Ferreira de Souza contributed to the development of the study. All authors contributed to the writing of the manuscript.

**Competing interests**

The authors declare that they have no conflict of interest.

**Acknowledgments**



We thank the National Institute for Amazonian Research (INPA) and the Max Planck Institute for
Biogeochemistry (MPI-BGC) for their ongoing support. We would like to acknowledge the
support of the ATTO project (German Federal Ministry of Education and Research, funds BMBF
01LB1001A, 01LK1602, 01LK2101; Brazilian Ministry of Science, Technology, Innovation and
Communications; contract FINEP/MCTIC 01.11.01248.00); UEA and FAPEAM, LBA/INPA and
SDS/CEUC/RDS-Uatumã. We would like to acknowledge the contribution of the DFG project
352322796, which enabled the use of the Ultraportable Greenhouse Gas Analyzer. We also truly
thank the PELD-MAUA project (CNPq/MCTI/CONFAP-FAPs, grant number: 441811/2020-5
(CNPq); 01.02.016301.02630/2022-76 (FAPEAM)) for the plots used in this study. We would also
like to thank the field assistants, Jose Raimundo Ferreira Nunes and Sipko Bulthuis; and all the
people involved in the logistical support of the ATTO project (André Almeida, Delano Campos,
Amaury Rodrigues, Nagib Alberto, Valmir and Antonio Huxley), especially Roberta de Souza,
who were essential to the development of this study. We also thank the technicians and assistants
at INPA's soil laboratories - LTSP and Routine Measurements & Analyses Lab (RoMA, MPI-
BGC) for their valuable lab analyses. We sincerely thank Carlos Alberto Quesada for his
contributions and knowledge to this study. We also thank all the indigenous communities that have
been bravely protecting the Amazon Forest and the people from riverside communities who have
always worked together with us. Without the "mateiros", we would never have achieved our
scientific goals. Débora Pinheiro Oliveira was supported by the Coordination for the Improvement
of Higher Education Personnel (CAPES), Brazilian Ministry of Education.
**Financial support**
This research was supported by the National Institute for Amazonian Research (INPA), the Max
Planck Institute for Biogeochemistry (MPI-BGC), and the ATTO project funded by the German
Federal Ministry of Education and Research (BMBF grants 01LB1001A, 01LK1602, 01LK2101),
the Brazilian Ministry of Science, Technology, Innovation and Communications (contract
FINEP/MCTIC 01.11.01248.00), the Amazonas State University (UEA), the Amazonas Research
Foundation (FAPEAM), LBA/INPA, and SDS/CEUC/RDS-Uatumã. Additional support was
provided by the Deutsche Forschungsgemeinschaft (DFG project 352322796) and the PELD-
MAUA project funded by CNPq/MCTI/CONFAP-FAPs (grant numbers 441811/2020-5 and



01.02.016301.02630/2022-76). D.P. Oliveira was supported by the Coordination for the
Improvement of Higher Education Personnel (CAPES), Brazilian Ministry of Education.

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
