# Peer review of "Forest Diversity and Environmental Factors Shape Contrasting"

_EGUsphere, 2025_

## Author Comment (AC1)

* * *
**Reviewer 1**

In this paper, the authors studied the influence of forest diversity, environmental soil factors, and microbial biomass on soil-litter BVOC emissions and GHG fluxes in Amazonian forests. Three different ecosystems were selected, and a broad range of drivers was analyzed. Two GHGs ( $CH_4$  and  $CO_2$ ) were measured, along with various BVOCs. This paper presents novel research in a field that requires further investigation. To date, limited research has focused on BVOCs from the soil compartment of forests. This is important, as the Amazon contains the largest tropical forest in the world, and global BVOC emissions are predominantly from natural sources. Despite the extensive number of variables measured and therefore, data obtained, the authors made a selection for their main text.

We would like to thank the reviewer for the time spent on the review, and for your positive words.

**General remarks:**

Overall, the authors should ensure consistency in the use of abbreviations; for example,  $"CH_4"$  is used in some instances, while "methane" is used in others. The same applies to the naming of forests, as well as to abbreviations like DMS and LMs.

Thank you for pointing this out, we checked the abbreviations carefully and have corrected this in the revised manuscript.

Regarding the use of Tedlar bags, do you expect any sorption/losses of gases when using these bags? Was this checked for the compounds of interest, or do you have any literature that supports this?

Yes, we are aware of the potential for adsorption and gas losses when using Tedlar bags. Although we have not previously tested these effects for the compounds of interest, we have followed literature recommendations. We briefly addressed the recommended time for storing samples in Tedlar bags in our manuscript (L204-205). We suggest to expand this information in the revised version, by adding the following lines:

"Beauchamp et al. (2008) demonstrated that compound losses can occur due to adsorption onto the inner walls of the bags and diffusion through the bag material, which can compromise the stability of stored samples. To minimize these effects, it is recommended to store samples at low temperatures and analyze them within 10 hours of collection. In our study, samples were stored for a maximum of eight hours before being analyzed using the PTR-QMS and Los Gatos instruments, which is within the period of time recommended by Beauchamp et al. (2008). Furthermore, to protect the integrity of the samples, the Tedlar bags were stored in opaque stainless-steel boxes placed inside containers with controlled air temperatures. These precautions ensured minimal adsorption and losses during the storage period."

 $N_2O$  is also a very important GHG emitted/consumed by soils; however, it is not mentioned, not even in the introduction. I wonder why the authors chose not to measure it

We agree with the reviewer that it would have been very important to obtain  $N_2O$  flux data from the soil, as it is indeed an important greenhouse gas. Such measurements could have been an important contribution to this study. But, unfortunately, there was not  $N_2O$  instrument available for this study.

**I am not sure if I overlooked this, but is there any explanation for the patterns of isoprene and monoterpenes along the transects?**

This is indeed a complex question. Initially, we did not expect such distinct patterns between transects. However, the observed patterns of isoprene and monoterpene fluxes along the transects likely reflect the complex spatial and temporal variability in soil conditions, as well as external environmental drivers. These factors contributed to notable differences both within and between transects, as outlined in our results (Figures 8 and 9) and discussed in the sections 4.2 and 4.3.1.

Analyzing the results for each forest type, the ancient river terrace forest was the most homogeneous, but without clear patterns when comparing gas fluxes and tested environmental variables. In contrast, for the white sand forest, a distinct pattern emerged: BVOC uptake occurred in transect 1, while transect 2 showed emissions. We attribute this difference to the rainfall event before measurements in transect 2, which caused a 58% increase in soil moisture. This rise in soil moisture positively influenced BVOC emissions, as supported by our linear models identifying soil moisture as a key predictor of emissions in the white sand forest. Such an effect aligns with previous studies showing that soil moisture influences BVOC emissions by affecting microbial activity (Bourtsoukidis et al., 2018; Trowbridge et al., 2020). For the upland forest, we observed a somewhat similar pattern to the white sand forest; however, this response was more evident for acetaldehyde and DMS rather than all gases. Unlike in the white sand forest, soil moisture did not play a major role in the upland forest, as transect 2 was slightly drier than transect 1. Instead, microbial biomass emerged as a primary driver of BVOC fluxes in this forest type, corroborating findings that microbial community structure and activity can strongly influence soil BVOC dynamics (Insam & Seewald et al., 2010; Penuelas et al., 2014; Abis et al., 2020).

In summary, our linear models support three main conclusions: (1) soil moisture is a critical driver of BVOC emissions, especially in white sand forest; (2) microbial biomass exerts greater influence on BVOC fluxes in upland forest; and (3) the spatial heterogeneity of soil conditions and external factors such as rainfall, wind, and cloud cover, combined with measurements taken on different days, contribute to the high variability observed in fluxes. This complexity highlights the challenges in capturing soil gas fluxes accurately and underscores the need for long-term, multi-factorial studies to better understand BVOC flux dynamics in tropical forest soils.

Lastly, I am curious whether the authors were able to identify other BVOCs in their samples that could be relevant for future studies. Based on the sampling strategy and analytical techniques used, I suspect that more compounds were observed, and they could be tentatively identified using an MS library. This is, in my opinion, of great added value for future studies that decide to expand (in advance) their current list of BVOCs of interest, especially given the limited number of studies on the subject.

Yes, we indeed identified additional BVOCs in our measurements. Using cartridges, we detected compounds such as  $\alpha\textsc{-}\textsc{Cubebene},\,\alpha\textsc{-}\textsc{Copaene},\,\textsc{Caryophyllene},\,\textsc{D-Limonene},\,\beta\textsc{-}\textsc{Pinene},\,\beta\textsc{-}\textsc{Phellandrene},\,\,among others. We agree that these findings could provide valuable insights for future studies, and we encourage any interested reader to check all of them in the Supplementary Material (Figure S2). However, this data was primarily used for qualitative comparisons between forest types. Additionally, given the large amount of data presented in this study and the complexity of the study, we opted to focus on the main results to maintain conciseness. We also want to add that all datasets will become available upon publication, aiming to contribute to any future studies.$

**Specific remarks**

**Line 42: Wouldn't it be better to use the plural form soil-litter microorganisms?**

Yes, we agree with the reviewer, and we added these key words in plural form in the revised manuscript.

Line 53: What about N2O fluxes? N2O is also produced and consumed by soils. Please add a few lines about this.

We agree with the reviewer, and will add this information to the Introduction, in the revised text:

"N2O can be produced and consumed by soils through microbial nitrification and denitrification processes (Butterbach-Bahl et al., 2013; Snyder et al., 2009). These microbial processes, like those affecting other soil gases, are strongly influenced by environmental factors such as soil moisture, temperature, and nutrient availability (Saggar et al., 2013; Butterbach-Bahl et al., 2013). Together, these processes drive the net ecosystem exchange of BVOCs and GHGs between the soil-litter compartment and

the atmosphere, and the magnitude and direction of this exchange may vary across different ecosystem types."

**Line 78: Indicate the most recent BVOC budget estimate for the Amazon basin.**

We will modify this so that the text represents the most recent BVOC budget estimate for the Amazon basin:

"Global emissions of BVOCs from terrestrial vegetation are estimated at approximately 760 Tg C yr $^{\neg}$ , with isoprene ( $C_5H_8$ ) and monoterpenes ( $C_{10}H_{16}$ ) accounting for around 70% and 11% of these emissions, respectively (Tripathi et al., 2025). Isoprene is a simple building block compound emitted in large quantities, particularly by tropical forests, whereas monoterpenes—such as  $\alpha$ -pinene,  $\beta$ -pinene, and limonene—are structurally more complex (Guenther et al., 2012; Gomes Alves et al., 2016).

Lines 78-84: This paragraph could be improved to better explain BVOC dynamics and their atmospheric effects. For example, BVOCs contribute to the formation of tropospheric ozone, an important GHG. Additionally, while SOAs indeed influence cloud properties, they also affect the Earth's radiation budget by scattering incoming solar radiation or absorbing outgoing longwave radiation. These aspects should be mentioned.

We agree with the reviewer, and we will add this information to the Introduction of the revised manuscript as following:

"Once released into the atmosphere, they actively participate in atmospheric chemistry and physics, influencing climate dynamics. BVOCs react with key atmospheric oxidants—including hydroxyl radicals (OH), ozone (O3), and nitrate radicals (NO3)—to form secondary organic aerosols (SOAs) (Artaxo et al., 2022; Yáñez-Serrano et al., 2020). SOAs, in turn, have a major influence on cloud properties, enhancing cloud condensation nuclei (CCN) concentrations, which impacts precipitation patterns and alters cloud lifecycles (Liu and Matsui, 2022). Depending on their chemical composition, SOAs can also influence the Earth's radiation budget by scattering incoming solar radiation (resulting in a cooling effect) or absorbing outgoing longwave radiation. Additionally, BVOCs contribute to the formation of tropospheric ozone—an important greenhouse gas and a major air pollutant (Vella et al., 2025). Given these large-scale impacts, accurately quantifying BVOC fluxes in terrestrial ecosystems is essential for advancing our understanding of forest—atmosphere interactions and for improving Earth system models—thereby improving climate predictions."

Line 82: Support this statement with quantitative data. Is there already any estimate of BVOC emissions from vegetation vs BVOC emissions from the soil-litter compartment?

Yes, there is an estimate of BVOC emission contributions from vegetation. Tripathi et al. (2025) estimated that terrestrial vegetation emits approximately 760 Tg C yr-1 in BVOCs, with isoprene and monoterpenes contributing 70% and 11% of this total, respectively. The Amazon rainforest accounts for 40% of global BVOC emissions, making it a significant component of the global carbon cycle (Guenther et al., 2012). However, there is currently no information available on the specific contribution of the soil-litter compartment to the total BVOC budget of the Amazon Forest.

As previously pointed out, we suggest to add in the revised manuscript the following text:

"Global emissions of BVOCs from terrestrial vegetation are estimated at approximately 760 Tg C yr-1, with isoprene ( $C_5H_8$ ) and monoterpenes ( $C_{10}H_{16}$ ) accounting for around 70% and 11% of these emissions, respectively (Tripathi et al., 2025). Isoprene is a simple building block compound emitted in large quantities, particularly by tropical forests, whereas monoterpenes—such as  $\alpha$ -pinene,  $\beta$ -pinene, and limonene—are structurally more complex (Guenther et al., 2012; Gomes Alves et al., 2016). The Amazon rainforest alone contributes about 40% of global BVOC emissions, playing a critical role in the global carbon cycle (Guenther et al., 2012; Wang et al., 2024; Tripathi et al., 2025)."

Line 103: Remove "and?"

We will remove 'and'

**Line 160: Why two Teflon inlets? What were they used for?**

Thanks for pointing this out. We will add the below text in the Material and Methods section (2.3 and 2.4) for clarification:

"The two Teflon inlets were necessary to ensure the proper functioning of the semi-static chamber system. One inlet was connected to the pump for sampling, while the other one was connected to an open Teflon line to equilibrate the system and maintain internal pressure balance. It is important to mention that this semi-dynamic method does not assume a steady state condition. Additionally, a blank chamber was used to detect potential chamber effects or external influences."

**Line 161: Sentence structure is not clear.**

We agree, and we suggest rephrasing it as follows:

"Two Teflon inlets were connected to the top of the chamber, and an internal fan mixed the gases within the chamber headspace."

Lines 173-178: Rephrase for clarity. For example, in line 177, you refer to a continuous flow. Do you mean an external flow?

What we mean by continuous flow is the presence of a constant airflow within the semi-dynamic chamber system. In this setup, air accumulates inside the chamber while a pump continuously removes 500 sccm of air over 20 minutes through one of the inlets. At the same time, the second inlet remains open to allow external air to enter, thereby ensuring the system remains balanced and maintains equalized pressure.

For clarity, we rephrased:

"Gas sampling was conducted in December 2021, during the dry-to-wet season transition. Soil-litter gas fluxes (BVOCs, CO2, methane) were collected using Tedlar bags (CEL Scientific, Cerritos, CA, USA). A semi-dynamic chamber system was employed, where constant airflow (500 sccm) was ensured by an air sampling pump (GilAir® Plus, Levitt Safety, Ottawa, ON, Canada) connected to one of the chamber outlets. While the pump continuously removed air from the chamber at 500 ml/min, a secondary inlet remained open to allow external air to enter the system, maintaining balanced pressure inside the chamber. After 20 minutes of air circulation under this continuous flow, a Tedlar sampling bag was connected to the outlet, and 5 L of air was collected over the course of 10 minutes. By the end of the 30-minute sampling process, a total of 15 L of air had flowed through the chamber."

Line 180: Specify what "los gatos analyzer" is. Someone not related to the field won't recognize the name. Refer to the section number rather than "see below".

We remove the "see below" and will put the section number (2.5) in the revised text. We give more details of Los Gatos instrument in the 2.5 section, with the following text:

"After PTR-QMS analysis, the bags were connected to a Los Gatos Ultraportable analyzer to measure the mixing ratios of  $CH_4$  and  $CO_2$  with high sensitivity and rapid response times. The Los Gatos analyzer is an instrument based on laser absorption spectroscopy specifically Off-axis Integrated Cavity Output Spectroscopy (OA-ICOS), enabling ultra-sensitive, precise, and real-time measurements of trace gases in gas samples (Pohlman et al., 2021; van Asperen et al., 2024). The air in the sample bag was measured for 3 minutes with an airflow of  $\sim$  0.1 LPM, and an average was taken from the last 2 minutes of the measurement."

**Line 184: Was the ventilation performed with ambient air?**

Yes, it was. We will clarify this in the revised text as follows:

"Before placing the lid on the collar, the chamber was manually ventilated with ambient air to minimize collar-induced  $CO_2$  accumulation. The chamber was then closed, the internal fan was turned on, and the lid was sealed with clamps."

Line 220: Why was the air stream humidified, and what was the resulting relative humidity?

The air stream was humidified to better replicate the atmospheric conditions of the Amazon, where relative humidity is exceptionally high (90-100% during the wet season). Conducting the calibration with dry air would not accurately reflect the ambient air in the forest, potentially affecting the reliability and representativeness of the measurements. Relative humidity was estimated to be between 90-100%.

Line 237: Did you compare the DMS results from PTR measurements with those of GC-MS?

Unfortunately, we did not detect dimethyl sulfide (DMS) in the sorbent cartridges. It is very difficult to capture DMS with cartridges, due to high volatility, which makes it prone to breakthroughs rather than being captured.

Materials and methods: Refer to the supplementary material sections in the text using the specific section numbers.

We have modified this in the revised manuscript.

Figure 3: Consider splitting this figure into two panels: A (fluxes) and B (environmental soil variables). Also, clarify what is meant by "monoterpenes." Which compounds are included in this group?

We agree with this idea and have split the plots. Our revised figure can be found below.

⇒ AR (Ancient River Terrace Forest) ⇒ WS (White Sand Forest) ⇒ Up (Upland Forest)

About the monoterpenes, as measurements were carried out with PTR-QMS, we can not specify which compounds were included in the sample. The PTR-QMS measures the total monoterpenes found in the sample. As for more general information on monoterpenes, we added the definition of this group in the introduction of the revised manuscript:

"Isoprene is a simple building block compound emitted in large quantities, particularly by tropical forests, whereas monoterpenes—such as  $\alpha$ -pinene,  $\beta$ -pinene, and limonene—are structurally more complex (Guenther et al., 2012; Gomes Alves et al., 2016)."

Figure S2: Use the correct Greek symbols for compound names; i.e., "α" instead of "a", "γ" instead of "g", etc., and match the forest type colors with Figure 3.

□ AR (Ancient River Terrace Forest) □ WS (White Sand Forest) □ Up (Upland Forest)

Ok Thanks for pointing this out. We have corrected it in the revised manuscript.

**Line 322: Is that consumption related to a specific compound? And a specific chamber?**

The observed consumption refers to monoterpenes and DMS. This result is shown in Figure 3, which presents the boxplot for the two transects across each forest type. However, when analyzing Figure 9, we observed that monoterpene consumption occurred in two chambers of Transect 1. For DMS, consumption was observed in all chambers of Transect 1 (as shown in Figure S10, Supplementary Material).

Table 4: Soil/litter characteristics and microbial biomass only explain a small proportion of the variance in CH4. What could be the reason?

We believe the issue here is related to the forest type. Table 4 refers to the ancient river terrace forest, which contributed very little to the fluxes of most gases. As a result, linear models did not reveal clear patterns. We found an interesting relationship with a high R² for some gases (methanol, acetaldehyde, isoprene, and monoterpenes), where most of the selected drivers were characteristics of the soil and litter. However, for methane, the relationship was indeed poor, and the R² was low. We believe this may be due to spatial and temporal variability: methane fluxes are highly variable over small spatial scales and over time because of changing environmental conditions (e.g., moisture pulses,

temperature fluctuations), which can obscure direct links to biomass or soil and litter traits measured at a single time point. Additionally, spatial and temporal heterogeneity play a role: soil properties can vary greatly over small areas, and conditions change rapidly during events such as wetting or drying. Microbial communities respond quickly to these changes, and snapshot measurements of soil, litter, or biomass may not adequately capture these dynamics.

Figure S8: Keep the number of digits consistent among plots for soil temperature.

Thanks for pointing this out. We have corrected it in the revised manuscript.

**Lines 459-462: Could you support these lines with numbers?**

Yes, we cite values of BVOC from soil in section 4.2 (lines 560-586). Regarding this topic, our intention was to start the discussion by presenting the general aspects of our findings. However, specifically regarding your comment, we respond below:

Bourtsoukidis et al. (2018) reported maximum sesquiterpene emissions in upland forest soils following rain of approximately 10  $\mu$ g m-2 h-1, with model simulations reaching up to 20  $\mu$ g m-2 h-1 (Fig. 6, A). Under drier conditions, sesquiterpene emissions decreased.

Llusià et al. (2022) showed that fertilization shifted soils from a sink to a source of BVOCs, with the highest terpene emissions recorded at upper elevations during the wet season after nitrogen addition (monoterpenes: 406  $\mu g$  m-2 h-1) and phosphorus addition (sesquiterpenes: 210  $\mu g$  m-2 h-1).

We suggest to modify the text to the following:

Previous studies investigated tropical soil BVOC fluxes using incubation (Bourtsoukidis et al., 2018) and fertilization experiments (Llusià et al., 2022), showing that fluxes are higher than previously anticipated. Bourtsoukidis et al. (2018) reported maximum sesquiterpene emissions of ~10  $\mu$ g m-2h-1 after rain (model simulations up to 20  $\mu$ g m-2h-1), decreasing under drier conditions. Llusià et al. (2022) showed that fertilization converted soils from BVOC sinks to sources, with peak monoterpene emissions of 406  $\mu$ g m-2h-1 after nitrogen addition and sesquiterpenes of 210  $\mu$ g m-2h-1 after phosphorus addition.

**References**

Abis, L., Loubet, B., Ciuraru, R. et al. Reduced microbial diversity induces larger volatile organic

compound emissions from soils. Sci Rep 10, 6104. <a href="https://doi.org/10.1038/s41598-020-63091-8">https://doi.org/10.1038/s41598-020-63091-8</a>, 2020

Alves, E. G., Jardine, K., Tota, J., Jardine, A., Yañez-Serrano, A. M., Karl, T., Tavares, J., Nelson, B., Gu, D., Stavrakou, T., Martin, S., Artaxo, P., Manzi, A., and Guenther, A.: Seasonality of isoprenoid emissions from a primary rainforest in central Amazonia, *Atmos. Chem. Phys.*, 16, 3903–3925, https://doi.org/10.5194/acp-16-3903-2016, 2016

Artaxo, P., Gatti, L. V., Leal, A. M. C., Longo, K. M., Freitas, S. R. D., Lara, L. L., e Rizzo, L. V.: Atmospheric chemistry in Amazonia: the forest and the biomass burning emissions controlling the composition of the Amazonian atmosphere, *Acta Amaz.* 35, 185–196, https://doi.org/10.1590/S0044-59672005000200008, 2005.

Artaxo, P., Hansson, H.-C., Andreae, M. O., Bäck, J., Alves, E. G., Barbosa, H. M. J., Bender, F., Bourtsoukidis, E., Carbone, S., Chi, J., Decesari, S., Despres, V. R., Ditas, F., Ezhova, E., Fuzzi, S., Hasselquist, N. J., Heintzenberg, J., Holanda, B. A., Guenther, A., ... Kesselmeier, J.. Tropical and Boreal Forest Atmosphere Interactions: A Review. Tellus. Series B: Chemical and Physical Meteorology, 74, 24-163. https://doi.org/10.16993/tellusb.34, 2022.

Butterbach-Bahl, K., Baggs, E. M., Dannenmann, M., Kiese, R., and Zechmeister-Boltenstern, S.: Nitrous oxide emissions from soils: how well do we understand the processes and their controls?, *Philos. Trans. R. Soc. B: Biol. Sci.*, 368, 20130122,https://doi.org/10.1098/rstb.2013.0122, 2013.

Guenther, A., Jiang, X., Palmer, P. I., Geron, C., Seco, R., Sarkar, C., Matsunaga, S. N., Harley, P., and Wiedinmyer, C.: The model of emissions of gases and aerosols from nature version 2.1 (MEGAN2.1): An extended and updated framework for modeling biogenic emissions, *Geosci. Model Dev.*, 5, 1471–1492, https://doi.org/10.5194/gmd-5-1471-2012, 2012.

Insam, H., Seewald, M.S.A. Volatile organic compounds (VOCs) in soils. Biol Fertil Soils 46, 199–213. https://doi.org/10.1007/s00374-010-0442-3, 2010.

Lelieveld, J., Butler, T., Crowley, J., Dillon, T., Fischer, H., Gromov, S., Harder, H., Lawrence, M. G., Martinez, M., Taraborrelli, D., Williams, J., and Yáñez-Serrano, A. M.: Atmospheric oxidation capacity sustained by a tropical forest, *Nature*, 452, 737–740, https://doi.org/10.1038/nature06870, 2008.

Liu, M., and Matsui, H.: Secondary organic aerosol formation regulates cloud condensation nuclei in the global remote troposphere, *Geophys. Res. Lett.*, 49, e2022GL100543,https://doi.org/10.1029/2022GL100543, 2022.

Miyama, T., Morishita, T., Kominami, Y., Noguchi, H., Yasuda, Y., Yoshifuji, N., Okano, M., Yamanoi, K., Mizoguchi, Y., Takanashi, S., Kitamura, K., and Matsumoto, K.: Increases in biogenic volatile organic compound concentrations observed after rains at six forest sites in non-summer periods, Atmosphere, 11, 13181, <a href="https://doi.org/10.3390/atmos11121381">https://doi.org/10.3390/atmos11121381</a>, 2020.

Nascimento, J. P., Barbosa, H. M. J., Banducci, A. L., Rizzo, L. V., Vara-Vela, A. L., Meller, B. B., Gomes, H., Cezar, A., Franco, M. A., Ponczek, M., Wolff, S., Bela, M. M., and Artaxo, P.:, Major Regional-Scale Production of O3 and Secondary Organic Aerosol in Remote Amazon

- Regions from the Dynamics and Photochemistry of Urban and Forest Emissions, *Environ. Sci. Technol.*, 56, 9924–9935. doi: 10.1021/acs.est.2c01358, 2022.
- Pohlman, J. W., Casso, M., Magen, C., and Bergeron, E.: Discrete Sample Introduction Module for Quantitative and Isotopic Analysis of Methane and Other Gases by Cavity Ring-Down Spectroscopy, *Environ. Sci. Technol.*, 55, 12066–12074, <a href="https://doi.org/10.1021/acs.est.1c01386">https://doi.org/10.1021/acs.est.1c01386</a>, 2021.
- Saggar, S., Jha, N., Deslippe, J., Bolan, N., Luo, J., Giltrap, D., Kim, D., Zaman, M., & Tillman, R. Denitrification and N2O:N2 production in temperate grasslands: Processes, measurements, modelling and mitigating negative impacts. Science of The Total Environment, 465, 173-195. https://doi.org/10.1016/j.scitotenv.2012.11.050, 2013.
- Snyder, C. S., Bruulsema, T. W., Jensen, T. L., and Fixen, P. E.: Review of greenhouse gas emissions from crop production systems and fertilizer management effects, *Agric. Ecosyst. Environ.*, 133, 247–266, https://doi.org/10.1016/j.agee.2009.04.021, 2009.
- Sporre, M. K., Blichner, S. M., Schrödner, R., Karset, I. H. H., Berntsen, T. K., van Noije, T., Bergman, T., O'Donnell, D., and Makkonen, R.: Large difference in aerosol radiative effects from BVOC-SOA treatment in three Earth system models, *Atmos. Chem. Phys.*, 20, 8953–8973, https://doi.org/10.5194/acp-20-8953-2020, 2020.
- Svendsen, S. H., Christensen, T. R., Westergaard-Nielsen, A., and Elmendorf, S. C.: Biogenic volatile organic compound emissions along a high Arctic soil moisture gradient, *Sci. Total Environ.*, 573, 131–138, https://doi.org/10.1016/j.scitotenv.2016.08.100, 2016.
- Tripathi, N., Krumm, B. E., Edtbauer, A., Spracklen, D. V., and Kanzow, T.: Impacts of convection, chemistry, and forest clearing on biogenic volatile organic compounds over the Amazon, *Nat. Commun.*, 16, 4692, <a href="https://doi.org/10.1038/s41467-025-59953-2">https://doi.org/10.1038/s41467-025-59953-2</a>, 2025.
- van Asperen, H., Warneke, T., Carioca de Araújo, A., Forsberg, B., José Filgueiras Ferreira, S., Röckmann, T., van der Veen, C., Bulthuis, S., Ramos de Oliveira, L., de Lima Xavier, T., da Mata, J., de Oliveira Sá, M., Ricardo Teixeira, P., Andrews de França e Silva, J., Trumbore, S., and Notholt, J.: The emission of CO from tropical rainforest soils, *Biogeosciences*, 21, 3183–3199, <a href="https://doi.org/10.5194/bg-21-3183-2024">https://doi.org/10.5194/bg-21-3183-2024</a>, 2024.
- Vella, R., Forrest, M., Pozzer, A., Tsimpidi, A. P., Hickler, T., Lelieveld, J., and Tost, H.: Influence of land cover change on atmospheric organic gases, aerosols, and radiative effects, Atmos. Chem. Phys., 25, 243–262, <a href="https://doi.org/10.5194/acp-25-243-2025">https://doi.org/10.5194/acp-25-243-2025</a>, 2025.
- Wang, H., Liu, X., Wu, C., and Lin, G.: Regional to global distributions, trends, and drivers of biogenic volatile organic compound emission from 2001 to 2020, *Atmos. Chem. Phys.*, 24, 3309–3328, <a href="https://doi.org/10.5194/acp-24-3309-2024">https://doi.org/10.5194/acp-24-3309-2024</a>, 2024.
- Yáñez-Serrano AM, Bourtsoukidis E, Alves EG, Bauwens M, Stavrakou T, Llusià J, Filella I, Guenther A, Williams J, Artaxo P, Sindelarova K, Doubalova J, Kesselmeier J, Peñuelas J. Amazonian biogenic volatile organic compounds under global change. Glob Chang Biol. 4722-4751, doi: 10.1111/gcb.15185, 2020.

Yokouchi, Y., and Ambe, Y.: Diurnal variations of atmospheric isoprene and monoterpene hydrocarbons in an agricultural area in summertime J. *Geophys. Res.*, 93, 3751–3759, <a href="https://doi.org/10.1029/JD093iD04p03751">https://doi.org/10.1029/JD093iD04p03751</a>, 1988.

---

## Author Comment (AC2)

**Responses to reviewer**

**Responses: Answer to reviewer 3**
* * *
**Reviewer 3' comments in blue**

Responses Answers from authors in black

Suggested new text blocks in italic

**Reviewer 3**

The manuscript by D. Pinheiro-Oliveira does an attempt to quantify CH4 and BVOC emissions from different forest ecosystems in the Amazon. Doing so gas samples were collected at one point in time (December 2021), along duplicated transects in three different forest types. Along with BVOC and CH4 data, data of potential drivers were collected.

In principle this are interesting data. Despite no major methodological flaws are presented in the paper/with the data set, I have several concerns. These concerns are listed below, both in general and specifically.

First, the data set has a very limited temporal coverage, i.e. one time point that has been repeated the next day for a spatially duplicated transect in one specific period, i.e. December 2021. As a result, one forest type has 12 samples from a very specific period. This also is the case for dynamic parameters, such as soil moisture and temperature, which are used as drivers to explain the observed BVOC and CH4 fluxes. As a result, this rather long paper is loaded with sections in the discussion that are very speculative (see examples below) that are not fully supported by data. A next step is than a danger of overstating some conclusions (see examples below) in the discussion. Indirectly the authors mention this limitation in the description of the statistical methods where they reduce the number of explanatory variables due to low amount of data per forest type, i.e. 12. In addition to this, it is also unclear using the PCA in figure 7, how predictor variables were finally selected/retained.

We appreciate the careful review and the concerns raised by the Reviewer regarding the limited temporal coverage of our dataset and its implications for dynamic parameters, as well as for the explanatory power of statistical analyses. This feedback is very constructive and has enabled us to revisit key points in the manuscript to improve clarity and to outline our study's limitations more explicitly.

Below we address each comment:

On the limited temporal coverage and sample size: We recognize that our study was conducted over a very specific period, with data collected during two consecutive days (for each forest type) in December 2021. This limited temporal (and spatial) coverage was partly due to fieldwork constraints: the measurements were physically and logistically challenging, as the team had to carry sampling material over long distances and difficult terrain.

We are aware that this restricted temporal coverage presents limitations, particularly for dynamic parameters, which can vary significantly across seasons and weather conditions. However, these measurements managed to capture spatial variability across forest types, while controlling for interseasonal variability under the same meteorological conditions. Thus, while this study does not cover large temporal variability (e.g., diurnal and seasonal) replicability, which might pose some limitations to the study's conclusions, this does not compromise the consistency of the distinct spatial patterns observed within and across forest types.

To the best of the authors' knowledge, this is the first study comparing different forest types regarding net soil/litter—atmosphere exchange of BVOCs and methane in Amazonian forest types, and can therefore be seen as an exploratory study focusing on the main differences and drivers of gas fluxes across forest types within the same season. A follow-up study could extend it through additional and longer field campaigns, or even long-term measurements. In the revised manuscript, we suggest discussing our temporal limitations in more detail; details on our revisions are given below.

On the speculation in the discussion: While it is true that some points raised in the discussion are exploratory and attempt to broaden the implications of the observed BVOC and CH4 fluxes, care has been taken to base our arguments on previous literature and the best available interpretation of our dataset. Nevertheless, we agree with the Reviewer that this aspect can be improved. To address this concern, we have reviewed the manuscript and revised speculative sections in the discussion to ensure that all statements are fully supported by the data. Details on our revisions are provided below.

On the PCA and Linear Models: For the PCA, we included all variables available in our dataset to explore the relationships between forest types and environmental predictors without imposing prior restrictions. While the PCA explained 48.5% of the total variance (PC1: 31.6%, PC2: 12.6%), the forest types did not separate completely in the biplot. This limited separation suggested that the relationships between drivers and fluxes were not fully captured by the PCA. Consequently, we applied linear regression models to identify specific drivers for each gas and forest type, as these models allow for a more targeted understanding of the variables influencing the fluxes.

To address the limitation of sample size and to maximize the statistical power of our linear regression models, we followed a systematic approach:

A maximum of two predictor variables was allowed for each gas and forest type to ensure sufficient degrees of freedom. An automated model selection process was performed using the *ols\_step\_all\_possible* function from the *olsrr* package in R, which generated up to ten possible combinations of predictor variables for each forest type and gas flux. The combinations suggested by the function were evaluated based on statistical criteria (Adjusted R², AIC, and VIF < 2.5). From these candidate combinations, the model with the best statistical parameters (highest adjusted R² and lowest AIC) was selected. The chosen variables were then tested for normality and evaluated individually for statistical significance (p-value), effect size (Cohen's f²), and overall model fit (R²). This methodology allowed us to identify strong links between drivers and fluxes.

In summary, we used the PCA as an initial exploratory tool to provide an overview of variable clustering and forest type differentiation. However, the limited explanatory power and moderate variance explained (48.5%) made it insufficient to delineate specific relationships between gas fluxes and their drivers. The subsequent linear regression models therefore complemented the PCA, providing robust, quantitative insights into the predictors associated with specific fluxes for each forest type.

Second, the method used to calculate BVOC/CH4 fluxes, i.e. collecting a large volume of gas and using a blank chamber is unusual, but not wrong per se. Likely this is needed for the large volumes that need to be collected to detect BVOCs with mass spectrometry? This could be better explained/justified in the text.

We acknowledge that it was not clearly explained why a large volume of air was collected. This is clarified below.

The large gas volume collected was necessary to ensure sufficient air for the different BVOC and GHG analyses conducted using multiple instruments:

- **PTR-QMS:** Measurements of BVOCs by the PTR-MS required a sufficiently large sample volume to capture BVOC concentrations with high precision.
- Los Gatos: Due to its relatively high sample flow (0.1 L min-1), at least 0.5 L of gas was needed to determine CH4 and CO2 concentrations.
- Offline GC-MS (cartridges): In addition to the two on-site instruments, cartridges were sampled to enable specific compound identification, which required at least 2 L of sample air. Results from these analyses are provided in the Supplementary Materials.

The blank chamber was used as a control to (i) account for any non-soil/litter contributions to gas concentrations, (ii) ensure that the measured fluxes originated solely from biogenic sources inside the chamber, and (iii) exclude possible interferences from the chamber itself (background measurement). To ensure identical conditions between the sample and the blank chamber, the exact same gas volumes were collected.

We have revised part of our methodology to clarify this section more effectively. The revised text is suggested to be:

"After 20 minutes of chamber closure with continuous flow, a sampling bag was connected to the outlet of the Teflon pump, and a 5 L sample was collected over 10 minutes. By the end of the 30-minute process, a total of 15 L of air had flowed through the chamber, of which the last 5 L was used for subsequent analyses.

PTR-QMS: Measurements of biogenic volatile organic compounds (BVOCs) using proton-transfer-reaction quadrupole mass spectrometry (PTR-MS) necessitated a sufficiently large sample volume to capture BVOC concentrations with the required precision, especially considering their typically trace levels.

Los Gatos Analyzer: Due to its relatively high sample flow of approximately  $0.1 L \text{ min}^{-1}$ , a minimum of 0.5 L of gas was needed to ensure stable and accurate determination of methane  $(CH_4)$  and carbon dioxide  $(CO_2)$  concentrations.

Offline GC-MS (cartridges): For specific compound identification and qualitative analysis through thermal-desorption gas chromatography time-of-flight mass spectrometry (TD-GC-TOF-MS), at least 2 L of sample air was required to effectively load the adsorbent cartridges. (Results from these analyses are further detailed in the Supplementary Materials)."

We improved the 2.4 Sampling design section with details about blank measurements:

"Chambers were installed directly in the field with minimal disturbance to the surrounding environment. To account for potential background signals and chamber interferences, three blank

chambers—featuring completely bottom-sealed collars—were deployed per transect and measured simultaneously with the sample chambers (Fig. 2b)."

Third, another issue is the way nutrients were extracted, i.e. using nitro perchloric solution. The latter is HNO3+HCl. This suggests total nutrient contents were measured and not bio-available levels. There is also a poor discussion on why these nutrient levels are what they are (section 4.1).

We thank the Reviewer for raising the important point regarding nutrient extraction and the interpretation of nutrient levels.

The nitro-perchloric acid digestion method we employed measures total nutrient contents by releasing nutrients bound in both mineral and organic fractions, rather than the immediately bioavailable pools. This method is a well-established standard for comprehensive nutrient quantification in soils and plant tissues (Shaw, 1959; Malavolta et al., 1997; Marschner et al.,2012). While total nutrient content does not represent the labile pool available to plants and microbes at a given moment, it provides fundamental insight into the overall nutrient reservoirs and long-term soil fertility status. These total nutrient contents are critical for understanding ecosystem nutrient cycling and nutrient supply capacity, and they provide important information on the overall nutrient stocks within the system (Sarto et al., 2011; Fontes et al., 2021; Dietrich et al., 2018). Therefore, for our purpose of characterizing differences across forest types, the use of total nutrient content in soil and litter was considered appropriate.

Additionally, measuring total nutrients is particularly relevant for our study because it provides a foundation for understanding nutrient pools that contribute to long-term processes in forest soil and litter. Much of the focus in Section 4.1 is on how these nutrients influence BVOC and GHG fluxes, as well as microbial biomass. Since these fluxes are not solely influenced by immediately available nutrients but can also be affected by larger nutrient reservoirs, assessing total stocks allows us to capture a broader perspective on nutrient—flux relationships. For example, nutrient release from organic matter decomposition contributes to both bioavailable and total pools, thereby influencing overall ecosystem processes.

Regarding why nutrient levels vary across forest types, the observed differences reflect distinct biogeochemical processes related to soil parent material, organic matter input and decomposition, nutrient leaching, mineral weathering, and biological uptake and resorption in each forest ecosystem. For instance, the elevated potassium and phosphorus in the ancient river terrace forest likely indicate greater nutrient retention and cycling efficiency, whereas the iron dominance in upland soils aligns with the intense leaching characteristic of highly weathered tropical soils (Mosquera et al., 2024; Li et al., 2023). The white-sand forest's unexpectedly high nutrient concentrations may result from dissolved organic nutrients and root mat effects enhancing nutrient retention beyond classical fertility expectations (Lange et al., 2024; Draper et al., 2014).

We agree with the reviewer that not all aspects were sufficiently discussed. We have revised Section 4.1 to expand the discussion, following the feedback of the reviewer

**"4.1 Differences in soil and litter nutrient contents across forest types**

Soil and litter properties showed strong differences between forest types, particularly concerning their total nutrient contents. These total nutrient pools reflect the long-term nutrient availability and reservoirs within each ecosystem, shaped by distinct biogeochemical processes. The ancient river terrace forest stood out for its high litter K and P total contents. These elevated levels likely indicate robust nutrient retention and efficient cycling mechanisms within this forest type, potentially influenced by its historical flooding patterns and allisols (Andreae et al., 2015), which are generally younger and richer in nutrients.

In the upland forest, the dominance of soil iron total content is likely related to the intense leaching common in its ferrasols (oxisols), resulting in iron enrichment due to the removal of other nutrients (Mosquera et al., 2024). In addition, the formation of iron oxides can reduce the mineralization of organic matter, promoting iron accumulation in the leaf litter (Li et al., 2023). Overall, the white sand forest exhibited distinct soil properties compared to the other studied forest types. Despite its well-documented low fertility (Mendonça et al., 2015; Demarchi et al., 2022) and arenosol characteristics, this forest type showed unexpectedly high soil nutrient and carbon total concentrations (Fig. 4). Phosphorus total levels, for instance, were up to four times higher than in upland and ancient river terrace forests. This observation may be explained by the significant role of dissolved organic nutrients in mitigating nutrient limitations (Lange et al., 2024), or by the efficient capture and retention of nutrients within the forest's extensive root mats, which can enhance carbon storage and nutrient cycling in structurally analogous ecosystems (Draper et al., 2014). Iron total concentrations in the soil of the white sand forest were lower than expected (Cornu et al., 1997), possibly due to spatial variability and seasonal dynamics. During the dry season, low water retention in sandy soils induces drought stress, while wetseason leaching redistributes iron, aluminum, and magnesium (García-Villacorta et al., 2016). This process can form cemented horizons, impeding drainage and elevating water tables (Franco & Dezzeo, 1994; Demarchi et al., 2022). Additionally, differences in tree species composition between forest types may influence the total stocks of nutrient levels (García-Villacorta et al., 2016; Gomes Alves et al., 2022) further contributing to the observed variations."

Fourth, if living roots are also a potential source of BVOCS (line 508-511) why wasn't a root exclusion experiment added as an additional treatment? This would have made the data set, that has a very limited temporal resolution, more interesting.

We agree with the Reviewer that the inclusion of BVOC measurements from living roots would have made the data set more interesting. While we agree that such an experiment would provide valuable insights, implementing this methodology in the context of our study would have presented significant methodological and logistical challenges, as outlined below:

- 1. Methodological differences and complexity: Working with root BVOC emissions requires approaches that differ significantly from those used to measure fluxes from the litter-soil system, which was the focus of our study. A root exclusion experiment would require separating roots from the soil and litter layers, which involves substantial physical disturbance. This process could itself alter BVOC fluxes due to mechanical damage to roots, enhanced microbial activity, and the introduction of artificial hotspots for biogenic emissions (Arellano et al., 2013; Brilli et al., 2011). Additionally, isolating root contributions without affecting microbial and litter processes simultaneously is not straightforward, as roots interact dynamically with the surrounding soil and microbial communities (Peñuelas et al., 2014).
- 2. Logistical challenges: Our experimental design already involved long sampling and analyzing days to ensure the collection of BVOCs, CH?, and CO? fluxes across multiple forest types. Adding a root exclusion treatment would have required separate chambers, additional time-intensive preparation, and better lab infrastructure in the field. As our main goal was to detect the net soil/litter gas exchange with the atmosphere, having this extra experiment with isolating roots was beyond the scope of this study. Moreover, separating roots from litter and soil for such experiments in tropical forests with complex root systems is particularly challenging and requires prolonged field manipulations (Lang et al., 2021). These logistical demands would have further reduced the temporal and spatial coverage of our sampling efforts.

3. Root BVOCs as a distinct research topic: Research on root BVOCs is an important but highly specialized area within BVOC studies. Fluxes from roots are driven by distinct physiological pathways, including interaction with rhizosphere microbes, exudation processes, and stress-induced responses. Addressing root contributions to BVOCs would require a different experimental framework designed specifically for that purpose. While our study focused on the soil-litter compartment, future work could complement our findings by targeting root emissions using specific methods.

**Fifth, I think more efforts could have been done resolve this issue with m/z 42. Was this not solvable?**

Unfortunately, with the PTR-QMS, resolving this specific mass fragment is not achievable due to the inherent limitations of the quadrupole-based mass spectrometer. The PTR-QMS excels at real-time detection of VOCs but has reduced selectivity when it comes to distinguishing between molecules with the same m/z value. In the case of m/z 42, this peak is often associated with multiple isobaric compounds such as acetonitrile and fragments of larger VOCs, making it impossible to deconvolute these contributions without additional mass resolution. Since the PTR-QMS does not possess high-resolution capabilities, resolving the exact identity of m/z 42 is beyond its capacity. To resolve this issue, a PTR-Time of Flight (Tof) - MS would have been required, but unfortunately, at the time of this study, we did not have access to a PTR-ToF-MS in our laboratory setup. As the m/z 42 presented interesting results, we encourage future studies to further explore it and with better technical and analytical resources.

**Some specific comments:**

Line 71: I wonder that you can really generalise that CH4 fluxes are higher from sandy soils? Or do you refer to CH4 uptake fluxes (i.e. CH4 oxidation)?

We acknowledge that the original statement is too specific for the cited study and that CH4 dynamics are generally more complex than indicated.

While sandy soils typically exhibit higher diffusivity, which can enhance CH4 diffusion and oxidation and potentially lead to increased CH4 uptake, the reality is more nuanced. Besides soil texture, soil structure also plays a role. For example, microaggregates can create local anaerobic zones, causing anaerobic gas production, such as methanotrophic CH4 production. In the revised text, we chose to discuss gas dynamics with relation to soil structure and texture more broadly rather than focusing specifically on CH4.

For the revised manuscript, we suggest the following text:

Soil flux dynamics are influenced by multiple, interacting factors. Soil texture affects gas diffusivity—for example, higher porosity in sandy soils can enhance oxidation processes, such as found by Cai et al (1999)—while soil structure can create localized anaerobic conditions that promote anaerobic gas production (Sey et al 2008). Consequently, the effects of soil moisture on CH4 fluxes depend on both texture and structure, highlighting the complex interplay of physical soil properties in regulating soil flux dynamics.

Line 78: How can you state the Amazon is the largest source of BVOCS, without existing data from the Congo basin? Mention also the role of BVOCs for O3 and NOx dynamics.

We acknowledge that other tropical forests, such as those in the Congo Basin, also make important contributions to the global BVOC budget, which remains poorly investigated. However, our statement is based on previous studies showing that the Amazon Basin is widely regarded as the

largest source of BVOCs to the atmosphere, supported by a combination of satellite-derived estimates, field measurements, and atmospheric modeling (e.g., Wells et al., 2022). We have rephrased the statement in the manuscript to clarify this point as follows:

"The Amazon rainforest alone contributes about 40% of global BVOC emissions, playing a critical role in the global carbon cycle (Guenther et al., 2012; Wang et al., 2024; Tripathi et al., 2025)."

In addition, following the comments of the reviewer, we have rewritten the introduction with the role of BVOC, Ozone and NOx:

"Biogenic Volatile Organic Compounds (BVOCs) play critical roles across scales, from cellular processes to global climate regulation. While primarily emitted by plants, BVOCs can also be produced and consumed by soils, litter and microorganisms. Once released into the atmosphere, they actively participate in atmospheric chemistry and physics, influencing climate dynamics. BVOCs react with key atmospheric oxidants—including hydroxyl radicals (OH), ozone (O3), and nitrate radicals (NO3)—to form secondary organic aerosols (SOAs) (Artaxo et al., 2022; Yáñez-Serrano et al., 2020). SOAs, in turn, have a major influence on cloud properties, enhancing cloud condensation nuclei (CCN) concentrations, which impacts precipitation patterns and alters cloud lifecycles (Liu and Matsui, 2022). Depending on their chemical composition, SOAs can also influence the Earth's radiation budget by scattering incoming solar radiation (resulting in a cooling effect) or absorbing outgoing longwave radiation. Additionally, BVOCs contribute to the formation of tropospheric ozone—an important greenhouse gas and a major air pollutant (Vella et al., 2025). Given these large-scale impacts, accurately quantifying BVOC fluxes in terrestrial ecosystems is essential for advancing our understanding of forest—atmosphere interactions and for improving Earth system models—thereby improving climate predictions.

Global emissions of BVOCs from terrestrial vegetation are estimated at approximately 760 Tg C  $vr^{-1}$ , with isoprene (C5H8) and monoterpenes (C10H16) accounting for around 70% and 11% of these emissions, respectively (Tripathi et al., 2025). Isoprene is a simple building block compound emitted in large quantities, particularly by tropical forests, whereas monoterpenes such as  $\alpha$ -pinene,  $\beta$ -pinene, and limonene—are structurally more complex (Guenther et al., 2012; Gomes Alves et al., 2016). The Amazon rainforest alone contributes about 40% of global BVOC emissions, playing a critical role in the global carbon cycle (Guenther et al., 2012; Wang et al., 2024; Tripathi et al., 2025). However, these global estimates primarily consider emissions from plants, neglecting potential contributions from soil and litter, which might also include a large variety of BVOC chemical species. This gap is particularly significant given recent evidence that the soil-litter together is a compartment that can also play a crucial role in BVOC emissions (Fan et al., 2020, 2024; Bourtsoukidis et al., 2018; Peñuelas et al., 2014; Tang et al., 2019). Within this compartment, multiple biological and physical processes influence BVOC dynamics. These include plant-related processes such as intra- and inter-organism communication, herbivore defense, and symbiotic interactions (Gfeller et al., 2013; Lin et al., 2007; Rasheed et al., 2021; Steeghs et al., 2004; Tang et al., 2019; Trowbridge et al., 2020). Additionally, soil microorganisms produce and consume BVOCs for communication and ecological interactions (e.g., defense and competition), with these compounds also being released as residual metabolic products (Isidorov & Jdanova, 2002; Leff & Fierer, 2008; Liu et al., 2024; Monard et al., 2021).

Greenhouse gases (GHGs), such as methane (CH4), carbon dioxide (CO2) and nitrous oxide (N2O) are also produced and consumed by soil microorganisms through key metabolic processes, including methanogenesis, methanotrophy, and respiration (Conrad, 2009; Hofmann et al., 2016). Greenhouse gases (GHGs), such as methane (CH4), carbon dioxide (CO2) and nitrous oxide (N2O) are also produced and consumed by soil microorganisms through key metabolic processes, including methanogenesis, methanotrophy, and respiration (Conrad, 2009; Hofmann et al., 2016). While CO2, but also methane, are not classified as a BVOC, theyplay a crucial role in the overall gas exchange and are included in this study alongside BVOCs to

provide a broader perspective of soil-litter gas (carbon) fluxes. This inclusion is also important because environmental factors such as soil moisture, temperature, and nutrient availability influence both BVOC and GHG fluxes, albeit through distinct-but interconnected-biological and physical mechanisms (Greenberg et al., 2012; Tang et al., 2019; Asensio et al., 2007). These interconnected processes drive the net ecosystem exchange of gases between the soil-litter compartment and the atmosphere, making methane and CO2 key components for understanding processes driving BVOC flux dynamics.

These GHGs and BVOCs can also be linked to litter decomposition. In the litter decomposition process, physical factors, such soil moisture, temperature, and nutrient availability greatly affect microbial activity that drives these fluxes (Greenberg et al., 2012; Tang et al., 2019; Mäki et al., 2017; Asensio et al., 2007). N2O can be produced and consumed by soils through microbial nitrification and denitrification processes (Butterbach-Bahl et al., 2013; Snyder et al., 2009). These microbial processes, like those affecting other soil gases, are strongly influenced by environmental factors such as soil moisture, temperature, and nutrient availability (Saggar et al., 2013; Butterbach-Bahl et al., 2013). Together, these processes drive the net ecosystem exchange of BVOCs and GHGs between the soil-litter compartment and the atmosphere, and the magnitude and direction of this exchange may vary across different ecosystem types.

The Amazon Basin is a mosaic of diverse forest types (Oliveira-Filho et al., 2020), each with distinct plant species compositions (Ter Steege et al., 2013), shaped by the region's highly variable soil properties (Quesada et al., 2011; Quesada et al., 2012). Although Amazonian heterogeneity is known to play a critical role in regulating biogeochemical cycles, comparative studies across forest types—especially at the soil—litter interface—are still scarce. Distinct interactions between vegetation and soil can lead to highly variable patterns of BVOC and GHG exchange, making forest type-specific measurements essential for accurately representing the Amazon in atmospheric budgets. This lack of representation underscores the urgent need for studies that account for the region's ecological diversity to better capture the unique contributions of each forest type to biogeochemical processes. Quantifying this variability is key to improving both regional and global models, as gas fluxes are unlikely to be uniform even within the Amazon

To address these gaps, we investigated soil-litter BVOC (acetaldehyde, methanol, m/z 42, dimethyl sulfide, isoprene and monoterpenes) and GHG (CH4 and CO2) fluxes, soil and litter nutrient content and microbial biomass, and soil temperature and moisture from three forest types in central Amazonia: (i) ancient river terrace forest - a forest that was flooded in the past and is no longer flooded due to changes in the river course (paleoigapó); (ii) white sand forest (locally called campinarana) - a less common forest type that occupies about 5% of the Amazon basin (Adeney et al., 2016); and (iii) upland forest (locally called terra-firme) - the most common forest in Amazonia, with the highest plant species richness (Emidio et al., 2016; Luize et al., 2018). While methane and CO2 are not classified as BVOCs, they play a crucial role in the overall gas exchange and are included in this study alongside BVOCs to provide a broader perspective of soil-litter gas fluxes. This inclusion is important because environmental factors such as soil moisture, temperature, and nutrient availability influence both BVOC and GHG fluxes, albeit through distinct biological and physical mechanisms (Greenberg et al., 2012; Tang et al., 2019; Asensio et al., 2007). These interconnected processes drive the net ecosystem exchange of gases between the soil-litter compartment and the atmosphere, making methane and CO2 key components for understanding the full scope of these dynamics. With this approach, we aimed to answer the following questions: (i) what is the emission/consumption of BVOCs, CO2, and CH4 in magnitude and chemical diversity, and; (ii) what are the main drivers of soil-litter gas exchanges across these three forest types in central Amazonia (specifically, soil moisture and temperature, nutrient content, and microbial biomass from soil and litter)?"

**Line 225: no good fit for acetone, ethanol and formaldehyde,... what does this exactly mean?**

What we mean by "no good fit" for acetone, ethanol, and formaldehyde in the calibration process is that we encountered challenges inherent to their detection using PTR-QMS. The calibration curves for these compounds—generated as normalized counts per second (ncps) against their mixing ratios—did not yield consistent or accurate results.

These difficulties are linked to known challenges associated with the PTR-QMS when analyzing these specific compounds. For example, formaldehyde detection is particularly affected by the instrument's sensitivity to humidity. The proton affinity of formaldehyde is comparable to that of water vapor, leading to proton transfer reactions that can reverse or be suppressed under humid conditions, introducing inaccuracies. Because our calibration required humidified air to mimic ambient conditions, water vapor interactions likely interfered with formaldehyde detection, preventing the formation of reliable calibration curves. This limitation has been previously described, with formaldehyde detection by PTR-QMS showing poor linearity and sensitivity (Vlasenko et al., 2010; Warneke et al., 2011).

Acetone and ethanol exhibited issues likely related to proton-transfer reactions within the instrument's drift tube. Ethanol, for instance, has been shown to undergo competing reactions with H3O+ ions, leading to incomplete or inconsistent protonation, which can compromise measurement accuracy (Sémon et al., 2017). Because we could not achieve good fits in the calibration curves (i.e., high R2 values), we decided to exclude these compounds from the analysis, as they could not be reliably quantified.

**Line 308-309 (as example): no need to use 3 decimals here.**

Thanks for pointing that out. We removed the excessive number of decimals in the flux values for BVOCs and GHGs.

**Line 329-332: You cannot make a "translation" to "soil moisture available to plants" without soil physical data such as a Pf curve.**

We acknowledge that a precise determination of soil moisture available to plants requires specific soil physical data, such as water retention curves (Pf curves) or field capacity measurements, which were not part of our dataset.

To address this, we revised the text in the result section. We ensured we did not overinterpret the soil moisture data as directly representative of plant-available water. Instead, we highlighted that soil texture significantly modulates how measured soil moisture translates into water accessibility for plants and microbes.

**Below follows the sentence that we rewrote:**

"The substantial differences in soil texture (see supplementary material, Table S1) between sites modulate how measured soil moisture corresponds into water available to plants and microbes. However, since individual transects were measured on different consecutive days, distinguishing temporal from spatial effects remains challenging."

**Line 462: higher than anticipated, how much higher?**

In this study, we found much higher BVOC emission fluxes compared to those reported by previous studies in tropical soils. Several studies investigating soil BVOC emissions in tropical forests have suggested relatively low emissions under undisturbed conditions. For example: Bourtsoukidis et al. (2018) measured sesquiterpene emissions in Amazonian soils and reported fluxes of sesquiterpenes around 0.01-0.02 mg m-2 h-1, which are significantly lower than, for instance, the acetaldehyde fluxes (29.911 mg m-2 h-1) or dimethyl sulfide (0.924 mg m-2 h-1) emissions observed in our study for the white sand forest. Llusià et al. (2022) observed monoterpene emissions of 0.0104 mg m-2 h-1 in unfertilized plots and 0.0390 mg m-2 h-1 in fertilized plots. In comparison, monoterpenes emissions in our study were higher (1.164 mg m-2 h-1), especially in the white sand forest.

**Line 516: I think this is rather a diffusion into the soil, then an effective (microbial) or (physical) uptake.**

We agree with the Reviewer that diffusion of plant-emitted isoprene into the soil via deposition is a plausible explanation that complements our suggestion of microbial activity. To address this, we will add to the discussion the possibility of isoprene diffusion and subsequent interactions in the soil, including both microbial and physical uptake, as these processes may occur simultaneously.

"The lower isoprene uptake by the soil observed in the upland forest (-0.1 mg m-2 h-1) compared to the higher uptake fluxes in drier conditions ( $\sim$  -2.38 mg m-2 h-1) reported by Pugliese et al. (2023) may reflect the combined effects of diffusion of isoprene emitted by plants into the soil and subsequent microbial and physical absorption processes. Diffusion acts as a transport mechanism, allowing isoprene to enter the soil matrix, where microbial communities metabolize it as a carbon source (Cleveland and Yavitt, 1998). In addition, physical processes such as adsorption or dissolution can influence isoprene loss, especially under variable soil moisture conditions (Mu et al., 2023). Thus, diffusion and microbial/physical uptake likely occur simultaneously, controlling the net isoprene fluxes observed in the soil."

**Line 695-696: how can a forest type with an aerial coverage of 5% offset all carbon losses from other forest types?**

We understand the Reviewer's concern regarding the relatively small area coverage of white-sand forests and their potential role in compensating carbon losses from other forest types. We would like to clarify that our intention was not to imply that white-sand forests serve as a carbon offset mechanism for the entire Amazon Basin. Rather, we aimed to highlight their unique ecological roles and the biogeochemical processes they influence. This illustrates that even in relatively small areas, such as those occupied by white-sand forests, the biogeochemical and atmospheric processes affected by BVOCs and CH4 may differ substantially, reinforcing the heterogeneity of the Amazon's diverse forest types.

**The revised sentence:**

"In these areas, higher productivity under dry conditions may maintain relatively stable carbon dynamics, presenting a contrasting response to the substantial carbon losses typically observed in deep water table - upland forests - during drought."

Examples of "speculation": lines 482, 487, 507, 530, 550-559 (the entire soil moisture and temperature issue as drivers for CH4 and BVOC fluxes, cannot be covered with this limited temporal data set!), 575-577 (idem), 582-584 (idem), line 675 (due to the low temporal coverage the extreme conditions of especially the white sand soils could not be covered),...

Examples of "overstating": lines 36, 468, 568-569, 692,...

We acknowledge your concern that our limited temporal dataset for soil moisture and temperature might lead to speculation or overstatement regarding these factors as drivers for CH4 and BVOC fluxes. While our measurements represent a snapshot in situ investigation rather than continuous long-term monitoring, the statistical analyses performed allowed us to identify significant correlations and patterns within the context of our study. We view these findings as valuable indications and hypotheses that warrant further, more extensive research, particularly given the scarcity of such data from Amazonian ecosystems like the white sand forest. As noted in our conclusion (Lines 714-715), this research represents a pioneering investigation aimed at establishing initial insights into these ecosystems.

To address your specific points regarding "speculation" and "overstating," we have carefully revised the identified lines to employ more cautious and precise language.

Here is a summary of the revisions:

Line 36 - "...WS can be a significant ecosystem for BVOC and methane fluxes, where these fluxes are influenced by soil moisture and temperature

Line 468 - Revision: "...key observations related to BVOC and CH4 fluxes and their drivers..."

Line 482 Revision: "...possibly reflecting the role of dissolved organic nutrients in mitigating nutrient limitations (Lange et al., 2024)."

Line 487 - Revision: "...possibly attributed to spatial variability and seasonal dynamics."

Line 507 - Revision: "...the observed high emissions might indicate contributions from the activity of microorganisms living in the soil and litter (Carruthers & Lee, 2021; Hernandez-Arranz et al., 2019)."

Lines 550-559- Revision: "A principal component analysis (PCA) was performed to identify variables that could collectively differentiate forest types and their gas fluxes. As the PCA showed a limited capacity for differentiation due to overlapping ellipses, a further investigation was carried out using linear models (LMs). The LMs suggested that soil temperature and soil moisture were influential physical factors for all three forest types, particularly for the white sand forest. The white sand forest, with its relatively open canopy, shorter trees, and shallow water table (Adeney et al., 2016; Rossetti et al., 2019), often experiences dynamic conditions, which was reflected in the highly variable temperature and soil moisture values measured across its two transects over two subsequent days. In the following sections, we discuss the observed roles of soil temperature and moisture, and other potential factors influencing gas fluxes."

Regarding lines 575-577, we suggest underlining that we are limited by a small temporal dataset as follows:

"However, since high soil moisture often coincides with low temperatures, and since our conclusions are based on a limited temporal dataset, it remains challenging to ascertain whether low temperatures or high moisture levels drive increased fluxes under field conditions."

Line 568-569 - Revision:

"However, in the white sand forest, high BVOC emissions were observed in the wetter transect, while low emissions and uptake were observed in the drier transect. While based on limited data, it indicates that Amazonian soil emissions may respond differently to soil moisture depending on the soil and forest type."

Line 675: The original sentence states that few studies exist partly due to

"challenging conditions... such as flooding and extreme temperatures."

This line describes the inherent nature of white sand forests, not an assertion that our study fully captured these extreme conditions. While our interpretation throughout the discussion focuses on the potential impact of such conditions, acknowledging our own temporal limitations, we fully agree with the reviewer's valid point that it is indeed important to explicitly state that our study, with its current temporal coverage, could not comprehensively observe the full range of these extremes.

To address this, we have revised line 675 by adding a clarifying sentence that explicitly states this limitation. The updated text now reads:

"This can partly be explained by the challenging conditions of this forest type, such as flooding and extreme temperatures, which require specific infrastructure for data collection. While our observations constitute one of the first characterizations of VOCs in this unique forest type, their capacity to capture the full spectrum of extreme conditions is inevitably limited by the short temporal coverage of the dataset..

Line 692 - Revision: "Our results suggested a stronger link between white sand forest gas fluxes and physical factors (more than other forest types), which indicates a possible sensitivity to upcoming climate extremes."

**References**

Alves, E. G., Jardine, K., Tota, J., Jardine, A., Yãnez-Serrano, A. M., Karl, T., Tavares, J., Nelson, B., Gu, D., Stavrakou, T., Martin, S., Artaxo, P., Manzi, A., and Guenther, A.: Seasonality of isoprenoid emissions from a primary rainforest in central Amazonia, Atmos. Chem. Phys., 16, 3903–3925, <a href="https://doi.org/10.5194/acp-16-3903-2016">https://doi.org/10.5194/acp-16-3903-2016</a>, 2016.

Asensio, D., Peñuelas, J., Filella, I. et al. On-line screening of soil VOCs exchange responses to moisture, temperature and root presence. Plant Soil 291, 249–261 <a href="https://doi.org/10.1007/s11104-006-9190-4">https://doi.org/10.1007/s11104-006-9190-4</a>, 2007.

Artaxo, P., Hansson, H.-C., Andreae, M. O., Bäck, J., Alves, E. G., Barbosa, H. M. J., Bender, F., Bourtsoukidis, E., Carbone, S., Chi, J., Decesari, S., Despres, V. R., Ditas, F., Ezhova, E., Fuzzi, S., Hasselquist, N. J., Heintzenberg, J., Holanda, B. A., Guenther, A., ... Kesselmeier, J.. Tropical and Boreal Forest Atmosphere Interactions: A Review. Tellus. Series B: Chemical and Physical Meteorology, 74, 24-163. <a href="https://doi.org/10.16993/tellusb.34">https://doi.org/10.16993/tellusb.34</a>, 2022.

Cleveland, C.C., and Yavitt, J.B. Microbial consumption of atmospheric isoprene in a temperate forest soil. Soil Biology and Biochemistry, 30(3), 345-355. <a href="https://doi.org/10.1016/S0038-0717(97)00180-1,1998">https://doi.org/10.1016/S0038-0717(97)00180-1,1998</a>.

Cornu, S., Ambrosi, J. P., Lucas, Y., and Fevrier, D.: A comparative study of the soil solution chemistry of two Amazonian forest soils (Central Amazonia, Brazil). In Hydrology and Earth System Sciences (Vol. 1, Issue 2, pp. 313–324). https://doi.org/10.5194/hess-1-313-1997, 1997.

Dietrich, L., et al. Comparing spatial heterogeneity of bioavailable nutrients in tropical forests. Biogeosciences, 15, 123-140. <a href="https://doi.org/10.5194/bg-15-123-2018">https://doi.org/10.5194/bg-15-123-2018</a>, 2018.

Draper, F.C., et al. Root mat influence on soil carbon storage in white sand ecosystems. Biogeosciences, 11, 3891-3906. https://doi.org/10.5194/bg-11-3891-2014, 2014.

Fan, J., Luo, R., McConkey, B. G., and Ziadi, N.: Effects of nitrogen deposition and litter layer management on soil CO2, N2O, and CH4 emissions in a subtropical pine forestland, Sci. Rep., 10, 1–11, https://doi.org/10.1038/s41598-020-65952-8, 2020

Fan, Y., Zhang, Y., Osborne, B., and Zou, J.: Global patterns of soil greenhouse gas fluxes in response to litter manipulation, Cell Rep. Sustain., 1, 100003, https://doi.org/10.1016/j.crsus.2023.100003, 2024

Fontes, M.P.F., et al. Comprehensive assessment of extraction methods for plant micronutrients including bioavailability implications. Biogeosciences, 18, 2543-2558. <a href="https://doi.org/10.5194/bg-18-2543-2021">https://doi.org/10.5194/bg-18-2543-2021</a>, 2021.

Guenther, A., Jiang, X., Palmer, P. I., Geron, C., Seco, R., Sarkar, C., Matsunaga, S. N., Harley, P., and Wiedinmyer, C.: The model of emissions of gases and aerosols from nature version 2.1 (MEGAN2.1): An extended and updated framework for modeling biogenic emissions, *Geosci. Model Dev.*, 5, 1471–1492, https://doi.org/10.5194/gmd-5-1471-2012, 2012.

Greenberg, J., Asensio, D., Turnipseed, A., Guenther, A., Karl, T., & Gochis, D. Contribution of leaf and needle litter to whole ecosystem BVOC fluxes. *Atmospheric Environment*, *59*, 302-311. <a href="https://doi.org/10.1016/j.atmosenv.2012.04.038">https://doi.org/10.1016/j.atmosenv.2012.04.038</a>, 2012.

Lange, C., et al. Influence of dissolved organic nutrients on P availability in white sand soils. Biogeosciences, 21, 1587-1601. https://doi.org/10.5194/bg-21-1587-2024, 2024.

Li, W., et al. Iron oxides control on organic matter mineralization in tropical forest soils. Biogeosciences, 20, 765-780. <a href="https://doi.org/10.5194/bg-20-765-2023">https://doi.org/10.5194/bg-20-765-2023</a>, 2023.

Malavolta, E., Vitti, G.C. and Oliveira, S.A. Avaliação do estadonutricional das plantas: Princípios e aplicaçães. [Evaluation of the Nutritional Status of Plants: Principles and Applications.] Piracicaba, Potafos, 1997.

Marques, H.A., et al. Soil carbon content variability in Amazonian upland forests. Biogeosciences, 14, 3297-3310. https://doi.org/10.5194/bg-14-3297-2017, 2017.

Mosquera, F., et al. Iron enrichment effects on nutrient cycling in oxisol soils of upland tropical forests. Biogeosciences, 21, 2203-2220. <a href="https://doi.org/10.5194/bg-21-2203-2024">https://doi.org/10.5194/bg-21-2203-2024</a>, 2024.

Mu, Z., et al. Soil uptake of isoprenoids in a Eucalyptus urophylla plantation. Forests and Global Change, 6, 123-134. https://doi.org/10.3389/ffgc.2023.1260327, 2023.

Oliveira-Filho, A. T., Dexter, K. G., Pennington, R. T., Simon, M. F., Bueno, M. L., & Neves, D. M. On the floristic identity of Amazonian vegetation types. *Biotropica*, *53*(3), 767-777. <a href="https://doi.org/10.1111/btp.12932">https://doi.org/10.1111/btp.12932</a>, 2021.

Ran, Z., et al. Standard methods for soil nutrient extraction and assessment: Environmental science applications. J. Environ. Sci., 47, 1-14. <a href="https://doi.org/10.1016/j.jes.2016.02.012">https://doi.org/10.1016/j.jes.2016.02.012</a>, 2016.

Sarto, M.V.M., et al. Comparison of soil micronutrient extractants: total versus available fractions. Biogeosciences, 8, 1217-1228. https://doi.org/10.5194/bg-8-1217-2011, 2011.

Shaw, G. Sulfur determination by nitric-perchloric acid digestion methods. Anal. Chem., 31, 1118-1122. https://doi.org/10.1021/ac60152a038, 1959.

Sparks, D.L. Environmental Soil Chemistry. 2nd ed., Academic Press, San Diego, https://doi.org/10.1016/B978-0-12-656446-4.X5000-2, 2003.

Tang, J., Schurgers, G., and Rinnan, R.: Process understanding of soil BVOC fluxes in natural ecosystems: a review, Rev. Geophys., 57, 966–986, <a href="https://doi.org/10.1029/2018RG000634">https://doi.org/10.1029/2018RG000634</a>, 2019.

Tripathi, N., Krumm, B. E., Edtbauer, A., Spracklen, D. V., and Kanzow, T.: Impacts of convection, chemistry, and forest clearing on biogenic volatile organic compounds over the Amazon, *Nat. Commun.*, 16, 4692, <a href="https://doi.org/10.1038/s41467-025-59953-2">https://doi.org/10.1038/s41467-025-59953-2</a>, 2025.

Vella, R., Forrest, M., Pozzer, A., Tsimpidi, A. P., Hickler, T., Lelieveld, J., and Tost, H.: Influence of land cover change on atmospheric organic gases, aerosols, and radiative effects, Atmos. Chem. Phys., 25, 243–262, https://doi.org/10.5194/acp-25-243-2025, 2025.

Yáñez-Serrano AM, Bourtsoukidis E, Alves EG, Bauwens M, Stavrakou T, Llusià J, Filella I, Guenther A, Williams J, Artaxo P, Sindelarova K, Doubalova J, Kesselmeier J, Peñuelas J. Amazonian biogenic volatile organic compounds under global change. Glob Chang Biol. Sep;26(9):4722-4751. doi: 10.1111/gcb.15185, 2020.

---

## Author Comment (AC3)

**Responses Answer to reviewer 2**

\_\_\_\_\_\_

**Reviewer 2'comments in blue**

Responses: Answers from authors in black

Suggested new text blocks in italic

\_\_\_\_\_\_

**Reviewer 2**

The manuscript "Forest Diversity and Environmental Factors Shape Contrasting Soil-Litter BVOC and Methane Fluxes in Three Central Amazonian Ecosystems" presents an extensive data set of surface gas (BVOC, CH4, and CO2) measurements across three forest types in the Amazon rainforest alongside a wide range of environmental, chemical, and microbial measurements. The data are valuable, rare, and were certainly collected with a tremendous amount of effort. The manuscript needs additional methodological clarification and organization/streamlining to clarify and highlight the findings. More information about the amount and type of litter contributing to the fluxes is crucial. This work will provide important new information to the scientific community.

We truly appreciate the reviewer's comments and the time dedicated to making important suggestions and comments that have greatly contributed to improving our manuscript.

**Major comments**

**Nature of the soil and litter fluxes:**

It's unclear in the abstract (e.g., L27) and introduction whether this paper will cover flux of BVOCs between soil and litter (in which case we need to know the direction so we know what uptake or emission means in later parts of the abstract), or their respective fluxes with the atmosphere. From the methods, it seems like these measurements are 'forest floor' and include both soil and litter inside the collar and thus represent the net soil/litter-atmosphere exchange. This should be clarified early on.

Yes, this study focuses on the flux from the forest floor - both soil and litter within the chamber. These measurements indeed represent the net soil/litter-atmosphere exchange. We have rewritten this to make the distinction clearer so that the terminology consistently reflects the inclusion of both soil and litter, as well as the direction of the fluxes. We modified the text as follows:

**Abstract**

"In this study, we investigated the net soil/litter-atmosphere exchange of BVOCs and methane, along with their potential drivers".

**Introduction**

"This gap is particularly significant given recent evidence that the soil—litter together is a compartment that can also play a crucial role in BVOC emissions (Fan et al., 2020, 2024; Bourtsoukidis et al., 2018; Peñuelas et al., 2014; Tang et al., 2019)."

"Together, these processes drive the net ecosystem exchange of BVOCs and GHGs between the soil-litter compartment and the atmosphere, and the magnitude and direction of this exchange may vary across different ecosystem types."

**Soil vs litter contributions:**

I do not find mention of the amount (mass or surface area) or plant species of the litter that was mentioned in the collars. Given the potential for litter emissions of VOCs (and maybe even uptake) to overwhelm soil fluxes, it is really important to indicate how much litter there was and how much it varied across measurements. I would suggest adding this as a table early on in your results. If these data are not available, this should be clearly stated, and some indication of how consistent this is across and within transects should be presented.

We agree that this information is very important, and we realize that the observed fluxes (emission or uptake) can be attributed to soil as well as litter fluxes. Unfortunately, at the time of this study, we did not measure the total weight of the litter inside the chamber collars. While some information is available for upland forests (1.17 t DW ha-1, Luizão et al. 2004), no information for the other studied forest ecosystems is known in our region. We therefore agree with the reviewer that we need to clarify better that we are measuring soil/litter-atmosphere exchange of BVOCs and methane across forest types. We therefore suggest the following changes.

For the revised manuscript, we suggest checking each wording carefully, making sure that we use soil/litter-atmosphere fluxes to clarify that we are measuring both. In addition, we suggest adding this to the manuscript at the start of the Discussion in 4.2:

**"4.2 Differences in gas fluxes across the different forest types:**

When comparing ecosystem fluxes, it is important to recognize that chamber measurements represent the **combined (net) flux** from both soil and litter. Observed differences between sites may therefore reflect variations in the relative contributions of soil and litter, for example, due to different amounts of litter. Because our measurements do not allow us to separate these sources, we cannot determine to what extent the observed differences are attributable to each component. Consequently, we treat soil and litter together as a single compartment in our analysis."

**Organization and streamlining:**

The introduction is missing a motivation for the study. Rather than starting with BVOC sources and known drivers, it would be more compelling to motivate why study this at all. I suggest adding a big-picture section first which could include your L78-84. Then talk about specific sources/processes. Make sure motivation for studying different soil/forest types is clear, and goes beyond just that the Amazon has different soil/forest types. Please work on the flow of the motivating sections.

We thank the reviewer for the suggestion of adding to the introduction a stronger motivation for the study. We restructured the introduction to start with a broader context that highlights the comprehensive importance of BVOC fluxes and their implications for the climate system, beyond the Amazon.

Below is a revised version of the Introduction. The main changes are indicated in **bold**.

Biogenic Volatile Organic Compounds (BVOCs) play critical roles across scales, from cellular processes to global climate regulation. While primarily emitted by plants, BVOCs can also be produced and consumed by soils, litter and microorganisms. Once released into the atmosphere, they actively participate in atmospheric chemistry and physics, influencing climate dynamics. BVOCs react with key atmospheric oxidants—including hydroxyl radicals (OH), ozone (O₃), and nitrate radicals (NO3)—to form secondary organic aerosols (SOAs) (Artaxo et al., 2022; Yáñez-Serrano et al., 2020). SOAs, in turn, have a major influence on cloud properties, enhancing cloud condensation nuclei (CCN) concentrations, which impacts precipitation patterns and alters cloud lifecycles (Liu and Matsui, 2022). Depending on their chemical composition, SOAs can also influence the Earth's radiation budget by scattering incoming solar radiation (resulting in a cooling effect) or absorbing outgoing longwave radiation. Additionally, BVOCs contribute to the formation of tropospheric ozone—an important greenhouse gas and a major air pollutant (Vella et al., 2025). Given these large-scale impacts, accurately quantifying BVOC fluxes in terrestrial ecosystems is essential for advancing our understanding of forest-atmosphere interactions and for improving Earth system models—thereby improving climate predictions.

Global emissions of BVOCs from terrestrial vegetation are estimated at approximately 760 Tg C yr-1, with isoprene ( $C_5H_8$ ) and monoterpenes ( $C_{10}H_{16}$ ) accounting for around 70% and 11% of these emissions, respectively (Tripathi et al., 2025). Isoprene is a simple building block compound emitted in large quantities, particularly by tropical forests, whereas monoterpenes—such as αpinene, β-pinene, and limonene—are structurally more complex (Guenther et al., 2012; Gomes Alves et al., 2016). The Amazon rainforest alone contributes about 40% of global BVOC emissions, playing a critical role in the global carbon cycle (Guenther et al., 2012; Wang et al., 2024; Tripathi et al., 2025). However, these alobal estimates primarily consider emissions from plants, neglecting potential contributions from soil and litter, which might also include a large variety of BVOC chemical species. This gap is particularly significant given recent evidence that the soil-litter together is a compartment that can also play a crucial role in BVOC emissions (Fan et al., 2020, 2024; Bourtsoukidis et al., 2018; Peñuelas et al., 2014; Tang et al., 2019). Within this compartment, multiple biological and physical processes influence BVOC dynamics. These include plant-related processes such as intra- and inter-organism communication, herbivore defense, and symbiotic interactions (Gfeller et al., 2013; Lin et al., 2007; Rasheed et al., 2021; Steeghs et al., 2004; Tang et al., 2019; Trowbridge et al., 2020). Additionally, soil microorganisms produce and consume BVOCs for communication and ecological interactions (e.g., defense and competition), with these compounds also being released as residual metabolic products (Isidorov & Jdanova, 2002: Leff & Fierer, 2008; Liu et al., 2024; Monard et al., 2021.

Greenhouse gases (GHGs), such as methane (CH4), carbon dioxide (CO2) and nitrous oxide (N2O) are also produced and consumed by soil microorganisms through key metabolic processes, including methanogenesis, methanotrophy,

and respiration (Conrad, 2009; Hofmann et al., 2016). While CO2, but also methane, are not classified as a BVOC, they play a crucial role in the overall gas exchange and are included in this study alongside BVOCs to provide a broader perspective of soil-litter gas (carbon) fluxes. This inclusion is also important because environmental factors such as soil moisture, temperature, and nutrient availability influence both BVOC and GHG fluxes, albeit through distinct-but interconnected-biological and physical mechanisms (Greenberg et al., 2012; Tang et al., 2019; Asensio et al., 2007). These interconnected processes drive the net ecosystem exchange of gases between the soil-litter compartment and the atmosphere, making methane and  $CO_2$  key components for understanding processes driving BVOC flux dynamics.

These GHGs and BVOCs can also be linked to litter decomposition. In the litter decomposition process, physical factors, such soil moisture, temperature, and nutrient availability greatly affect microbial activity that drives these fluxes (Greenberg et al., 2012; Tang et al., 2019; Mäki et al., 2017; Asensio et al., 2007).  $N_2O$  can be produced and consumed by soils through microbial nitrification and denitrification processes (Butterbach-Bahl et al., 2013; Snyder et al., 2009). These microbial processes, like those affecting other soil gases, are strongly influenced by environmental factors such as soil moisture, temperature, and nutrient availability (Saggar et al., 2013; Butterbach-Bahl et al., 2013). Together, these processes drive the net ecosystem exchange of BVOCs and GHGs between the soil-litter compartment and the atmosphere, and the magnitude and direction of this exchange may vary across different ecosystem types.

The Amazon Basin is a mosaic of diverse forest types (Oliveira-Filho et al., 2020), each with distinct plant species compositions (Ter Steege et al., 2013), shaped by the region's highly variable soil properties (Quesada et al., 2011; Quesada et al., 2012). Although Amazonian heterogeneity is known to play a critical role in regulating biogeochemical cycles, comparative studies across forest types—especially at the soil—litter interface—are still scarce. Distinct interactions between vegetation and soil can lead to highly variable patterns of BVOC and GHG exchange, making forest type-specific measurements essential for accurately representing the Amazon in atmospheric budgets. This lack of representation underscores the urgent need for studies that account for the region's ecological diversity to better capture the unique contributions of each forest type to biogeochemical processes. Quantifying this variability is key to improving both regional and global models, as gas fluxes are unlikely to be uniform even within the Amazon

To address these gaps, we investigated soil-litter BVOC (acetaldehyde, methanol, m/z 42, dimethyl sulfide, isoprene and monoterpenes) and GHG (CH4 and CO2) fluxes, soil and litter nutrient content and microbial biomass, and soil temperature and moisture from three forest types in central Amazonia: (i) ancient river terrace forest - a forest that was flooded in the past and is no longer flooded due to changes in the river course (paleoigapó); (ii) white sand forest (locally called campinarana) - a less common forest type that occupies about 5% of the Amazon basin (Adeney et al., 2016); and (iii) upland forest (locally called terra-firme) - the most common forest in Amazonia, with the highest plant species richness (Emidio et al., 2016; Luize et al., 2018). We aimed to answer the following questions: (i) what is the emission/consumption of BVOCs, CO2, and CH4 in magnitude and chemical diversity, and; (ii) what are the main drivers of soil-litter gas exchanges across these three forest types in central Amazonia (specifically, soil moisture and temperature, nutrient content, and microbial biomass from soil and litter)?"

The results section could be more intentional about highlighting data that are most important in explaining the observed patterns. Other data could be moved to the supplement. For example, it would help to have the authors summarize and integrate the main findings in terms of potential drivers of gas fluxes. Listing the results in Tables 4-6 for each site is a bit overwhelming for the reader. These could be moved to the supplement or somehow combined and summarized. This is alluded to in L467-470, but I think the authors could also be more selective with the variables presented in figures and tables in the main text if they are not central to the findings and instead have them in the supplement. The role of the transects is not clear until section 3.3 where they are shown in terms of spatial variability. It would make more sense to me to present these results early on, alongside the site averages. I assume that this variability is considered in the comparison with the environmental, chemical, microbial data anyway. This would help reveal that there is a strong transect effect at WS likely driven by soil moisture earlier rather than later in the results.

Regarding the tables (Tables 4–6), our initial goal was to include the full dataset in the main text to provide comprehensive information, as we believe these data could be valuable for other researchers in the field. However, we understand that presenting such detailed tables in the main text may overwhelm the reader. We agree with the Reviewer's suggestion to combine and summarize the data, consolidating all forest types into a single, more concise table. We will also try to move some of the detailed results back to the supplementary material to streamline the results section and focus on the most critical findings in the main text.

On the role of transects, our initial aim was to first provide an overview of the results, highlighting the differences between forest types, and discuss the drivers that could explain those differences. After that, we delve deeper into spatial and temporal variability. However, we agree with the reviewer that the temporal variability, and in this case the differences between transects, was crucial in finding the effects of external factors, such as the rain event and the soil moisture. We will try to present the aspects of the transects clearly and earlier in the Methodology and Discussion.

Regarding the role of the transects, we suggest adding the following to the Methodology:

For each forest type, a PELD-MAUA plot (~1 hectare) (<a href="https://peld-maua.inpa.gov.br">https://peld-maua.inpa.gov.br</a>) was selected, within which two 150 m transects were established in homogeneous areas characterized by consistent vegetation structure, soil characteristics, and topography to minimize spatial variability and avoid pseudoreplication. Along each transect, six sampling points were marked at ~30 m intervals, resulting in a total of 36 soil chamber measurements conducted on consecutive days; although this design was necessary for logistical reasons, it also allowed us to examine the influence of external factors beyond forest-type differences. Chamber-based methods (Section 2.3; Fig. 1) were used to quantify in situ fluxes of CO2, CH4, and BVOCs from the soil–litter compartment, and three blank chambers with sealed collars were deployed per transect to account for background signals and potential chamber interferences (Fig. 2b).

**4.3.1 Soil moisture and soil temperature as drivers of soil and litter gas fluxes**

"While efforts were made in this study to minimize the effects of spatial and temporal variability, it is important to consider that external factors inevitably influenced our results. For example, while transects were measured at the same time (08:00-10:00 am, local time), they were measured on different days under different weather conditions. This allowed us to assess the effect of external factors (e.g., soil moisture and temperature) on gas fluxes within the same forest type. Soil temperature and moisture were found to be significant drivers for most gases, which agrees with what has been observed in other ecosystems (Trowbridge et al., 2020; Pugliese et al., 2023; Liu et al., 2024). For example, Pugliese et al. (2023) observed that rainforest soils acted as net BVOC sinks under moist conditions and as net BVOC sources under dry conditions. In the upland forest, we observed a similar pattern as Pugliese et al (2023), with the wetter transect showing BVOC consumption while the drier transect showed emissions. The white sand forest showed even stronger inter-transect differences, with high BVOC emissions observed in the wetter transect, and low emissions and uptake observed in the drier transect. We expect that the heavy rainfall, which occurred just before the measurement of transect 2, had a strong impact on the BVOC emissions. Bourtsoukidis et al. (2018) also found that sesquiterpenes emissions from upland forest soils in the dry season (after a rain event) were comparable to those from vegetation, suggesting that soil moisture is a crucial factor influencing sesquiterpenes emissions from Amazonian soils. As we observed substantially high isoprene, monoterpenes, and acetaldehyde emissions in transect 2 of the white sand forest, we argue that these observed BVOC emissions represent a burst induced by the preceding rainfall event, similar to the observed increase in BVOC emissions during and immediately after rainfall in a Ponderosa pine plantation (Greenberg et al., 2012). Likewise, Jardine et al. (2016) observed a peak in DMS soil emissions after rainfall. Therefore, higher emissions are expected to result from the interlinked effects of soil temperature and moisture, and as described above, the possible physical effects of rainfall (Miyama et al., 2020). In addition to temporal variability, a strong spatial variability needs to be considered within and between transects. For example, the complex terrain in each ecosystem can affect local hydrology and water drainage patterns on a small scale. This highlights the inherent complexity and heterogeneity of soil-litter gas exchange with the atmosphere, especially when both spatial and temporal variability are considered."

The discussion should be organized around major topics (xxxxxnot order of results presented or analyses) and streamlined significantly. For example, rather than summarizing results by analysis in the discussion (e.g., separate section on PCA in L550 where you are repeating methods and results) I would summarize by major findings and discuss across your results. Try to remove repetition in the discussion with respect to your moisture and temperature results. Also, you relate isoprene emissions to microbes in two sections, and you could organize to discuss that in one location only.

We appreciate the reviewers' comments and also recognize that we are repeating certain discussion points at different locations in the manuscript. Regarding the moisture and temperature results, we have eliminated the redundancy and streamlined the discussion on these topics, as you can see in the newly suggested text of 4.3.1, which was posted here before as part of the review.

We also agree that we need to remove the PCA and LM as separate sections, and we suggest discussing them now together in **4.3** 'Drivers of Soil and Litter Gas Fluxes'.

Furthermore, we suggest centralizing the discussion around isoprene and microorganisms, and have merged all references to isoprene emissions and their potential links to microorganisms into a single cohesive discussion within 4.3.2 **Forest typespecific drivers of soil and litter gas fluxes**. Below we show the newly suggested text for 4.3.2:

**4.3.2 Forest type-specific drivers of soil and litter gas fluxes**

We observed that drivers of soil and litter gas fluxes varied across forest types, reflecting their unique environmental conditions and nutrient dynamics. Here, we focus on the key factors influencing gas fluxes—soil nutrients, microbial biomass, and their interactions with environmental conditions.

**Potassium, carbon, and phosphorus** emerged as significant drivers of gas fluxes, often varying in their influence across forest types. For the ancient river terrace and upland forests, potassium was identified as a significant predictor for fluxes of methanol and monoterpenes, and for m/z 42 fluxes in the upland forest. Although potassium's role in BVOC and GHG fluxes remains underexplored, its availability is crucial for plant growth and metabolism (Wang et al., 2013), which could indirectly influence BVOC production through changes in plant and microbial physiology (Mazahar and Umar, 2022). In the upland forest, methane consumption fluxes were well-correlated with soil carbon, in conjunction with soil moisture. This is consistent with the role of soil organic carbon in supporting methanotrophic bacteria responsible for methane oxidation (Lee et al., 2023).

**Phosphorus**, another essential nutrient, significantly influenced BVOC fluxes, particularly for methanol in the white sand forest. This forest type often experiences extreme nutrient limitation and environmental stress. Phosphorus availability can alter BVOC production both directly in plants (Ndah et al., 2022) and indirectly by modifying soil pH and microbial health (Stotzky et al., 1976; Liu et al., 2024). Interestingly, higher isoprene emissions were associated with lower phosphorus levels, suggesting a potential phosphorus limitation in the white sand forest. This observation aligns with studies linking nutrient balance (e.g., high nitrogen relative to phosphorus) to enhanced emissions of BVOCs like monoterpenes and sesquiterpenes (Llusià et al., 2022). While the exact mechanisms for isoprene remain unclear, our findings suggest that the interaction between nitrogen, phosphorus, and BVOCs require further exploration, particularly in phosphorus-limited tropical systems.

Microbial biomass was a significant driver of nearly all gas fluxes in the upland forest, except for methane and isoprene. Previous studies have demonstrated the critical role of microbial biomass in soil gas emissions (Leff & Fierer, 2008; Lamers et al., 2013; Mancuso et al., 2015; Azevedo et al., 2024). For example, dimethyl sulfide (DMS) emissions in Amazonian soils are linked to microbial degradation of organic matter, which often occurs in saturated or anaerobic environments (Jardine et al., 2015; Lehnert et al., 2023). This was evident in the white sand forest, where wetter transects with lower oxygen availability exhibited high DMS emissions, likely driven by anaerobic microbial processes. Conversely, drier transects, such as those in the upland forest, showed DMS consumption, which may result from microbial uptake of carbon in DMS as an energy source (Eyice et al., 2015).

Although microbial community data were unavailable in this study, our findings highlight the dual role of microorganisms as both producers and consumers of BVOCs and sulfur compounds, underscoring the complexity of soil flux dynamics. Several studies have linked microbial diversity to BVOC emissions, suggesting that changes in microbial abundance and activity can both amplify and suppress volatile fluxes (Abis et al., 2020;

Saunier et al., 2020). For example, some bacteria, such as Proteobacteria, Actinobacteria, and Firmicutes, are capable of producing isoprene under stress conditions (Kuzma et al., 1995; McGenity et al., 2018). Although specific microbial pathways were not examined here, the observed patterns in upland and white sand forests reinforce the influence of microbial biomass and community composition on BVOC emissions.

Our results highlight the distinct and interconnected roles of soil nutrients and microbial processes as key factors influencing soil and litter gas fluxes across contrasting Amazonian forest types. Specifically, potassium and carbon were prominent drivers in the ancient river terrace and upland forests, likely reflecting their relatively nutrient-rich soils and plant-microbial interactions. In the white sand forest, where extreme environmental variability and lower nutrient availability are characteristic, phosphorus and microbial activity in sulfur cycling emerged as crucial drivers. These findings underscore the importance of integrating microbial analyses into gas flux studies to gain deeper insights into the complex interactions between soil processes, microbial communities, and atmospheric emissions in Amazonia. Therefore, future research combining soil nutrient manipulations, microbial community profiling, and gas flux measurements will be critical to unraveling these dynamics and predicting how forest gas fluxes may respond to environmental changes."

**Flux results:**

Fluxes and their averages should be reported with associated uncertainty. It would be helpful to update the flux figures (3 & 8) in a way that makes it possible to see the fluxes of Up and AR sites in cases where the axes are overwhelmed by WS.

Regarding the uncertainty, we realize that this is not stated in the Material and Methods. Typical PTR-MS uncertainties are within 10–20% for compounds calibrated with gas standards (Yañez-Serrano et al., 2021). Since there were machine performance issues during this work, we decided to work with somewhat higher uncertainties to take a conservative stance and to ensure that potential effects of instrument performance are adequately captured. More details about how we evaluated the error of the PTR-MS concentration, and how it will be stated in the revised manuscript, are given later in this review.

Regarding your specific question, we have chosen not to include the propagated uncertainty in the figures, since it reduces the clarity of the figures. However, we agree with the reviewer that Figures 3 and 8 can be improved. We propose replacing Figure 3 with the figure below with a double axis. This way, it is easier to note the flux magnitudes of the Upland and Ancient River Terrace Forest.

For Figure 8, we had broken the axis so that upland and ancient river terrace forest would be more evident. However, following your recommendation, we created new plot with a zoom on these forest types:

**Minor comments**

L24-26: This sentence reads awkwardly; suggest revising.

We agree, and we propose to rephrase the sentence as follows:

"Recent studies suggest that the carbon-rich soil-litter compartment plays a significant role in gas fluxes. However, the drivers, variability, and magnitude of these fluxes across different forest types remain poorly understood. This is particularly notable in the Amazon rainforest, the world's largest source of BVOCs, where measurements remain scarce."

L27: It's unclear whether this paper will cover flux of BVOCs between soil and litter (in which case we need to know the direction so we know what uptake or emission means in later parts of abstract), or their respective fluxes with the atmosphere. Please indicate.

The paper covers BVOC exchanges between soil-litter together and the atmosphere, including emission or uptake.

We suggest rephrasing this in the abstract as follows:

In this study, we investigated the net soil-litter gas exchange of BVOC and methane, along with their potential drivers—including nutrient content, microbial biomass, soil temperature, and moisture—across three forest types in central Amazonia: white sand forest (WS), upland forest (UP), and ancient river terrace forest (AR).

L30-33: Unclear whether the summarized results are referring to both soil and litter fluxes.

We understand the confusion and suggest rephrasing it in the text as follows:

"Our results showed distinct soil-litter gas exchange patterns across the forest types. The white sand forest (WS) exhibited both high emissions and consumption of gases, notably high acetaldehyde and methane emissions, along with strong uptake of isoprene and monoterpenes. The upland forest (UP) showed lower overall fluxes, with moderate emissions and consumption of dimethyl sulfide (DMS), isoprene, and acetaldehyde. In contrast, the ancient river terrace forest (AR) presented no significant fluxes."

L33: Of the variables tested, the models suggest these were strongest drivers.

We suggest rephrasing it as follows:

"Among the variables tested, the models indicated that soil moisture and temperature were the strongest drivers of fluxes in WS, whereas microbial biomass was the main driver in UP."

**L44: Are they a compartment together or each a compartment?**

We realize that the meaning of this sentence is not clear. With this sentence, we referred to soil and litter as one combined compartment. In our rewritten Introduction, where this

terminology is better introduced, we have also clarified this sentence, which we suggest being like this:

"The soil-litter together is a compartment that can also play a crucial role in BVOC emissions (Fan et al., 2020, 2024; Bourtsoukidis et al., 2018; Peñuelas et al., 2014; Tang et al., 2019)."

L46: What does essential mean in this context, please be more specific.

We agree that the term "essential" could be more precise in this context. What we mean is that biological processes, such as BVOC release in the soil, are crucial for root-microorganism interaction, while physical processes, such as deposition, influence the storage and exchange of gases with the atmosphere.

We rephrased:

"Within this compartment, multiple biological and physical processes influence BVOC dynamics".

L52: Microbes also consume VOCs as part of their metabolism (not just release).

We agree, and we suggest rephrasing the sentence as follows:

"soil microorganisms produce and consume BVOCs for communication and ecological interactions (e.g., defense and competition), with these compounds also being released as residual metabolic products (Isidorov & Jdanova, 2002; Leff & Fierer, 2008; Liu et al., 2024; Monard et al., 2021)"

L53: CH4 is also consumed by microbes in soil. Clarify in the introduction how you are considering CH4 alongside BVOCs—is it or is it not a BVOC, and if it is categorized as GHG here, why is that important to distinguish? Why are you also measuring CH4 and CO2 here?

We acknowledge this is an important point. Methane is not classified as a BVOC, but rather as a greenhouse gas, as considered by the IPCC (Szopa et al 2021). However, in this study, we included measurements of methane (CH4) and carbon dioxide (CO2) alongside BVOCs to provide a more comprehensive understanding of soil–litter carbon dynamics. CH4 and CO2 fluxes serve as direct indicators of decomposition processes and oxygen availability—processes that can be predominantly aerobic (e.g., CO2 production and CH4 consumption) or anaerobic (e.g., CH4 production) and often involve a complex interplay between the two. While BVOC emissions from the soil–litter layer can be linked to decomposition, they are also associated with plant activity, microbial interactions, and ecological signaling. By measuring both GHGs and BVOCs, we aimed to better capture a broader spectrum of carbon fluxes across forest types and to explore potential interactions between the processes driving BVOC and GHG emissions.

To address this better, we suggest to rewrite this paragraph:

"Greenhouse gases (GHGs), such as methane (CH4) and carbon dioxide (CO2), are produced and consumed by soil microorganisms through key metabolic processes, including methanogenesis, methanotrophy, and respiration (Conrad, 2009; Hofmann et al., 2016). While CO2, but also methane, are not classified as BVOC, they play a crucial role in the overall gas exchange and are included in this study alongside BVOCs to provide a broader perspective of soil-litter gas (carbon) fluxes. This inclusion is also important because environmental factors such as soil moisture, temperature, and

nutrient availability influence both BVOC and GHG fluxes, albeit through distinct-but interconnected-biological and physical mechanisms (Greenberg et al., 2012; Tang et al., 2019; Asensio et al., 2007). These interconnected processes drive the net ecosystem exchange of gases between the soil-litter compartment and the atmosphere, making methane and  $CO_2$  key components for understanding processes driving BVOC flux dynamics."

L70: This reference does not demonstrate this for BVOCs. Please make sure the sentence makes it clear what you are inferring from this paper.

We acknowledge that Onwuka 2018 does not directly address BVOCs but rather discusses the effects of soil type on gas movement and evaporation in general. After restructuring the Introduction, it was decided to remove this sentence completely.

L73-74: Are there additional references that can support the influence of vegetation? I imagine there may be others. It's not clear how vegetation cover is different from plant species referenced in next sentence. Please clarify

By vegetation cover, we refer to the forest structure, including the density of trees and other plant types, and vegetation in the area, which can influence factors such as shading, microclimatic conditions, and nutrient input to the soil. On the other hand, plant species composition is more specific to the plant species occurring in the area This is also important as different species can vary in their contribution to BVOC fluxes, root-microbe interactions, litter composition, and nutrient pool.

Based on the input from the different reviewers, we have completely changed the structure of the introduction so that this direct statement is not in the revised manuscript. However, regarding the influence of vegetation, the following new sentence is suggested to be placed in the Introduction:

"The Amazon Basin is a mosaic of diverse forest types (Oliveira-Filho et al., 2020), each with distinct plant species compositions (Ter Steege et al., 2013), shaped by the region's highly variable soil properties (Quesada et al., 2011; Quesada et al., 2012). Although Amazonian heterogeneity is known to play a critical role in regulating biogeochemical cycles, comparative studies across forest types—especially at the soil—litter interface—are still scarce."

L84: It would be more helpful to cite the specific studies than this review.

As previously noted, the Introduction has undergone a complete structural revision. Consequently, the specific sentence you referred to is no longer present in the revised manuscript. However, we fully accepted your recommendation to cite specific studies.

The following new sentence is suggested to be placed in the Introduction:

"However, these global estimates primarily consider emissions from plants, neglecting potential contributions from soil and litter, which might also include a large variety of BVOC chemical species. This gap is particularly significant given recent evidence that the soil—litter together is a compartment that can also play a crucial role in BVOC emissions (Fan et al., 2020, 2024; Bourtsoukidis et al., 2018; Peñuelas et al., 2014; Tang et al., 2019)."

L94: I don't know that this is a helpful/informative way to start the section: "With a unique set of measurements". It would be more helpful to motivate why the suite of measurements was particularly important for answering the questions.

We agree that the sentence could be more informative and better highlight the relevance of the suite of measurements for addressing the research questions. Below is the revised sentence:

"To address these gaps, we investigated soil-litter BVOC (acetaldehyde, methanol, m/z 42, dimethyl sulfide, isoprene and monoterpenes) and GHG (CH4 and CO2) fluxes, soil and litter nutrient content and microbial biomass, and soil temperature and moisture from three forest types in central Amazonia: (i) ancient river terrace forest - a forest that was flooded in the past and is no longer flooded due to changes in the river course (paleoigapó); (ii) white sand forest (locally called campinarana) - a less common forest type that occupies about 5% of the Amazon basin (Adeney et al., 2016); and (iii) upland forest (locally called terra-firme) - the most common forest in Amazonia, with the highest plant species richness (Emidio et al., 2016; Luize et al., 2018). We aimed to answer the following questions: (i) what is the emission/consumption of BVOCs, CO2, and CH4 in magnitude and chemical diversity, and; (ii) what are the main drivers of soil-litter gas exchanges across these three forest types in central Amazonia (specifically, soil moisture and temperature, nutrient content, and microbial biomass from soil and litter)"

L102: The emission/consumption rates of BVOCs, CO2, and CH4 (rather than gases). Check extra and? Here. List the drivers in (ii) that you tested (out of x, y, and z variables, what are the main drivers).

We thank the reviewer for the comment."And?" was a typographical error that we have now removed.

The sentence was rephrased:

"We aimed to answer the following questions: (i) what is the emission/consumption of BVOCs, CO2, and CH4 in magnitude and chemical diversity, and; (ii) what are the main drivers of soil-litter gas exchanges across these three forest types in central Amazonia (specifically, soil moisture and temperature, nutrient content, and microbial biomass from soil and litter)?"

L141: Please make sure soil chamber measurements are described – we don't know yet what these refer to. What are they and what do they measure? I don't think it's been clarified yet whether this is an in situ study or not, and this should be done before giving details on blanks etc.

We thank the reviewer for the comment and acknowledge that some information is missing in this part. We added details about the chamber and the in-situ measurements. Below is the revised text:

"For each forest type, a PELD-MAUA plot (~1 hectare) (<a href="https://peld-maua.inpa.gov.br">https://peld-maua.inpa.gov.br</a>) was selected, within which two 150 m transects were established in homogeneous areas characterized by consistent vegetation structure, soil characteristics, and topography to minimize spatial variability and avoid pseudoreplication. Along each transect, six sampling points were marked at ~30 m intervals, resulting in a total of 36 soil chamber measurements conducted on consecutive days; although this design was necessary for logistical reasons, it also allowed us to examine the influence of external factors beyond

forest-type differences. Chamber-based methods (Section 2.3; Fig. 1) were used to quantify in situ fluxes of CO2, CH4, and BVOCs from the soil–litter compartment, and three blank chambers with sealed collars were deployed per transect to account for background signals and potential chamber interferences (Fig. 2b)."

**L146: Using a probe?**

Yes, this sensor for soil volumetric water content can be considered a probe.

We add the word "probe" in the text.

"volumetric water content (VWC, %) was measured around the collar five times using a probe (AT SMT150, Cambridge, UK), and the average was calculated."

**L147: What depth of surface soil was collected?**

The collection of surface soil was performed in the organic layer, approximately from the first 5 cm of the soil.

**L151: Meaning they were pooled and homogenized?**

Yes. The term mixed samples refers to samples that were pooled together from two soil collars and homogenized.

Here is the revised sentence:

"Granulometry was determined from pooled and homogenized mixed samples of two soil collars."

L152: We don't know what these bag samples refer to. Please give an overview earlier.

Further details on the methodology are provided in Section 2.3; here, we rewrote to describe the sampling bags.

"Tedlar bags were used to collect gas samples directly from the outlet of the pump connected to the chambers, capturing the air for subsequent analysis of BVOCs,  $CO_2$ , and  $CH_4$ ."

155: I would suggest capitalizing 'Transect' and the names of the sites 'White Sand Forest' etc. as proper nouns. Also 'Section X'

We agree and we corrected the entire text.

L163: Describe the collar materials and size. How deep were collars placed in soil? What does it mean to 'seal with the surrounding soil' in L170, please also describe here. Was the effective volume 21L including the collars? If not, please give range of volumes with collars.

We recognize the need to provide further details about the collar materials, size, positioning, and sealing method. To clarify:

- 1. The flux chambers and collars were made of 100% stainless steel with a PTFE-coated Viton O-ring to ensure a tight seal between the chamber and the collar.
- 2. The collars were positioned above the soil and litter layer (O horizon).
- 3. What we mean by sealed with the surrounding soil refers to ensuring that the collars were gently pressed into the litter and surface soil, and that the soil around

the chamber was carefully pressed against the collar edges to create a tight seal, effectively preventing gas exchange between the inside of the chamber and the outside environment, minimizing any potential leakage. This approach was taken because we avoided pressing the collars too deep, as this would damage roots and therefore increase BVOC emissions from damaged roots, creating an artifact in our measurements.

4. The chamber and collar system, including the collars themselves, had a total volume of 21 L.

We added this information to the manuscript to provide a clear and complete description.

"The flux chambers used in this study were produced by the Max Planck Institute for Biogeochemistry. The soil chamber, consisting of the lid and the soil collar (Fig. 2a), was made entirely of 100% stainless steel, with a total volume of 21 L and a surface area of 855 cm² (0.0855 m²). Two Teflon inlets were connected to the top of the chamber, and inside the chamber was a fan that provided air mixing of the gases in the chamber headspace. A PTFE-coated Viton O-ring was positioned at the edge of the collar over which the chamber was placed, ensuring a tight seal between the chamber and the collar.

Before gas sampling, each collar was carefully installed in a non-invasive manner by gently pressing its edge into the litter and surface soil to minimize disturbance to plant shoots and roots. To further ensure a tight seal preventing any potential leakage, the surrounding soil was carefully pressed against the outer edges of the collar (Aaltonen et al., 2011). This method ensured that the chamber system was effectively isolated from external gas exchange. The chamber and collar were sealed together with multiple clamps to prevent outside air from entering the chamber during measurements. The collars were installed approximately 24 hours prior to sampling."

Fig. 2: The teflon line is illustrated with an arrow out. There is an apparently empty fitting on the top of the lid. Which of these show the air inlet to the chamber, and could you have arrows indicate the direction of the gas flow? The use of arrows to show clamps, soil collar, O-ring, etc seem counterintuitive to have the arrowhead point to the label instead of the named item. Are the litter and soil positioned above the clamps (entire soil collar buried) or could you be more specific about their position within the soil collar and the collar's vertical dimensions in general and wrt the soil?

Thank you for pointing out the confusing aspect of the figure. We have redesigned the chamber figure according to your suggestions. To clarify: the litter and the soil were located inside the soil collar, and the soil collar was not buried; the soil collar was positioned gently on top of the soil to avoid damaging roots. Surrounding soil was used to seal the collar at the edges, following the method of Aaltonen et al. (2011).

How was the blank chamber bottom 'completely sealed'? Does it matter if the blank and soil chambers had different internal volumes because the soil collar was partially in the soil?

The soil and the blank chambers were both used in combination with stainless steel collars. The collar of the blank chamber had the same height and diameter as the sample collars, but had its bottom fully closed. This was implemented to prevent any interaction between the chamber interior and the soil or litter layer. In contrast, the sample collars had open bottoms, allowing direct contact with soil and litter, enabling gas exchange to be captured.

The sample collar was not pushed into the soil. Instead, it was gently placed on the soil, and the surrounding soil was pressed against the outer edges of the collar. As described in the Methods section, this approach follows the paper of Aaltonen et al. (2011) and prevents disturbance of the soil due to collar insertion. Because the soil collar was not pushed into the soil, the volume remained the same, hence the soil and blank chamber have the same volume.

**Consider merging sections 2.3 and 2.4.**

We followed the Reviewer's recommendation, and the section is now as follows:

**"2.3 Flux Chamber Measurements**

The flux chambers used in this study were produced by the Max Planck Institute for Biogeochemistry. The soil chamber, consisting of the lid and the soil collar (Fig. 2a), was made of 100% stainless steel, with a total volume of 21 L and a surface area of 855 cm² (0.0855 m²). Two Teflon inlets were connected to the top of the chamber, and inside the chamber was a fan that provided air mixing of the gases in the chamber headspace. A PTFE-coated Viton O-ring was positioned at the edge of the collar over which the chamber was placed, ensuring a tight seal between the chamber and the collar. To further ensure a tight seal preventing any potential leakage, the surrounding soil was carefully pressed against the outer edges of the collar (Aaltonen et al., 2011). Multiple clamps were used to seal the chamber and the collar together, preventing outside air from entering during measurements. The collars were installed approximately 24 hours prior to sampling to allow the surrounding environment to stabilize.

Gas collection took place in December 2021, during the dry-to-wet season transition. Tedlar bags (CEL Scientific, Cerritos, CA, USA) were used to sample soil-litter gas fluxes (BVOCs, CO2, CH4). Before placing the lid on the collar, the chamber was manually ventilated to minimize collar-induced CO2 accumulation. The chamber was then closed, the internal fan was turned on, and the lid was sealed with clamps. An air sampling pump (GilAir® Plus, Levitt Safety, Ottawa, ON), operated at a flow rate of 500 sccm, ensured continuous flow from the chamber outlet. After 20 minutes of chamber closure with continuous flow, a sampling bag was connected to the outlet of the Teflon pump, and a 5 L sample was collected over 10 minutes. By the end of the 30-minute process, a total of 15 L of air had flowed through the chamber, of which the last 5 L was used for subsequent analyses.

For logistical reasons, measurements were conducted with three chambers simultaneously, pairing two sample chambers with one blank chamber, followed by two additional sets, resulting in the measurement of six samples and three blank chambers per day. Because air was continuously extracted from the chamber headspace by the pump, ambient air entered the chamber through one of the two Teflon inlets. This inlet consisted of a 2 m long open Teflon tube, fixed approximately 2 m above the ground and positioned at the same location for both sample and blank chambers. The setup ensured that both chambers (sample and blank) were diluted or affected by ambient air to the same degree, minimizing potential biases.

After sampling, bags were handled carefully to prevent leakage. Potential compound losses due to adsorption onto the inner walls or diffusion through the bag material were minimized by storing all samples in a dark, stainless-steel box to avoid light exposure, and keeping them in air-conditioned lab containers at low temperatures until analysis. All samples were analyzed on the same day, within a maximum of 8 hours post-collection, following the recommendations of Beauchamp et al. (2008). Gas analysis began with the quantification of BVOCs using a proton-transfer-reaction quadrupole mass spectrometer (PTR-QMS; IONICON Analytik, Innsbruck, Austria). Subsequent analyses of CO2 and CH4 concentrations were conducted using a Los Gatos gas analyzer. Each sample bag was then used to fill stainless steel cartridges (containing Tenax TA and Carbograph 5TD adsorbents), which were later analyzed via thermal desorption gas chromatography coupled with time-of-flight mass spectrometry (TD-GC-TOF-MS; Bench ToF Tandem Ionisation, Markes International, Bridgend, UK). For detailed descriptions of the analytical procedures and results, refer to the Supplementary Material, Sections 3 and 3.1.

L176: The theory/rationale behind the sampling approach should be presented. Clarify whether at this volume and flow rate the approach is considered closed/static or flowthrough? The flux chamber would not have been at steady state after 20 mins (10 min\*0.5 LPM = 5 L removed out of 21 L, so headspace had not turned over even once). So was the goal to sample gases that had accumulated after 20 mins? Please specify in more detail.

We agree with the reviewer that the system has not reached steady state, and for this reason, we avoid using this term. Since waiting for a steady state was not feasible under these field conditions, we decided to use a static system, wherein the gas accumulation was used to calculate the gas flux. The gas sample was collected after 20 minutes and filled for 10 minutes. For this reason, we report the gas collection time as 25 minutes. To ensure constant conditions during the entire chamber closure, the sample flow was initiated immediately after closing the chamber, even while the air of the first 20 minutes was not sampled. In addition, blank chambers of the same material and operated under identical conditions were used to account for potential effects of the chamber material and external factors on flux measurements. This approach allowed us to identify and separate possible external and chamber-related influences on the measured flux.

To clarify this in the manuscript, we suggest the following changes:

"Gas collection took place in December 2021, during the dry-to-wet season transition. Tedlar bags (CEL Scientific, Cerritos, CA, USA) were used to sample soil-litter gas fluxes (BVOCs, CO2, CH4). Before placing the lid on the collar, the chamber was manually ventilated to minimize collar-induced CO2 accumulation. The chamber was then closed, the internal fan was turned on, and the lid was sealed with clamps. An air sampling pump (GilAir® Plus, Levitt Safety, Ottawa, ON), operated at a flow rate of 500 sccm, ensured continuous flow from the chamber outlet. The flow was initiated immediately after chamber closure to maintain constant conditions during measurements. After 20 minutes of continuous flow, a sampling bag was connected to the outlet of the Teflon pump, and a 5 L air sample was collected over 10 minutes. The same procedure was followed for the blank chamber, which was measured under identical conditions. By the end of the measurement, we had obtained two bag samples representing air accumulated over 25 minutes (20–30 min) in each chamber, with the difference between the two bags indicating the contribution from soil and litter fluxes."

Could you have detected gas/BVOC uptake by this method? Please also specify. You do report negative flux values – do you trust these as indicating uptake? Please state.

Yes, the method is suitable for measuring both uptake and emission fluxes. The fluxes are calculated by subtracting the sample chamber mixing ratio (VMR) from the blank chamber mixing ratio (VMRb). When this difference is negative, it indicates a net reduction of gas concentrations in the chamber headspace due to soil and litter uptake. The use of the blank chamber ensures that any potential (negative) differences in concentrations due to external effects are excluded. For this reason, we are confident in these findings.

L186: This is the inlet line to the chamber? Please clarify.

Yes, this line is attached in the chamber. We clarified in the text.

"Because air was continuously extracted from the chamber headspace through one Teflon inlet by the pump, ambient air entered the chamber through the other Teflon inlet. This inlet was connected to a 2 m long open Teflon tube, fixed approximately 2 m above the ground and positioned at the same location for both sample and blank chambers. The setup ensured that both chambers (sample and blank) were diluted or affected by ambient air to the same degree, minimizing potential biases."

Section 2.5. Could you indicate whether this analyzer can be used for high-resolution analysis of mass spectra or whether the results are all analyzed at unit mass resolution for this study? Do the specific masses need to be chosen in advance, and they are chosen at unit mass? If so, how were they chosen? A little background info on this will help.

Regarding the resolution of the PTR-QMS and the selection of specific masses for this study. Below are the clarifications:

**High-resolution analysis versus unit mass resolution:** The PTR-QMS used in this study was operated at unit mass resolution, meaning that detected protonated masses (m/z) were analyzed as integral values. The PTR-QMS used in this study does not allow for distinguishing between isobaric compounds at the same mass due to its unit mass resolution limitation.

**Selection of specific masses and pre-determination:** The masses were selected based on previous studies investigating soil and litter fluxes, as well as the compounds available in the gas standard cylinder used for calibration. Calibrating the PTR-QMS with known concentrations of BVOCs was essential to ensure the accuracy and reliability of our results.

L226-241: Much of this section may be better suited to the discussion. I understand that it is providing rationale for some of the methods, but it also includes the discussion of your results. Reconsider placement.

We agree that the discussion of the m/z 42 and m/z 63 masses includes interpretations that are more appropriate for the discussion section of the article. We have decided to relocate this section to **4.2** ("Differences in gas fluxes across the different forest types"), where we already detail our observations regarding fluxes for compounds related to these masses.

L252: I would suggest performing a sensitivity analysis to prove this. You could take the concentrations from the blank and the dilution information from your flow rate and chamber/aboveground collar dimensions and estimate it.

We have evaluated the influence of the constant inlet/outlet flow on measured gas buildup in the chamber.

For a closed chamber WITHOUT flow, the concentration change (where the calculated fluxes are based on) would be act linear:

$$C(t) = r(t) *t$$

Wherein r(t) is the concentration change per min (concentration min-1), t the time of chamber closure (min).

Now with our camber volume V (21 L) and a constant flow Q (0.5 L min-1), the concentration change under a continuous flow is

$$C(t) = r * V/Q * (1 - e^{-Qt/V}))$$

Solving the **bold part** of the equation with the given parameters gives:

$$C(t) = r^* 21.4$$

Comparing the 2 chambers with flow gives:

Chamber change without dilution = 30 r

Chamber change without dilution = 21.4 r

Expected Dilution on flux rate r = 1-(21.4/30) = 29%

In other words, continuous ventilation reduces the concentration increase by only ~29% compared with a closed, non-ventilated chamber. This dilution would therefore make our flux estimates conservative, reinforcing the conclusion that fluxes from the soil-litter compartment can be larger than previously reported. However, to maintain clarity and focus on the main findings, and because this effect does not alter our overall conclusions, we have chosen not to elaborate further on this point so as not to distract the reader. We suggest changing the text as follows:

"A dilution effect due to the constant sample flow is expected, which would result in a slight underestimation of our fluxes; however, to maintain conservative estimates, we chose not to apply a correction."

L308: Please clarify how you propagated uncertainty to arrive at 3-5 significant figures in your flux results. Add a measure of variability / confidence to the reported fluxes (stdev, ci, etc) throughout the manuscript.

Typical PTR-MS uncertainties are within 10–20% for compounds calibrated with gas standards (Yañez-Serrano et al., 2021). Since there were machine performance issues during this work, we decided to work with somewhat higher uncertainties to take a conservative stance and to ensure that potential effects of instrument performance are adequately captured. We evaluated the error of the PTR-MS concentration measurements as follows:

The error of PTR-MS concentration measurements consists of a systematic part and a statistical part. The systematic error consists of the uncertainty of the calibration gas standard (+-5%), the error of the flow measurements (-5%), and the error of the calibration slope (different for each compound, see table).

The statistical error consists of the repeatability of the concentration measurement during the calibration routine (see table, different for each compound).

|                     | 5,510                           |                |  |  |
|---------------------|---------------------------------|----------------|--|--|
| M/Z                 | error of slope (Calfaktor) in % | precision in % |  |  |
| Methanol (m33)      | 14.83                           | 14.37          |  |  |
| m42                 | 52,40                           | 26.17          |  |  |
| Acetaldehyde (m45)  | 12,52                           | 14.77          |  |  |
| DMS (M63)           | 18,37                           | 13.72          |  |  |
| Isoprene (m69)      | 18,22                           | 6.05           |  |  |
| Monoterpenes (m137) | 20,38                           | 8.45           |  |  |
|                     |                                 |                |  |  |

statistical error

systematic error

Based on these uncertainties, we agree with the reviewer that it is not meaningful to report fluxes with 3–5 significant figures. We therefore suggest reducing the number of significant figures reported throughout the text.

To assess the systematic part of the flux uncertainties, we evaluate the effect of the systematic error on the flux calculation (see formula for flux calculation below).

F=dVMR \* N \* (V / A)\*(1/T)

Since the systematic error (for example, for isoprene, 18.2%) will have the same direction for both bags in the bag pair, dVMR will have the same systematic uncertainty (18.2%) as the individual bags. Since the systematic error is expressed in a percentage, the propagated flux based on the systematic error will remain the same percentage.

It is harder to propagate the statistical error, since they can be in different directions for the blank and the sample bag in the bag pair. Therefore, we first calculated the uncertainty of each bag individually and then calculated the root mean square of them together. After this, we evaluated whether the absolute bag concentration difference ( $\Delta VMR$  = sample bag – blank bag) is larger than this value. For fluxes that did not meet the filter criterion (i.e.,  $\Delta VMR$  was not greater than the combined uncertainty), we assigned a value of zero. This indicates that, within measurement uncertainty, these fluxes are not significantly different from zero. Importantly, a zero value here represents a flux below the detection limit of our combined methods (PTRMS + flux quantification), not a complete absence of flux. Assigning zero rather than discarding these data preserved the full sample size—essential for model robustness given our limited observations—and prevented bias in the average fluxes.

The table below gives an overview of how many flux measurements passed this filter per gas per ecosystem. While for some transects the amount of fluxes above the detection limit was limited, it is important to note the following: applying this combined approach (filter and zero assignment) resulted in a more robust and less noisy dataset for modeling. A comparative analysis of the average fluxes before and after this process showed minor changes, indicating that the overall trends in the fluxes remained unchanged. More importantly, implementing this method in linear models reinforced the main story of our manuscript. Although some individual coefficients or the adjusted R² may have been adjusted due to noise removal, the primary conclusions about the drivers of the fluxes in each forest type persisted, providing a more solid and reliable basis for our interpretations. Here below, we are showing the new chamber plots based on this filter approach, which we propose for the revised manuscript. For completeness, we are also showing the old figures for comparison.

|                       | methanol | mz42 | acetaldehyde | dimethyl_sulfide | isoprene | monoterpenes |  |
|-----------------------|----------|------|--------------|------------------|----------|--------------|--|
| White sand forest     | 12       | 9    | 10           | 10               | 12       | 12           |  |
|                       | 12       | _    |              | _                | 12       | 12           |  |
| Upland forest         | 8        | 6    | 6            | 9                | 11       | 10           |  |
| Ancient terrace river | 9        | 0    | 4            | 8                | 10       | 10           |  |

**Filtered chambers (new suggested figures for revised manuscript):**

**Without filtered chambers (figures as displayed in Discussion paper):**

Filtered chambers (new suggested figures for revised manuscript):

**Without filtered chambers (figures as displayed in Discussion paper):**

In the manuscript, we suggest to add the following lines to 2.5 (PTR-QMS measurements and Los Gatos analyzer measurements)

"Curves were calculated considering the normalized counts per second as a function of the mixing ratio. Previously, some compounds important for soil-litter processes (Peñuelas et al., 2014), - such as acetone, ethanol, and formaldehyde - were considered for this study, but they did not show a good fit, they were excluded from this work. The error of PTR-MS concentration measurements of the 6 presented compounds is expected to consist of a systematic part and a statistical part. The systematic error consists of the uncertainty of the calibration gas standard (+-5%), the error of the flow measurements (+-5%), and the error of the calibration slope (14.8%, 52.4%, 12.5%,

18.4%, 18.2%, 20.4% for resp. methanol, m/z42, acetaldehyde, DMS; monoterpenes and sesquiterpenes). The statistical error is based on the repeatability of the concentration measurement during the calibration routine, and was found to be 14.4%, 26.2%, 14.8%, 13.7%, 6.0%, 8.5% for resp. methanol, m/z42, acetaldehyde, DMS; monoterpenes and sesquiterpenes. The systematic error affects both bags in the same direction, whereas the statistical error can differ between the two bags in a pair. Therefore, to evaluate the uncertainty of the fluxes, we focused on the propagated statistical uncertainty, as described below."

In the manuscript, we suggest adding the following lines to 2.6 (**BVOC & GHG flux** calculation)

"By subtracting the mixing ratios of a blank chamber, dVMR represents the concentration difference attributable solely to soil and litter fluxes, corrected for potential chamber effects or the influence of ambient air entering the system. To ensure data reliability, bag pairs for which the concentration difference (dVMR) was less than or equal to the combined statistical uncertainty (calculated using the Root-Sum-Square method from the individual bag uncertainties) were assigned a value of zero. This approach ensures that only reliably detected fluxes are considered, while retaining the full sample size for modeling purposes. The reported fluxes represent mean values and their corresponding standard deviations for the full sample set within each forest type. Propagated uncertainty ranges for each individual chamber measurement are provided in the accompanying dataset".

Furthermore, when studying the fluxes of different forest types, we applied statistical tests to identify significant differences among them. In response to your feedback, we have revised the manuscript to include these statistical measures of variability—specifically, the standard deviation (SD) and 95% confidence intervals (CI) — for fluxes grouped by forest type and transect. For example, average fluxes will now be reported as:

L308 "Acetaldehyde emissions showed the most significant differences between forest types, with emission averages of 35.87  $\pm$  46.86 mg m-2 h-1 (mean  $\pm$  SD) for the white sand forest, -0.09  $\pm$  0.02 (mean  $\pm$  SD) mg m-2 h-1 for the upland forest, and -0.02  $\pm$  0.008 (mean  $\pm$  SD) mg m-2 h-1 for the ancient river terrace forest."

Fig. 3: The yellow rectangle is not necessary.' It would be helpful to indicate the 'zero' line in flux plots. The fluxes for many VOCs for AR and Up are indistinguishable from zero in this plot. Could you indicate if they are significantly different from zero, and if so, find way to also visualize them? Cases of uptake and emission from these sites are mentioned in the results (e.g., L324, L309), but without quantification and we can't see them on the figures.

Below, we address each point:

- 1. Yellow rectangle: We used the yellow rectangle to distinguish soil moisture and temperature data from BVOC and GHG flux results, as we believed it would help avoid confusion for the reader. However, we acknowledge that it may not be necessary, so we will remove the rectangle.
- 2. Zero line in flux plots. We will revise the plot to include a horizontal reference line at zero.
- 3. Visualizing smaller/negative fluxes: we acknowledge that the relatively high flux values for WS have stretched the y-axis scale. This is why Figure 8 includes a broken y-axis to emphasize smaller fluxes. Additionally, we will make modifications to Figure 3 (as shown above, with double axis) to improve the presentation of values closer to zero and negative fluxes, particularly for UP and AR, as their fluxes are near zero.

**L329-332: This is a discussion point.**

We moved to the discussion (section 4.3).

Fig. 4 and Fig.5: I would suggest using names (not shorthand) for the figure titles (Aluminum instead of al\_soil, pH instead of ph\_soil). If the caption says soil or litter, you don't need to add to each title.

We have made these suggested changes. Below you can find the revised Figure 4 and revised Figure 5:

Revised Figure 5:

L362: This first sentence should be qualified by saying 'only for P and for N but only in litter. You are not showing significant differences between forest types in most cases.

Thank you for pointing this out. We suggest the following revision:

"Microbial biomass (soil and litter), used here as a proxy for microbial activity, showed significant differences between forest types only for soil phosphorus, which was higher in the white sand forest compared to the ancient river terrace forest (no data available for the upland forest). For carbon and nitrogen, significant differences were observed only in litter microbial biomass, with the upland forest exhibiting the highest values and the white sand forest the lowest. However, for most microbial biomass parameters measured across soil and litter, differences between forest types were not statistically significant."

Fig. 7: It would be helpful to construct this PCA for all sites (as shown) but also for each site individually. This could help you explore potentially important variables associated with forest type or variables within a forest type, and whether they align in the same way with the various gases. For this figure, were gas fluxes (or other variables) transformed (e.g., center log transformed) in some way to account for any non-normality?

We confirm that all variables included in the PCA were centered (mean = 0) and scaled (variance = 1) before analysis, ensuring that all variables contributed equally to the components, regardless of their original scales or units. This standardization method automatically accounts for differences in measurement scales or magnitudes across variables and reduces any potential distortions caused by outliers or non-normality.

**About the PCAs separated by forest type:** After careful analysis, we observed that the overall structure of the relationships between variables—such as soil nutrient, litter, and

gas fluxes—and the proportion of variance explained by the principal components remained largely consistent across all individual forest types. For example, the distinction between the conditions of the white sand forest and the other two forests (AR and UP) was already captured in the combined PCA (Figure 7 of the manuscript), and the individual PCAs seem to reaffirm these patterns without introducing significant new insights or contrasts in the direction or strength of variable associations.

L369: This acronym was already defined, and is defined at least 3 times in the paper.

We removed the definition in this part of the text and the other parts with repetition

L472-491: It is not clear how these findings (e.g., soil iron) relate to your questions in the introduction related to gases. It is not brought up again. I would focus the discussion on topics pertaining to the gases.

We acknowledge that the initial characterization of soil and litter properties may be dense, and we understand that its immediate relevance to gas fluxes may not have been sufficiently highlighted. However, the soil and litter properties discussed in this section are important for understanding the observed gas fluxes, as they serve as significant predictors in our linear models (Section 4.3.2, Tables 4, 5, and 6). For example, soil iron and phosphorus were indeed identified as drivers for specific BVOC and GHG fluxes. But, it is right that although we observed associations between some soil nutrients and gas fluxes in our linear models, the mechanisms underlying these relationships remain unclear.

To address your suggestion, we will revise Section 4.1 to more clearly focus the discussion on topics pertaining to the gases.

"Variations in soil and litter properties were observed among forest types, contributing to differences in gas fluxes. In the ancient river terrace forest, high potassium and phosphorus contents in the litter were identified as predictors of BVOC fluxes, such as methanol and monoterpenes. In the upland forest, the dominance of soil iron—resulting from intense leaching and organic matter accumulation (Mosquera et al., 2024; Li et al., 2023)—emerged as a significant predictor of acetaldehyde and isoprene fluxes. However, the mechanism underlying the relationship between iron content and acetaldehyde and isoprene fluxes remains unclear

The white sand forest exhibited distinct soil properties that may have influenced the gas exchange dynamics. Despite its generally low fertility (Mendonça et al., 2015; Demarchi et al., 2022), this forest type displayed unexpectedly high soil phosphorus and carbon concentrations. This nutrient profile, particularly phosphorus, was found to be significantly associated with methanol and isoprene fluxes in the white sand forest. Furthermore, varying iron concentrations (Cornu et al., 1997), along with processes like wet-season leaching and redistribution of elements (García-Villacorta et al., 2016; Franco & Dezzeo, 1994; Demarchi et al., 2022), illustrate how the specific environmental conditions in white sand forests contribute to their distinct BVOC and GHG flux patterns. Collectively, these observed differences in soil and litter properties across forest types, including nutrient levels and composition, represent important factors influencing the unique gas fluxes measured in each ecosystem (García-Villacorta et al., 2016; Gomes Alves et al., 2022)."

L498: These references pertain to other sites, right? Please clarify they aren't references to your sites. Could you discuss the potential drivers in terms of your results first, and

then later you could consider other factors you didn't measure as other drivers that may explain additional variability?

The references mentioned in this section pertain to other study sites and not to our study sites.

We revised the text and add references from our specific site:

"The white sand forest exhibited the highest emissions and consumption of BVOCs and GHGs, accompanied by the greatest chemical diversity in gas fluxes. This elevated chemical diversity likely reflects the distinctive characteristics of the white sand forest, including its unique microbiome, strong seasonality, and specialized species composition (Adeney et al., 2016; Demarchi et al., 2022)."

Additionally, while we can discuss the potential drivers first, we intend to provide an overall background of the forest types, followed by a discussion of the flux variability between forest types, and finally, a presentation of the drivers that may explain and predict these fluxes. We believe that this flow allows the reader to first gain an understanding of the characteristics of each forest type, then to interpret the fluxes, and lastly, to explore the drivers that help explain the observed variability in our fluxes.

While our study addressed the primary drivers influencing soil fluxes, as highlighted in Tang et al. (2019), certain additional factors discussed in the literature could further explain variability in emissions but were not measured.

**Biotic Drivers**

Plant species composition could be considered an important driver as it significantly influences emission patterns. Different vegetation types produce distinct BVOC profiles; for example, Salix arctica-dominated areas emit isoprene, while areas dominated by Cassiope tetragona emit monoterpenes (Svendsen et al., 2016). Litter characteristics, such as dry weight and density, can also act as important drivers of emissions (Mu et al., 2020). Additionally, litter age and quality are relevant, as fresh litter decomposes more readily than older, more recalcitrant material (Cornwell et al., 2008). This difference in decomposition dynamics can influence the temporal patterns of BVOC emissions (Svendsen et al., 2016).

The microbial role is another key driver linked to BVOC emissions. Changes in microbial diversity have been associated with higher total emissions but reduced chemical diversity (Abis et al., 2020). Root exudates also influence emissions, as microbes rapidly use these compounds as carbon and energy sources, directly affecting the rhizosphere (Kuzyakov & Larionova, 2005).

**Abiotic Drivers**

While our study included key abiotic variables such as temperature and soil moisture, other factors could also play a role. For instance, gas transport in the vadose zone driven by changes in atmospheric pressure has been identified as a contributor to flux variability (Elberling et al., 1998; Tillman et al., 2003). This advection flux occurs in conditions where soil moisture near the surface reduces diffusion, thereby amplifying the importance of advection in gas transport (Choi et al., 2002). Despite its relevance, advection-driven transport is often overlooked in soil gas transport models (Smith et al., 2018).

L515: Do you mean soil uptake or soil and litter uptake in your study? Could the difference be due to isoprene emission from your litter? Please consider the absence/presence of litter when you compare to other studies. Pugliese et al., 2023 also reports deposition velocity so you can compare the concentration-independent uptake as well and determine whether there a difference persists when you correct for the atmosphere.

In our study, we worked with soil and litter together, and therefore, the uptake measurements represent the combined contributions from both components. We did not separate the emissions or uptake exclusively from litter or soil, which makes it difficult to isolate the specific contributions from each. The suggestion to account for the absence or presence of litter when comparing our results to other studies is well taken, and we will mention this limitation in the manuscript.

About the potential influence of isoprene emission from litter on the differences observed in our results, this is unlikely to be the reason because dead leaves do not emit isoprene. We found one study attributing isoprene emission to microorganisms in the soil (McGenity et al., 2018; Murrel et al., 2020), if that is the case in our study, we can suggest that microorganisms in soil and litter could be emitting isoprene; still, our results indicate uptake by soil and litter.

About deposition velocity, we cannot calculate it because we do not have atmospheric BVOC concentration measurements. Pugliese et al. (2023) calculated deposition velocities using the ratio of BVOC uptake rates to their ambient concentrations. However, while we are unable to calculate deposition velocities directly, our results demonstrate clear evidence of BVOC uptake based on net negative fluxes. This observation aligns with other studies, such as Trowbridge et al. (2020), which reported negative fluxes as a direct indicator of uptake. Therefore, although we cannot provide *concentration-independent* uptake rates as suggested by the deposition velocity approach, the negative fluxes measured in our study robustly indicate soil and litter uptake.

**L546: I don't know if attributing to 'natural variations' is particularly useful.**

What we tried to express here is that we have a huge spatial variability in the white sand forest and this can explain why we have some chambers emitting while others are uptaking gases.

However, we agree with the reviewer that this is not clear, and we propose to rephrase the sentence as follows:

"However, the discrepancy in white sand forest fluxes, with uptake reported in their study (-0.38 to -0.25 mg m-2 h-1) and emissions observed here, can likely be attributed to the high spatial variability characteristic of white sand forest ecosystems. This variability is strongly influenced by differences in water table depth and soil hydrology (Franco & Dezzeo, 1994; Demarchi et al., 2022)."

L569: Could you expand more on why this may be? Why would the response to soil moisture differ in different soil and forest types?

Soil moisture response varies significantly across Amazonian soil and forest types due to differences in soil properties, rooting depth, microbial communities, and climatic seasonality (Buscardo et al., 2018; Quesada et al., 2011; Davidson et al., 2012). Soils differ in texture and water retention capacity, with sandy soils draining quickly and clayrich soils retaining water longer, which affects moisture availability for plants and microbes (Quesada et al., 2011; Umair et al., 2025). Rooting depths also vary among species and forest types; some Amazon trees access deep soil moisture, especially in upland forests, which buffers them from surface drying during dry seasons (Nepstad et al., 1994). Different rooting patterns and water uptake strategies lead to divergent transpiration (Teuling et al., 2005), influencing how soil moisture dynamics affect soil biogeochemical processes (Koch et al., 2019), which can impact gas fluxes. In addiction, microbial communities exhibit variable respiration and metabolic responses to drying and rewetting cycles, causing spatial heterogeneity in soil emission responses (Hu et al., 2021; Placella et al., 2012). Climatic variation in the Amazon, like gradients in precipitation and seasonality, also impact soil water storage and soil moisture differently in each ecosystem (Malhi et al., 2009). Collectively, these factors can explain why soil moisture sensitivity varies according to soil and forest type in the region.

L582: This is not a strong way to conclude your discussion paragraph here. The temperature only varied by 2C across all your sites, which is very little. You could calculate the sensitivity of expected soil VOC emissions to that temperature change based on other papers, and I think it would be small. More than temperature, light penetration to the surface may have an effect. Could you discuss this and try to arrive at a more specific conclusion even if is just for 1-2 gases?

In this paragraph, our goal was to emphasize the complex interplay between soil temperature and moisture, rather than attributing BVOC flux changes only to temperature. We acknowledge that the observed variation in soil temperature across our sites (approximately 2°C) is relatively small which, in addition, clouds to what extent we can study the *direct* effect of temperature change. Because temperatures are relatively stable in tropical ecosystems, the limited variation that does occur appears to be strongly induced by external factors, such as previous rain events affecting soil moisture. This strong interdependence between soil moisture and soil temperature was also observed by van Asperen et al. (2024) at a nearby field site, who emphasized the difficulty of disentangling the effects of soil moisture and soil temperature in tropical environments.

We like the suggestion to calculate the sensitivity of temperature changes on VOC emissions, even only for 1 or 2 gases, based on findings from other studies, and have looked for appropriate literature to do so. However, as far as we are aware, no other study has studied the *direct* temperature effects under field conditions on soil VOC emissions, and most studies only state tendencies (for example VOC emissions increased with temperature, (Trowbridge et al., 2020; Legros et al.,2025; Rinnan et al., 2020; Monard et al., 2021) but don't provide an actual number - Instead, they show statistically significant correlations, model effect sizes, or qualitative increases.

The reviewer raises a second important issue, namely the role of light penetration. In our case, while direct light was not available during the measurement itself (opaque chamber), it still could influence the local VOC dynamics *before* the chamber measurement. Furthermore, given the unique and heterogeneous canopy structures of the forest types studied here, we suspect that differences in light penetration may act as another driver of flux variation. Unfortunately, our study did not include measurements of light penetration to the soil surface, which limits our ability to evaluate its role explicitly.

We agree with the reviewer that both raised concerns are important and should be addressed better. With regard to temperature, we suggest emphasizing that, although our results show a relationship between certain VOCs and temperature, we do not claim this to be a *direct* causal link. Rather, temperature should be viewed as one of several interacting factors influencing the complex dynamics of VOC emissions. With regard to

light penetration, we suggest including this better into the discussion. We suggest the following new text:

For certain BVOC fluxes a positive association with soil moisture and a negative association with soil temperature was observed. However, since elevated soil moisture frequently coincides with lower temperatures—particularly in tropical ecosystems (van Asperen et al., 2024)—it remains challenging to disentangle whether increased fluxes are primarily driven by temperature or moisture. This interplay is further complicated by the biological nature of BVOC exchange processes, which generally exhibit positive temperature dependence. Elevated temperatures can enhance both BVOC emissions and biological uptake (Baggesen et al., 2022). If the temperature sensitivity of uptake exceeds that of emission, this may result in reduced net emissions or even a net sink for BVOCs (Asensio et al., 2007; Peñuelas et al., 2014; Jiao et al., 2023).

In the present dataset, the temperature range was relatively narrow, precluding clear inference regarding direct temperature effects. Consequently, the observed variability could reflect natural fluctuations rather than a distinct thermal response. Moreover, although radiation was not quantified during the campaign, its potential influence should be considered. Despite the use of opaque chambers, the surrounding ecosystem—and thus the prevailing BVOC flux dynamics—may still be modulated by incoming radiation. Given the heterogeneous canopy structure within and among the investigated forest types, spatial heterogeneity in light penetration likely constitutes an additional driver of BVOC flux variability. Overall, these results highlight that temperature, moisture, radiation, and biological activity are closely linked, and their combined effects on BVOC fluxes are difficult to separate under natural field conditions.

In section 4.3.2 please be very clear where the studies you cite are for plant emissions of VOCs versus soil or litter studies. I would suggest streamlining this discussion and being very clear about how previous work informs the fluxes you observed from soil and litter.

We clarified in the text whether the studies we reference pertain to plant VOC emissions, soil VOC emissions, or litter VOC emissions. We revised Section 4.3.2 to explicitly distinguish the sources referenced and streamline the discussion.

Below follows the revised topic of the discussion, in bold is the main corrections:

"In general, ancient river terrace and upland forests showed many similarities in the predictors of certain gases. In contrast, other drivers were found in the white sand forest. Here we discuss the observed key drivers (soil potassium, carbon, phosphorus and microbial biomass) for each forest type. For ancient river terrace and upland forests, soil potassium was a significant factor influencing soil fluxes, being identified as a predictor of methanol and monoterpenes. In addition, it was also identified as a predictor of m/z 42 fluxes in the upland forest. Although we did not find studies directly relating BVOC and GHG soil or soil-litter fluxes to soil potassium content, potassium is an essential macronutrient for plant growth and metabolism. Its availability is known to affect plant physiological processes (Wang et al., 2013), and its cycling within the soil environment, often mediated by microbial activity, influences potassium's uptake by plants (Mazahar & Umar, 2022). These plant- and soil-mediated processes can, in turn, indirectly influence the BVOC production and release observed within the soil-litter compartment of our study, by affecting the overall ecosystem health and the quality of organic matter available for decomposition. In the upland forest, our methane consumption fluxes

[revised manuscript text omitted]

From the few studies investigating the relationship between microorganisms and BVOC dynamics, it has been shown that some Proteobacteria, Actinobacteria, and Firmicutes can produce isoprene (Kuzma et al., 1995; McGenity et al., 2018). Bacillus subtilis can produce isoprene in response to stress; however, the mechanism is still not clear (McGenity et al., 2018). Some studies have shown that reduced microbial diversity, whether in soil (Abis et al., 2020; Sillo et al., 2024) or associated with plant surfaces (Saunier et al., 2020), can increase BVOC fluxes and alter the chemical composition of emitted compounds. Although microbial community data were unavailable in this study, we suggest that potential differences in microbial diversity have influenced emission and consumption patterns. Therefore, we strongly recommend that future studies investigate gas flux measurements with microbial community analyses to better understand these dynamics."

L653: Making measurements on equidistant points does not influence the inherent spatial heterogeneity of a system. Please clarify in methods how you selected homogeneous areas—how and why was this done?

We can provide more details about the selection of the areas. Within our research site, we are part of the Maua-PELD project (<a href="https://peld-maua.inpa.gov.br">https://peld-maua.inpa.gov.br</a>), which has established 1-hectare plots in the different forest types. However, due to the large size of these plots, we observed significant variability within them, both in terms of topography and vegetation structure. To deal with this heterogeneity, we chose to use transects within the most homogeneous areas of these plots. It is important to mention that the selection of homogeneous areas was guided by ecological principles to minimize spatial heterogeneity and avoid pseudoreplication (Hurlbert, 1984; Oksanen, 2001).

This approach reduces variability within sampling plots, allowing for more robust comparisons between forest types. Transects were established in homogeneous regions with clear criteria, such as consistent vegetation structure, soil characteristics, and topography. The sampling points within the transects were then distributed equidistantly to ensure spatial independence.

We rephrased this sentence:

"For each forest type, a PELD-MAUA plot (~1 hectare) (<a href="https://peld-maua.inpa.gov.br">https://peld-maua.inpa.gov.br</a>) was selected, within which two 150 m transects were established in homogeneous areas characterized by consistent vegetation structure, soil characteristics, and topography to minimize spatial variability and avoid pseudoreplication. Along each transect, six sampling points were marked at ~30 m intervals, resulting in a total of 36 soil chamber measurements conducted on consecutive days; although this design was necessary for logistical reasons, it also allowed us to examine the influence of external factors beyond forest-type differences. Chamber-based methods (Section 2.3; Fig. 1) were used to quantify in situ fluxes of CO2, CH4, and BVOCs from the soil–litter compartment, and three blank chambers with sealed collars were deployed per transect to account for background signals and potential chamber interferences (Fig. 2b)."

What drives the differences between forest types? How could forest-type-specific fluxes be inferred, or is the recommendation to go inventory all forest types?

This is an excellent question, and we thank the Reviewer for raising this important point. Prior to conducting this study, we hypothesized that a single environmental driver—such as soil moisture—could explain the observed fluxes across all forest types. We expected that soil water availability would act as a universal regulator, consistently influencing gas

fluxes regardless of forest type. Under this assumption, differences among forest types would be primarily attributed to variations in flux magnitude and/or the chemical composition of BVOCs, but still fundamentally driven by soil moisture. However, our findings revealed that the soil–litter system is far more complex than we had initially anticipated. Rather than being governed by a single driver, it operates as a dynamic and intricate system—a 'universe' of interacting biological and physicochemical processes.

So, below are the main conclusions about that:

The differences between forest types in our study are primarily driven by variations in soil nutrient availability, microbial biomass, and environmental conditions such as soil moisture and topography. These factors contribute to the distinct flux patterns observed. Since soil and litter gas fluxes result from a combination of biological (microbial activity, vegetation contributions) and abiotic (e.g., soil properties and moisture) processes, forest-type-specific drivers emerge as a result of these interactions.

Considering the complexity of Amazonian forest types and the lack of observational studies, the ideal approach would be the inventory of all forest types. But considering the logistical challenges, one approach could be the identification of key drivers of variability that consistently explain differences among the most representative forest types. For example:

- 1. Soil properties: Measuring nutrient content (e.g., phosphorus, potassium, carbon), pH, and soil texture can provide critical insights into soil processes contributing to BVOC and GHG fluxes.
- 2. Vegetation structure and functional traits: Linking forest canopy coverage, species composition, and root systems to observed fluxes can help predict forest-specific contributions.
- 3. Microbial communities: Variability in microbial biomass and composition can be a driver of gas production and consumption dynamics, particularly for methane and dimethyl sulfide, for example.

From our findings, we suggest future research efforts to focus on monitoring these parameters in a representative subset of the most common forest types. By identifying the most relevant forest traits and soil characteristics, general patterns of BVOC and GHG fluxes could be inferred for similar forest systems without the need for inventories. Our study was a first attempt to identify key drivers of soil-litter gas exchange across different forest types, and our results point out these three aspects mentioned above.

Section 4.5: This section should be streamlined to focus on the results more than making a separate broader point that would be more well suited for a commentary or opinion piece. The point is already made succinctly in Section 5.

We will focus more on the results themselves rather than making broader points, as suggested. The revised Section 4.5 is presented below (in bold), with a concentrated focus on how our findings contribute to understanding BVOC and GHG fluxes in white sand forest ecosystems.

"This study showed large variability across forest types and unexpectedly high BVOC emissions from the white sand forest. Relatively few studies have been performed on white sand forests, which can partly be explained by the challenging conditions of this forest type, such as flooding and extreme temperatures, which require specific infrastructure for data collection (Adeney et al., 2016). In addition, the complex nature of this ecosystem - characterized by scattered patches of differentiated vegetation distributed within extensive upland forests (Demarchi et al., 2022) - can make access to

these sites even more difficult. It is acknowledged that BVOC and GHG studies in white sand forests are limited: so far, only one study has provided data on BVOC fluxes with soil incubation lab measurements (Bourtsoukidis et al., 2018), and another measuring GHGs in situ (van Asperen et al., 2020). Despite representing only 5% of the Amazon basin area (Adeney et al., 2016) and 8% of the Reserve of this study (Demarchi et al., 2022), white sand forests are extremely important environments. Their sandy, nutrientpoor soil type has created a challenging ecosystem for plant growth (Fine & Baraloto, 2016), and this unique condition has selected specialized flora and fauna adapted to thrive in these ecosystems (Adeney et al., 2016; Demarchi et al., 2022). This high level of endemism contributes significantly to the overall biodiversity of the Amazon Basin (García-Villacorta et al. 2016). Moreover, white sand forests have been shown to play a crucial role in the chemistry of dissolved organic matter (DOM) in Amazonian blackwater rivers, linking terrestrial ecosystem processes to aquatic biogeochemistry (Simon et al., 2021). Our results demonstrated that white sand forest gas fluxes clearly depend on physical drivers (more than other forest types), which indicates a possible sensitivity to upcoming climate extremes. For example, the high BVOC emissions observed after a rain event in transect 2 of the white sand forest align with its shallow water table, a characteristic identified as a potential hydrological refuge during droughts (Costa et al., 2023). This dependence on physical drivers suggests a significant role for white sand forests in both the emission and uptake of BVOCs and GHGs, thereby influencing regional carbon and trace gas fluxes. Notably, high atmospheric isoprene concentrations have been reported in the northwestern Amazon, a region characterized by extensive white sand forest cover (Wells et al., 2022; Borges et al., 2014), which corroborates the potential importance of this ecosystem's findings in flux studies. Therefore, it is crucial to recognize that white sand forests have historically been neglected, even with their critical role in regulating the carbon cycle and maintaining Amazonian biodiversity (Rossetti et al., 2019). As for BVOC and GHG measurements, even less information is available for this ecosystem. However, our results suggest that white sand forests may play a significant role in both the emission and uptake of these compounds, reinforcing their importance in regional carbon and trace gas fluxes. Notably, a recent study reported high atmospheric isoprene concentrations in the northwestern Amazon throughout most of the year (Wells et al., 2022) — a region characterized by extensive and continuous white sand forest cover (Borges et al., 2014). Together, these findings highlight the need to better integrate white sand forests into future flux studies and atmospheric models."

**Supplement**

Is there a discrepancy in the sampling rate, and thus total volume sampled, between section 2.4 and the supplemental methods section 3? Confirm that the determined concentrations accounted for the air volume sampled.

We would like to provide clarification by distinguishing between two steps in our methodology: (i) the air sampling into collection bags, and (ii) the subsequent sample analyses.

- (i) Air sampling: Air was sampled into each bag at a constant flow rate of 500 sccm, resulting in a total sampled volume of 5 L.
- (ii) Sample analysis: The collected air samples were then analyzed using three different methods:
  - PTR-QMS: Sampled at 50 sccm for 5 minutes (total volume: 250 ml),

- Los Gatos analyzer: Sampled at 0.1 lpm for 3 minutes (total volume: 300ml),
- Adsorbent cartridges for offline GC analysis: Filled at 200 sccm for 10 minutes (total volume: 2 L), as detailed in Section 3 (Supplemental Methods).

Both the PTR-QMS and the cartridge-based offline sampling require low flow rates to ensure accurate quantification by the PTR-QMS and effective adsorption onto the cartridge material, thereby preventing breakthrough of target compounds. It is important to note that we assumed homogeneous mixing ratios within each bag. This implies that the sub-samples taken for different instruments should reflect similar mixing ratios. Importantly, the determined concentrations for both techniques account for the total air volume sampled, ensuring that the reported values are accurate and comparable. We will clarify this distinction further in the manuscript to avoid any misunderstanding.

**What is the purpose of Fig. S2? Please include uncertainty bars. How are these results included in the main manuscript?**

The purpose of Figure S2 is to provide additional insights into the chemical diversity of BVOC fluxes from soil and litter across the three forest types using data obtained from the GC-MS analysis. These results highlight chemical diversity, particularly in the white sand forest, which displayed greater chemical diversity compared to the other forest types.

Figure S2 was not included in the main manuscript because their fluxes were significantly lower compared to those measured with the PTR-QMS. This data was primarily used for qualitative comparisons between forest types. Additionally, given the large amount of data presented in this study and the complexity of the paper, we opted to focus on the main highlights to maintain conciseness.

Below is the revised figure following your recommendations:

⇒ AR (Ancient River Terrace Forest) ⇒ WS (White Sand Forest) ⇒ Up (Upland Forest)

**What did the blank chamber measurements look like? Were concentrations fairly steady across all the samples?**

The blank chamber concentrations were generally stable, particularly for DMS, CH4, and CO2. Interestingly, some blanks showed elevated concentrations of certain compounds, such as acetaldehyde in the UP site, and isoprene and monoterpenes in the WS site. Although initially surprising, this is likely not unexpected. Our study shows that the soil and litter emit substantial and highly variable amounts of BVOCs. As reported in previous studies, other ecosystem components are also known to emit high levels of BVOCs (Svendsen et al., 2016; Yanez-serrano et al., 2020; Zeng et al., 2022; Duan et al. 2024), Therefore, it is reasonable to expect that the ambient air surrounding the chambers—sampled by the blanks—contains elevated BVOC concentrations.

Indeed, we often observed that both the blank and the corresponding sample chamber in a pair showed elevated levels, suggesting influence from background conditions. To ensure that our measurements captured BVOC emissions specifically from the chamber's soil and litter, and not just background variability, we implemented two key controls. First, the Teflon inlet lines for the sample and blank chambers were positioned at identical locations, ensuring that both experienced the same background air conditions (Fig 2b). Second, we applied a filter that excluded chamber pairs with concentration differences smaller than the combined uncertainty of the individual measurements, which was explained in more detail earlier in this review. This ensured that only significant differences—attributable to chamber soil and litter emissions—were considered.

As earlier described in the review, we suggest elaborating on these 2 key controls in the Material and Method:

**In 2.4**

As air was continuously extracted from the chamber headspace, both the blank and sample chambers were fitted with 2 m-long open Teflon tubes to allow the inflow of replacement air. The tubes were installed at identical heights (approximately 2 m above ground level) and at the same location. Measurements were conducted simultaneously to ensure that both chambers experienced identical background air conditions and were equally influenced by ambient air dilution.

**In 2.6**

"By subtracting the mixing ratios of a blank chamber, dVMR represents the concentration difference attributable solely to soil and litter fluxes, corrected for potential chamber effects or the influence of ambient air entering the system. To ensure data reliability, bag pairs for which the absolute concentration difference (dVMR) was less than or equal to the combined statistical uncertainty (calculated using the Root-Sum-Square (RSS) method from the individual bag uncertainties) were assigned a value of zero. This approach ensures that only reliably detected fluxes are considered, while retaining the full sample size for modeling purposes (freedom of degrees).""

4.1: I don't understand the difference between Fig. S3a and S3b is even after reading the text a few times. Would you please make this more obvious?

The difference between Figures S3a and S3b is about the variables being analyzed in the Pearson correlation matrix: Figure S3a presents the Pearson correlations only among the predictor variables, which include soil and litter characteristics, microbial biomass, and soil moisture and temperature. This figure highlights the relationships between the environmental drivers in our study. Figure S3b presents the Pearson correlations between the predictor variables and the response variables, which are the BVOC and GHG fluxes we measured. This figure shows how each gas flux is associated with specific environmental drivers.

We revised the text in the supplementary material to better explain the differences between Figures S3a and S3b. Below is the revised version of Section 4.1 from the supplementary material:

"Firstly, we investigated the relationships among the potential predictor variables in the white sand forest ecosystem. These predictor variables include soil and litter characteristics, microbial biomass, soil moisture, and temperature. The Pearson correlations for these variables are presented in Figure S3a, which highlights the interactions and dependencies between the environmental drivers only.

Secondly, we analyzed the correlations between the potential predictor variables and the response variables, which are the BVOC and GHG fluxes measured in this study. The Pearson correlations for these relationships are shown in Figure S3b, which specifically highlights how the fluxes of BVOCs and GHGs are associated with environmental drivers. This distinction between Figure S3a (predictor-predictor correlations) and Figure S3b (predictor-response correlations) underscores the separate analyses of interactions among environmental variables and their direct associations with gas fluxes."

**Fig. S6, S7: What is the p-value on this correlation? Is it significant? These differences in temperature are small.**

The difference in temperature is very small and the main differences are in the values of the soil moisture. We have calculated the correlation and the p-value.

For figure S6, the correlation between soil moisture and soil temperature (Fig. S6) was moderate (r = 0.371) and statistically significant (p = 0.027, p = 36). Although the differences in temperature were small, this relationship highlights potential interactions between soil moisture and soil temperature in the forest types.

When we separately analyzed the forest types:

In the white sand forest, there was a strong negative correlation (r = -0.704, p = 0.013, n = 12), indicating that higher soil moisture was associated with lower soil temperature. This pattern was the same in the upland forest, where a very strong negative correlation was observed (r = -0.799, p = 0.003, n = 12). In the ancient river terrace forest, however, the correlation was weak and not statistically significant (r = -0.188, p = 0.570, n = 12). These results suggest that the relationship between soil moisture and temperature varies considerably depending on forest type, potentially reflecting differences in soil composition, canopy structure, and hydrological conditions among forests.

For figure S7, the relationship between soil moisture and  $CH_4$  fluxes varied among forest types. In the white sand forest, we observed a strong positive correlation (r = 0.776, p = 0.0034, n = 12), indicating that  $CH_4$  fluxes significantly increased with higher soil

moisture. This trend was less evident in the upland forest, where the correlation was weak and not significant (r = 0.269, p = 0.400, n = 12). In the ancient river terrace forest, we found a moderate positive correlation (r = 0.498, p = 0.100, n = 12), but it was not statistically significant. These differences suggest that the relationship between  $CH_4$  fluxes and soil moisture is ecosystem-specific, being especially pronounced in the the white sand forest.

When all the forest types were analyzed together, the correlation between soil moisture and  $CH_4$  fluxes was positive and moderate (r = 0.528) and statistically significant (p = 0.0015, n = 36), which was likely influenced by the strong and significant correlation in the white sand forest.